# Combined Radiations: Biological Effects of Mixed Exposures Across the Radiation Spectrum

**DOI:** 10.3390/biom15091282

**Published:** 2025-09-05

**Authors:** Orfeas Parousis-Paraskevas, Angeliki Gkikoudi, Amer Al-Qaaod, Spyridon N. Vasilopoulos, Gina Manda, Christina Beinke, Siamak Haghdoost, Georgia I. Terzoudi, Faton Krasniqi, Alexandros G. Georgakilas

**Affiliations:** 1DNA Damage Laboratory, Physics Department, School of Applied Mathematical and Physical Sciences, National Technical University of Athens (NTUA), Zografou Campus, 15780 Athens, Greece; ge17035@mail.ntua.gr (O.P.-P.); angelikigkikoudi@mail.ntua.gr (A.G.); svasilopoulos@mail.ntua.gr (S.N.V.); 2Laboratory of Health Physics, Radiobiology & Cytogenetics, Institute of Nuclear & Radiological Sciences & Technology, Energy & Safety, National Centre for Scientific Research “Demokritos”, 15341 Athens, Greece; gterzoudi@rrp.demokritos.gr; 3Physikalisch-Technische Bundesanstalt (PTB), Bundesallee 100, 38116 Braunschweig, Germany; amer.al-qaaod.ext@ptb.de (A.A.-Q.); faton.krasniqi@ptb.de (F.K.); 4Radiobiology Laboratory, “Victor Babes” National Institute of Pathology, 99-101 Splaiul Independentei, 050096 Bucharest, Romania; gina.manda@ivb.ro; 5Bundeswehr Institute of Radiobiology, University of Ulm, Neuherbergstraβe 11, 80937 Munich, Germany; christinabeinke@bundeswehr.org; 6ABTE/ToxEMAC Laboratory, University of Caen Normandy, F-14050 Caen, France; siamak.haghdoost@unicaen.fr

**Keywords:** mixed radiation fields, ionizing radiation, non-ionizing radiation, DNA damage, combined radiations, simultaneous irradiation, synergistic effects, radiation protection, cancer risk

## Abstract

Combined radiation exposures—pairings of ionizing and non-ionizing radiation—are increasingly relevant in medical, spaceflight, and environmental contexts. This systematic review evaluates their radiobiological effects and therapeutic applications, focusing on synergistic interactions and underlying biological mechanisms. A comprehensive search of PubMed, Google Scholar, Semantic Scholar, bioRxiv, and Europe PMC identified studies published from the 1960s through 2025. Eligible studies assessed biological responses to different radiation types applied either simultaneously or within 24 h, with minor exceptions. A total of 172 studies were included and categorized into radiobiological, therapeutic, and space radiation domains. Due to the predominance of mechanistic research, no formal risk-of-bias tool was applied; methodological limitations were assessed qualitatively. Findings were synthesized narratively by radiation type and domain. Synergistic and additive effects were frequently observed, with responses influenced by dose, sequence, radiation type, and DNA repair dynamics. Therapeutic combinations often enhanced efficacy, while space radiation studies revealed multifaceted biological damage. This review provides a consolidated reference for advancing research and applications involving combined radiation exposures, emphasizing the need for mechanistic insight and standardized protocols in therapy, radiation protection, and spaceflight. This study was funded by project 21GRD02 BIOSPHERE (European Partnership on Metrology, Horizon Europe) and reported per PRISMA 2020 guidelines; no protocol was registered.

## 1. Introduction

Different types of radiation can be combined in various contexts, whether therapeutic or experimental; they may be applied to biological or non-biological targets, simultaneously or sequentially, with intervals between irradiations ranging from minutes to weeks. The dose of each combined radiation type is another important factor, as are other dosimetric features of a given experimental configuration. For biological targets, a number of measurable endpoints are used to assess the induced effects, which are typically the primary focus of related studies. “Combining different radiation types” thus constitutes a broad set of practices, involving a variety of methods developed and applied for distinct purposes.

Although some narrative reviews exist focusing individually on conventional radiobiology, space radiobiology, or therapeutic radiation combinations, to our knowledge, none provides a systematic analysis of combined exposure types across biological systems and application domains. In many respects, this systematic review pursued an ambitious aim: to map a curated body of studies on conventional and space radiobiology, as well as the therapeutic use of combined radiation exposures, analyzing their qualitative and quantitative features.

More specifically, this review aimed to identify, categorize, summarize, and analyze the results of eligible studies involving biological systems (human, animal, and in vitro) exposed to mixed radiation types in therapeutic, conventional, and space radiobiological contexts. As a result, it provides a concise overview of the diverse processes and outcomes that define the multifaceted applications of combined radiation exposure in biological systems, reflecting the current scientific understanding of the field.

## 2. Methods

The challenges impeding a rapid compilation of all relevant publications required for such an extensive review arise from a range of factors, including the dispersion of publications across a wide variety of databases, shifting terminology, and evolving epistemological frameworks. The selection criteria applied during the literature review are presented here to clarify the rationale behind the inclusion of each study in the compiled data set. First, we determined which types of combined radiations to include—specifically, which wavelength and/or energy ranges were relevant. Our focus aligned with conventional and space radiobiology, as well as several combined therapeutic modalities. Consequently, other forms of mixed exposure, such as environmental exposure to extremely low-frequency electromagnetic fields (ELF-EMF), microwave radiation, or radiofrequency electromagnetic fields (RF-EMF) in combination with other types were knowingly excluded from this review. Similarly, combinations involving radiation and non-radiation agents, such as chemicals or microgravity, were also excluded, as they fall outside the scope of radiation–radiation interactions. In rare cases where non-radiation factors were present (e.g., the NASA Twins Study), the inclusion of those publications was primarily justified by their focused investigation of combined radiation exposures, with any additional variables considered secondary.

Regarding the combinatorial nature of the exposures, studies involving the sequential application of radiation with time intervals longer than those typically defining genuine combinatorial exposure (e.g., >24 h) were mostly excluded. That said, nearly all included studies involved either simultaneous exposures or sequential irradiations administered within 24 h of each other.

In reviewing therapeutic studies specifically, it became clear that the term “combined modalities” is frequently applied to regimens in which different radiation types are administered with substantial time intervals between them—often separated by several days or even weeks—rather than in a truly sequential or simultaneous fashion. To maintain consistency with our definition of combinatorial exposures as involving either concurrent delivery or close-sequence administration (≤24 h apart but for rare cases specifically annotated), such studies were generally excluded from this data set. Only protocols where different radiation modalities were applied in the same treatment timeframe—not exceeding 24 h—were included to avoid conflating simple sequential boosts with genuine mixed exposures.

The radiation types included in the search were near-infrared (NIR), visible (VIS), ultraviolet (UV), X-rays, gamma rays (γ), alpha radiation (α), beta radiation (β), protons (p), neutrons (n), and heavy ions. In addition, space-relevant mixed fields—such as high atomic number and energy (HZE) particles and simulated ion beam combinations (e.g., 5- or 33- beam GCRsims)—were incorporated to reflect realistic space radiation exposures. Therapeutic modalities considered included conventional radiotherapy (RT), intensity-modulated radiation therapy (IMRT), boron neutron capture therapy (BNCT), gadolinium neutron capture therapy (GdNCT), carbon ion radiation therapy (CIRT), radionuclides therapy (RNT), proton therapy (PT), photodynamic therapy (PDT), photothermal therapy (PTT), near-infrared photoimmunotherapy (NIR-PIT), electron beam therapy (EBT), fast neutron therapy (FNT), megavoltage X-ray therapy (megavoltage XRT), and californium-252 neutron therapy (Cf-252 NT).

The methodological approach for the literature review involved several phases. Initially, databases and search engines—such as PubMed, Google Scholar, Semantic Scholar, bioRxiv, and Europe PMC—were searched manually using combinations of keywords, Boolean operators, and filters, covering publications from the 1960s to 20 June 2025 (the date of the final search for all databases). This temporal range emerged organically due to the historical spread of relevant studies and is indicative of the sustained and evolving nature of radiobiological research over multiple decades.

The majority of results were retrieved through PubMed, which served as the primary database for this review. To capture the diversity of radiation pairings and biological systems represented in the literature, over 50 manual keyword searches were conducted using tailored combinations of terms relevant to specific exposures, contexts, and biological endpoints. Representative full PubMed search strings included the following:(“alpha radiation”[tiab] OR “alpha particles”[MeSH Terms]) AND (“X-rays”[MeSH Terms] OR “X radiation”[tiab]).(“simultaneous”[tiab] OR “sequential”[tiab]) AND (“irradiation”[tiab] OR “radiation”[MeSH Terms]).“proton therapy”[tiab] AND “targeted radionuclide therapy”[tiab].“space radiation”[tiab] OR “simulated galactic cosmic rays”[tiab] OR “GCRsim”[tiab].

Similar search logic was applied across all other databases consulted. Filters used during searches included article type (journal articles) and language (English), with no species filters or field-specific restrictions applied. The “related articles” or “related results” features were also used to identify additional relevant publications. For radiation pairing categories initially represented by only one or very few reports, targeted follow-up searches were conducted to obtain a more representative sample. Despite these efforts, some combinations remain represented by a single entry; however, their inclusion was preferred over omitting those radiation combinations entirely from the review.

At least four reviewers independently screened titles and abstracts for eligibility. Discrepancies were resolved by re-evaluating studies against the predefined inclusion criteria and, where appropriate, through discussion of their relevance, representativeness, or scientific value in diversifying the data set—particularly with respect to biological systems or radiation combinations. The data extraction process involved up to three contributors. Data were collected manually into structured tables, with contributions verified and refined iteratively during the review process. Final consistency checks and consolidation were conducted by the lead author to ensure alignment with predefined data fields and classification categories. Following close reevaluation, 18 initially included reports were substituted with more relevant or representative studies based on refined eligibility interpretations. The final data set comprises 172 studies.

A master table was created, consisting of columns detailing the combined radiation types and doses, biological systems, biological endpoints, and effects recorded for each study. Abstracts and relevant sections of the selected publications were reviewed to extract key data, which were organized into structured entries corresponding to the table’s columns. Extracted biological outcomes included DNA damage, oxidative stress, cell viability, apoptosis, cell cycle arrest, gene expression changes, and, where applicable, organism-level responses.

In the final phase, the master table was split into separate tables, each corresponding to a specific combination of radiation types as studied across the selected publications. Rows were redistributed accordingly and reviewed manually. Within each table, rows were first sorted by radiation type combination and then grouped by biological system. To facilitate identification of studies involving specific models within a given combined radiation context, biological systems were broadly categorized in the following order: human-derived systems, rodent models, other animals and plants, and microorganisms (including bacteria, viruses, and unicellular eukaryotes). The tables were then regrouped into three overarching categories: “Radiobiological Studies” (with the sub-categories “Non-Ionizing–Ionizing”, “Non-Ionizing–Non-Ionizing”, and “Ionizing–Ionizing”), “Therapeutic Studies”, and “Space Radiation Studies”. This restructuring resulted in five tables, which were inserted into the manuscript. The finalized data set then underwent analysis, processing, and discussion.

No data conversions or statistical imputation were applied during synthesis. A statistical synthesis of results was not undertaken, reflecting substantial variability in study designs, exposure protocols, and outcome measures. Given that the majority of included studies were mechanistic, conventional risk-of-bias instruments developed for clinical or preclinical investigations were not directly applicable across the entire data set. Methodological rigor was therefore appraised qualitatively, with attention to factors most relevant to the diverse experimental contexts represented. Effect sizes were not standardized, as most studies reported results qualitatively or using heterogeneous outcome metrics that precluded meaningful numerical comparison. Results were instead synthesized based on reported biological outcomes and qualitative trends. Potential reporting biases were not formally evaluated, and certainty of evidence was not rated due to heterogeneity in radiation types, biological models, endpoints, and application contexts. All relevant data are reported within the manuscript; no additional data sets were generated or analyzed.

## 3. Results

A total of approximately 300 records were identified through database searches. After deduplication, screening, and eligibility assessment, 172 studies were included in the final review, as detailed in the PRISMA 2020 flow diagram (Figure 1). As outlined in the Methods, findings are presented narratively to reflect the diversity of study designs, biological systems, and outcome measures. Risk of bias and certainty of evidence were not formally assessed, in keeping with the qualitative synthesis approach.

In 18 cases, studies initially identified as eligible were replaced with equivalent reports more closely aligned with the predefined inclusion criteria. In accordance with PRISMA 2020 Item 16b, we include examples of studies that appeared to meet the criteria on initial review but were excluded after closer assessment. One such case is a report by Blanco et al., [1] “Effects of gamma and electron radiation on the structural integrity of organic molecules and macromolecular biomarkers measured by microarray immunoassays and their astrobiological implications”, which does describe combined irradiation effects on biological targets; however, these targets do not constitute true biological systems, and the publication was therefore excluded. Moreover, several therapeutic studies were excluded during eligibility reassessment due to technical mismatches with inclusion parameters. For example, a report by Aljabab et al., [2] “A combined neutron and proton regimen for advanced salivary tumors: early clinical experience”, was excluded due to extended intervals between therapeutic modalities that far exceeded the temporal threshold set in our criteria. Kumar et al., [3] in “Senolytic agent ABT-263 mitigates low- and high-LET radiation-induced gastrointestinal cancer development in *Apc*^1638N/+^ mice”, was similarly excluded upon reevaluation, as the study compared but did not combine radiation types and therefore did not meet the fundamental inclusion requirement.

The studies that were eventually included span a wide range of radiation exposures, including combinations of ionizing and non-ionizing radiation, as well as high- and low-LET (linear energy transfer) modalities. Given this diversity, a central aim of the review was to assess whether combined exposures yielded effects beyond those expected from individual radiation types. In this context, the term “synergy” is used to describe biological outcomes that exceed additive expectations based on single-exposure responses. Such outcomes include, for example, elevated levels of chromosomal aberrations, decreased cell survival, or gene expression changes that cannot be accounted for by a simple summation of effects from each radiation modality alone.

Quantitatively, synergy was inferred when the combined effect was statistically greater than the predicted additive response or when modeled interaction terms (e.g., interaction index > 1) indicated non-linear enhancement. Studies in which the effect of combined exposure depended on irradiation order or the involvement of specific DNA repair pathways were also considered synergistic, provided that the observed biological damage exceeded what would be expected from the individual exposures alone.

The presented studies are summarized in five structured tables, each organized by radiation type and dose, biological system, assessed endpoints, and observed outcomes. The included summary tables provide an overview of study design and scope, while specific findings are detailed in the corresponding text sections.

### 3.1. Radiobiological Studies

#### 3.1.1. Combinations of Non-Ionizing and Ionizing Radiation

Table 1 summarizes 17 experimental studies investigating combined exposures to non-ionizing and ionizing radiation, with a primary focus on ultraviolet (UV) radiation paired with X-rays, gamma rays, beta particles, or protons [4,5,6,7,8,9,10,11,12,13,14,15,16,17,18]. The research spans human, mammalian, and microbial cell models and examines endpoints such as chromosomal aberrations, clonogenic survival, DNA repair, and enzyme activity. Many of the included studies report synergistic effects. These outcomes often depended on factors such as cell cycle stage, radiation sequence, dose, and DNA repair competence. Selected examples are presented below to illustrate recurring patterns and mechanistic trends.

Among the included studies, one found no detectable synergy between UV and X-ray exposure in PHA-stimulated human lymphocytes in the G_1_ phase, as dicentric chromosome yields were equivalent to those induced by X-rays alone [4]. In contrast, another study using G_0_-phase human peripheral blood lymphocytes reported approximately a two-fold increase in dicentric chromosome formation when UV and X-rays were administered within 30 s of each other, regardless of UV dose or exposure order [5].

Furthermore, one study reported that when UV preceded X-rays, the increase in dicentric chromosome formation remained stable across inter-exposure intervals up to 90 min; in contrast, when the order was reversed, the effect declined with a half-life of approximately 20 min [7]. Similarly, in V79 cells, UV followed by X-rays resulted in greater loss of clonogenic survival and repair capacity compared to the reverse sequence [8].

Beyond mammalian systems, synergistic interactions were also reported in microbial models. For instance, in bacteria, UV pretreatment sensitized *Escherichia coli* to X-ray-induced lethality, particularly in wild-type and *uvr* mutants. However, no synergy was detected in *recA* and *recB* mutants, while *polA* mutants showed impaired repair recovery after irradiation [9,10]. In yeast models, UV exposure prior to gamma irradiation led to more than 100-fold reductions in survival beyond additive predictions and revealed strong synergy in *Schizosaccharomyces pombe* wild-type cells but not in recombination-deficient mutants [13]. UV pretreatment enhanced the lethality of incorporated ^32^P decay in *Salmonella*, with the synergistic effect almost completely eliminated by photoreactivation [16]. Moreover, UV followed by proton irradiation caused a stronger reduction in bacterial survival than the reverse sequence [18]. Applied microbial studies further demonstrated that combined UV and gamma irradiation increased antifungal activity and mutagenesis in *Bacillus velezensis* strains evaluated through bioindustrial screening, with increased surfactin production and broader isoform profiles observed in selected mutants [15].

Notably, in a full-thickness human skin model, pre-irradiation with water-filtered infrared-A radiation (wIRA, a subtype of near-infrared radiation) prior to X-ray exposure resulted in delayed repair kinetics, with prolonged persistence of γH2AX and 53BP1 foci. Reduced fibroblast apoptosis and preserved tissue morphology were also observed following combined exposure [19].

#### 3.1.2. Combinations of Non-Ionizing Radiation Types

Table 2 presents findings from 18 experimental studies investigating combined exposures involving non-ionizing radiation types, primarily various ultraviolet wavelengths in combination with visible light, near-infrared, or other UV subtypes. The studies span human, rodent, invertebrate, plant, and microbial biological models and assess endpoints such as microbial inactivation, gene expression, oxidative stress, DNA damage, and tumor incidence. While experimental systems and dose ranges vary, many studies report synergistic effects.

In *E. coli*, simultaneous UV-A and UV-B exposure produced approximately 100-fold greater inactivation than UV-B alone, mediated by s^4^U modifications in tRNA [26]. In parallel, human skin in vivo showed enhanced effects on photoaging biomarkers following natural sunlight exposure (full spectrum, including UV-B, UV-A, visible, and near-infrared), exceeding those observed under UV-filtered (visible/infrared-only) conditions [22]. Furthermore, combined UV and visible light exposure in ragweed-allergic patients significantly inhibited allergen-induced wheal formation, whereas neither UV-A nor visible light alone produced this effect. The mixed UV-visible exposure (mUV/VIS) suppressed mast-cell-mediated responses in a dose-dependent manner, including at suberythematous doses [21].

Additional synergistic effects were observed in plant, invertebrate, and microbial systems. In tomato plants, UV-B and UV-C exposure increased disease resistance and antioxidant gene expression [25]. In marine invertebrates, long-term UV-A and UV-B irradiation reduced reproductive output and feeding capacity [24]. In environmental disinfection studies, complete microbial inactivation of *Enterococcus faecalis* and *E. coli* was achieved in synthetic water matrices using dual-UV systems, with synergy modulated by fluence and pH in certain setups [31]. Sequential UV-C exposure with KrCl excimer (222 nm) and low-pressure mercury lamps (254 nm) synergistically enhanced viral inactivation, with energy requirements influenced by total dose and exposure order [37].

#### 3.1.3. Combinations of Ionizing Radiation Types

Table 3 provides an overview of 52 radiobiological studies investigating combinations of ionizing radiation types, primarily involving alpha particles, X-rays, gamma rays, protons, and neutrons. Most studies employed in vitro models, with additional in vivo and environmental systems, and explored endpoints such as clonogenic survival, DNA damage signaling, chromosomal aberrations, gene expression, inflammatory markers, and mutation rates. The biological responses varied widely depending on radiation quality, dose, sequence, and cell type. While synergistic effects were frequently reported—particularly in mixed alpha–X-ray and neutron–gamma exposures—several studies observed purely additive or even antagonistic outcomes. Select examples are outlined below to highlight key patterns in damage complexity, repair interference, and sequence dependence.

In human peripheral blood lymphocytes, combined alpha and X-ray exposure significantly increased complex chromosomal aberrations in a dose-dependent manner, with non-additive effects and a linear-quadratic response observed at higher mixed doses [43]. In U2OS osteosarcoma cells, co-exposure delayed the decay of 53BP1 foci and prolonged ATM and p53 signaling, with the strongest effects at lower total doses [45]. Similar findings were reported for sequential alpha and gamma radiation, where alpha-first exposure produced larger and more persistent foci than the reverse sequence [55]. A related study in human–hamster hybrid AL cells showed that low-dose X-ray priming suppressed bystander mutagenesis at the CD59 locus, while alpha exposure followed by an X-ray challenge produced supra-additive increases in mutant yield with a more complex mutation spectrum [54].

Mixed-exposure effects on clonogenic survival varied substantially by system and irradiation parameters. In V79 Chinese hamster lung fibroblasts, neutron–gamma co-exposure produced supra-additive reductions in survival, particularly under simultaneous delivery. Survival curves fitted better with quadratic models, indicating interaction between damage types [78]. In rat lung epithelial cells (LECs), combined alpha (1 Gy) and X-ray exposure eliminated the survival curve shoulder and reduced clonogenic survival beyond additive expectations. Micronuclei frequencies were also elevated under combined exposure, consistent with non-linear interaction effects [53]. In contrast, a study in Chinese hamster ovary (AA8) cells using mixed alpha and X-ray beams reported no evidence of synergy, with survival data aligning closely with predictions from mathematical additivity models [51].

At the transcriptional level, combined radiation exposures were often found to modulate key stress and damage response genes across multiple systems. In human lymphocytes, combined alpha–X-ray exposure increased FDXR, GADD45A, and MDM2 expression beyond levels induced by alpha radiation alone in most donors, with synergy confirmed in three out of four cases using envelope-of-additivity analysis. Furthermore, ATM inhibition reduced this response, implicating checkpoint signaling [42]. Neutron–photon mixtures elicited strong transcriptomic effects even at low neutron fractions. In murine models, as little as 5% neutron contribution suppressed EIF2/mTOR signaling and ribosomal protein expression—alterations not observed with X-rays alone [62]. In human peripheral blood, increasing neutron percentages led to enhanced TP53 signaling and broader immune dysregulation [61]. Another study reported dose-dependent modulation of *BAX*, *DDB2*, and *FDXR* expression following neutron irradiation, with combined neutron–gamma exposure activating overlapping gene sets [72].

Beyond transcriptional responses, several studies show that mixed-field irradiation can also trigger broader cellular and physiological adaptations. In murine blood cells, neutron–gamma exposure altered membrane architecture and lectin-binding patterns, with lymphocytes and platelets showing the most marked ultrastructural remodeling [80]. In separate mouse models, combined exposures impaired hippocampus-dependent memory and shifted neuroimmune profiles toward an anti-inflammatory state [81].

A rare entry in the data set examined neutron–proton co-exposure in human breast cancer cell lines, offering insight into mixed-beam effects on cancer stem cell (CSC) populations. The response varied by cell line: CSC fractions declined additively in MCF-7 cells but antagonistically in MDA-MB-231, with no significant changes in canonical stemness gene expression [86]. This study was among the few to examine neutron–proton co-exposure, providing rare data on beam combinations with potential relevance to radiation protection and therapy

Additional underrepresented combinations include alpha–beta inhalation in rats, which were found to elicit additive impairments in pulmonary function [60], and mixed radionuclide exposure in plants, where barley grown in contaminated soil accumulated mutations at rates exceeding those predicted by dose alone [58]. Though differing in species and context, these models replicate real-world exposure scenarios such as internal contamination and chronic environmental irradiation, with effects surpassing dose-based predictions.

### 3.2. Therapeutic Studies

Table 4 outlines 37 clinical, preclinical, and in vitro studies evaluating therapeutic applications of mixed-radiation modalities. The reported outcomes reflect the influence of beam combinations, sequencing, timing, and model system, with examples spanning conventional photons, protons, neutrons, heavy ions, and photodynamic approaches.

Clinical investigations of mixed proton–photon regimens have demonstrated favorable outcomes across several tumor types. In supratentorial glioblastoma, a combined modality approach using daily photon irradiation with a proton concomitant boost delivered more than 6 h later achieved a median survival of 21.6 months and manageable toxicity, albeit with occasional late leukoencephalopathy [91]. Similarly, in stage II–IV oropharyngeal squamous cell carcinoma, a regimen incorporating photons and a proton boost produced a 5-year locoregional control rate of 84%, although acute mucosal toxicity and grade 3 late effects were observed in some patients [92]. For skull base and cervical spine chordomas and chondrosarcomas, sequential photon–proton treatments resulted in excellent long-term control for chondrosarcomas (94% at 10 years) and moderate control for chordomas, with late complications including temporal lobe injury and endocrinopathy [93].

In Chinese hamster fibrosarcoma cells, sequential carbon-to-proton irradiation with a 45% carbon contribution induced significant synergy, while reversing the order diminished the effect or even led to antagonism [94]. In a related study using combined proton and heavy recoil (HR) irradiations, greater high-LET contributions suppressed survival more effectively, particularly when the heavy recoils were delivered before protons. Notably, a proton-to-HR sequence permitted partial recovery, indicating an order-dependent effect [97].

Several studies evaluated neutron-based combinations. In a murine study, neutron–proton sequencing influenced both toxicity and tumor outcomes in solid Ehrlich ascites carcinoma. Neutron-first exposures exacerbated skin damage and reduced survival, while the reverse sequence was better tolerated [98]. These findings were echoed in vitro, where neutron-before-proton combinations consistently produced greater reductions in clonogenic survival than the opposite order [99]. In clinical settings, mixed neutron–photon protocols have been investigated in patients with malignant gliomas, where neutron boosts were delivered within minutes or hours of photon irradiation. One such study administered a neutron boost 5–20 min before whole-brain photon delivery and reported markedly prolonged survival in anaplastic astrocytoma but not glioblastoma [118]. In a separate randomized trial, twice-weekly neutron boosts were given within 3 h of photon therapy, achieving frequent tumor sterilization at autopsy but also substantial radiation injury [119].

Other investigations have combined external beam therapy with biologically targeted or nanotechnology-enhanced modalities. In mouse xenograft models, combining proton irradiation with ^177^Lu-labeled targeted radionuclides produced additive or synergistic tumor suppression depending on the tumor model [95]. Similarly, preclinical work combining low-energy X-rays with ^177^Lu-PSMA-617 radionuclide therapy in prostate cancer xenografts prolonged tumor doubling time and increased median survival compared to external beam radiation alone [96]. Photodynamic therapy (PDT) integrated with photon radiotherapy (RT) has demonstrated synergistic antitumor effects in various contexts. For example, studies in bladder cancer organoids showed enhanced cell death and immune migration [102], while pancreatic cancer spheroids and co-cultures exhibited increased apoptosis, DNA damage, and reduced viability [104]. In breast cancer cells, PDT with indocyanine green (ICG) and low-dose X-rays substantially increased cell killing compared to monotherapies [106]. Additionally, X-ray-triggered UV-C emission from nanoscintillators enhanced cytotoxicity and G2/M arrest in lung cancer spheroids [107] and potentiated CPD DNA damage and clonogenic suppression in UV-sensitive fibroblasts [108]. In early-phase clinical studies of cervical cancer, initiating HDR brachytherapy during pelvic external beam radiotherapy resulted in approximately 70–75% tumor volume reduction by the first brachytherapy fraction—an instance of integrating biologically targeted and external beam modalities [110].

One of the included studies investigated a combined photon–electron beam technique for breast irradiation. Seven patients with early-stage breast cancer underwent accelerated partial breast irradiation following lumpectomy, receiving a total dose of 38.5 Gy in 10 fractions over five days. Treatment consisted of both modulated electron beams (9–15 MeV) and intensity-modulated photon fields (6 MV) delivered within each fraction. The reported results showed adequate target coverage, lower radiation exposure to the ipsilateral lung and heart compared to IMRT alone, no grade 2 or higher acute toxicity, and favorable cosmetic outcomes in the majority of cases [111].

Several studies have investigated radiation combinations falling within the broader framework of boron neutron capture therapy (BNCT). In vitro mixed-field exposures have demonstrated additive effects on survival endpoints, for example in alpha–gamma co-irradiation models of V79-4 Chinese hamster cells [123]. In contrast, neutron–gamma exposures have been associated with persistent and spatially clustered DNA damage, with larger and more complex foci compared to gamma irradiation alone, consistent with increased damage complexity and altered repair dynamics [122]. Notably, one study reported a strong synergistic effect on cell killing when alpha particles and X-rays were delivered simultaneously at higher alpha doses, suggesting that alpha exposure can impair DNA-repair responses induced by low-LET X-rays [124]. Furthermore, BNCT-associated mixed-beam exposures caused more pronounced effects in wild-type than in repair-deficient cell lines [125]. A related investigation demonstrated that thermal neutrons generated during proton beam therapy could also be harnessed to trigger boron neutron capture reactions, significantly enhancing cell killing in vitro [121]. More recently, a combined boron and gadolinium neutron capture therapy approach was tested, delivering both high-LET alpha particles and gamma rays in a single neutron exposure. In preclinical head-and-neck tumors, this approach achieved nearly complete tumor eradication and strong suppression of recurrence biomarkers [126].

### 3.3. Space Radiation Studies

Table 5 presents 48 studies reporting biological effects of combined radiation exposures relevant to spaceflight. These are mostly ground-based mixed-field exposures designed to mimic space radiation—primarily galactic cosmic radiation (GCR). The studies include in vitro models, animal experiments, and one real spaceflight case, the NASA Twins Study. Endpoints include DNA damage, chromosomal aberrations, cellular transformation, tissue-specific effects, immune and cardiovascular outcomes, neurocognitive and behavioral assessments. Observed effects varied with exposure parameters—such as sequence, timing, and composition—as well as biological factors including sex, tissue type, and model system.

Notably, many of these studies employed the full NASA GCR simulation (GCRsim) model, which sequentially delivers 33 ion beams. In this model, each beam represents one of several energies sampled from fewer than 33 distinct ion species. Simplified versions with fewer beams were also used, often referred to as “simplified GCRsim” or “Smf-GCR”. Terminology in the literature is inconsistent—for example, 6-beam simplified GCRsims often included 5 ion species, with one delivered at two energies, yet were reported as “5-ion GCRsims”. Therefore, to avoid confusion, results are reported with the actual number of beams or ion types used in each case.

Many studies using simplified GCRsim beams reported dose- and sex-specific deficits in short-term memory, spatial learning, and cognitive flexibility at total doses of ~0.5–1 Gy. In a two-ion study, these effects were linked to dendritic simplification, reduced mushroom spine density, and altered expression of GluR1, NR2A, and Synapsin-1 [132]. A related study reported similar structural changes and SNP alterations, though synaptic proteins were not assessed [133]. In another, five-ion study, male-specific cognitive deficits were causally linked to microglial activation and hippocampal remodeling, with early blood monocytes indicating late-stage impairment [142].

In many cases, low-dose GCRsim exposure was found to alter complex behaviors in a sex-dependent manner. For instance, in mice, females showed increased grooming and burrowing activity at 15–50 cGy, with nest-building performance varying by dose and sex, despite intact sensorimotor function [145]. In a similar experiment, female rats displayed increased risk-taking after 10 cGy, while males showed slower decision speeds, both normalizing by 90 days [149]. In a touchscreen-based task, 10 cGy impaired switch accuracy in females (~20%), increased perseverative errors, and reduced training success under cognitive load [151].

Chronic and acute exposures impaired memory updating and object recognition in a sex-dependent manner. Long-term potentiation (LTP) was reduced, and changes in postsynaptic density and axonal myelination were observed in mice exposed to a 33-beam GCRsim [162]. Complementary touchscreen-based testing showed that male mice developed attentional deficits and slower response times, alongside disrupted prefrontal dopamine signaling and altered neurotransmitter network organization [163]. One study using a six-beam simplified GCRsim found that pharmacological HDAC3 inhibition reversed LTP deficits and restored behavioral performance, indicating that epigenetic modulation can mitigate GCR-induced cognitive impairments [160].

Parallel investigations assessed cancer risk and tissue remodeling under space-relevant mixed radiation. In *K-ras*^LA1^ mice, chronic 33-beam GCRsim exposure increased lung adenocarcinoma incidence, an effect further enhanced by a delayed neutron dose [165]. In C57Bl/6J mice, exposure to a simplified five-ion GCR simulation impaired novel object recognition and altered brain connectivity, effects that were partially mitigated by prophylactic amifostine treatment [156]. Mammary gland studies in *Apc*^Min/+^ mice showed increased ductal hyperplasia and overgrowth, along with elevated expression of ERα, ERRα, and SPP1—markers associated with estrogen signaling and tumor development [161,168].

In primary human fibroblasts, sequential exposure to protons followed by titanium or iron ions with a 15 min interval produced a marked synergistic increase in anchorage-independent growth, even at low proton doses. This effect was not observed with reversed ion order or same-ion split doses [130]. This pattern was reinforced in neonatal fibroblasts, where transformation rates were up to three times higher than additive predictions when protons were followed by HZE ions within a 2.5 min to 6 h window, but not when the order was reversed [131]. In human mammary epithelial cells, chromosome aberration frequency was highest when iron ions were delivered 30 min after protons. Combined exposure produced more damage than the predicted additive response, with both intra- and inter-chromosomal exchanges elevated at this interval [128].

Radiation-induced cardiovascular and systemic effects were documented across multiple models. In WAG/RijCmcr rats, whole-body exposure to a mixed field of protons, silicon, and iron ions led to perivascular cardiac fibrosis, increased systolic blood pressure, and infiltration of CD68+ macrophages in the heart and kidneys—effects not observed with single-ion exposures [137]. In BALB/c and CD1 mice, five-ion simplified GCRsim caused sex- and strain-specific changes in cardiac structure, including increased collagen deposition, altered capillary density, and modulation of immune markers such as CD2 and TLR4 [144]. To mimic the persistent radiation-induced biological effects observed in space exposure scenarios, another study using a myocardial infarction model administered proton and iron irradiation over a 48 h interval—exceeding the ≤24 h threshold used in almost every study reviewed here—which significantly influenced myocardial fibrosis and infarct size [134]. Peripheral immune alterations included changes in leukocyte profiles and phagocytic activity, with notable sex-specific transcriptional shifts [140].

Studies also implicated the gut microbiome as a sensitive and interactive component. In several mouse models, microbiota composition changed after GCRsim in a sex- and dose-dependent manner and correlated with cognitive, affective, and neuropathological outcomes [135,148,159].

Other systemic interactions included viral reactivation, endocrine modulation, and altered intercellular communication. In latently infected human myeloblasts, GCRsim and high-LET ions reactivated CMV and induced viral gene expression in a dose- and LET-dependent fashion [157]. In co-culture systems, proton pre-exposure suppressed the bystander response to iron ions in human fibroblasts, underscoring the impact of prior radiation history on intercellular signaling dynamics [129].

In the NASA Twins Study, 340 days of spaceflight—including exposure to approximately 76 mGy of galactic cosmic radiation—resulted in persistent chromosomal inversions, telomere elongation followed by rapid shortening, immune dysregulation, cognitive decline, and altered microbiome composition. Multi-omic analyses showed that many of these effects persisted post-flight and mirrored findings from rodent GCR models [175].

Several studies have tested countermeasures against GCR-induced effects. Low-dose X-ray pretreatment improved viability and reduced ROS in cardiomyoblasts exposed to GCRsim [154]. The antioxidant CDDO-EA mitigated cognitive deficits and restored neurogenesis in female mice [166], though its effects on social behavior were variable [167]. Furthermore, HDAC3 inhibition was shown to reverse synaptic plasticity impairments [160]. In contrast, another investigation found that aspirin failed to protect and worsened cognitive performance in control animals [171].

## 4. Discussion

The systematic collection, classification, and synthesis of studies on combined irradiation of biological systems conducted in this review provided critical insights into the current state of the field. The research landscape is consistently structured around three main contexts: radiobiology, therapeutic applications, and space radiation research. This tripartite framework reflects fundamentally distinct objectives—mechanistic elucidation, clinical application, and astronaut health risk mitigation, respectively.

### 4.1. Mechanistic Basis of Mixed-Radiation Effects

Radiobiological studies have consistently revealed that combinations of ionizing and non-ionizing radiation produce distinctive biological effects compared to single-modality exposures. These effects are shaped by variables such as radiation type, dose, timing, and cellular context. As summarized in Figure 2, such interactions often lead to complex DNA damage, saturation of repair capacity, and the emergence of genomic instability.

#### 4.1.1. Exposure Sequence Dependence in Mixed-Radiation Exposures

Evidence from a wide range of biological systems—including bacteria, primary human cells, established mammalian lines, and whole organisms—demonstrates that the impact of sequential mixed-radiation exposures extends beyond simple dose accumulation, showing that timing and cellular context can be critical determinants of outcome. In combinations of non-ionizing and ionizing radiation, many studies of UV–ionizing pairings either did not systematically assess the role of exposure sequence or reported it as having limited impact, yet several investigations have clearly shown that the temporal order of exposures can determine whether damage responses remain additive or become strongly synergistic. For example, in human lymphocytes, UV irradiation preceding X-rays resulted in a stable two-fold increase in chromosome aberrations, whereas reversing the order caused a rapid decline in synergy [7]. Similarly, UV pre-exposure sensitized *E. coli* to proton irradiation, while the reverse sequence did not produce synergy [18], and in *Micrococcus radiophilus*, UV pretreatment yielded pronounced synergistic killing, contrasting with additive effects when gamma rays were applied first [14].

Expanding on this pattern, similar effects have been observed across varied biological models and irradiation conditions. Sequential exposure of MS2 bacteriophage to different UV wavelengths led to significantly greater inactivation when low-wavelength UV preceded higher-wavelength UV [37]. In U2OS cells, alpha radiation delivered before gamma irradiation induced more persistent DNA repair foci and chromatin remodeling [55]. In Chinese hamster fibrosarcoma cells, carbon ion priming heightened the cytotoxic impact of subsequent proton irradiation [94], while neutron–proton combinations proved most effective when neutrons were applied first [99]. More recent studies modeling space radiation environments also illustrate this phenomenon: in murine lung tissue, delivering protons prior to silicon ions markedly increased premalignant lesions [136], and in primary human fibroblasts, proton priming before HZE ion exposure tripled transformation markers compared to the reverse sequence [131].

This set of examples helps demonstrate that sequence-dependent interactions are neither confined to any one radiation pairing nor restricted to a single biological context. Instead, they represent a broadly observed effect that likely arises from the interplay between DNA damage complexity, cellular stress responses, and the inherent repair competence of each system.

#### 4.1.2. DNA Repair Dependencies, Lesion Interactions, and Temporal Modulation

Microbial systems have been especially valuable for clarifying the molecular basis of DNA damage–repair interactions, as they allow for precise dissection of DNA repair pathways through genetic manipulation. In *E. coli*, synergistic lethality from combined UV and X-ray exposures was observed in wild-type and *polA* mutants but abolished in *recA*, *recB*, and *recC* mutants, implicating homologous recombination (Type III repair) as a major contributor to synergy. The partial effect seen in *polA* mutants suggests that base excision or type II repair may modulate the response as well, though neither is sufficient on its own [9,10]. Further work using *B/r* and *Bs-1* strains confirmed that UV pretreatment interferes with the repair of strand breaks: synergistic X-ray sensitization occurred only in wild-type *B/r* cells and was abolished when DNA metabolism was perturbed by 5-bromouracil substitution or purine starvation—conditions known to inhibit DNA repair [11]. In human leukocytes, combined UV and X-ray exposure led to greater chromosome damage in both healthy and Down syndrome donors, although the increment was less pronounced in the latter, with diminished DNA repair synthesis also observed in these cells [6]. Overlapping repair pathways may therefore mediate synergistic outcomes, with these effects being strongly influenced by cellular repair competence.

Critically, the temporal arrangement of UV and ionizing radiation exposures emerged as a key determinant of synergistic responses in mammalian cells. In peripheral human lymphocytes, a near-two-fold increase in dicentric chromosome yield was observed when UV irradiation preceded X-rays—even when inter-exposure intervals extended up to 90 min [7]. This stability indicates that UV-induced lesions can persist long enough to interfere with the repair of subsequent ionizing damage, thereby elevating the probability of chromosome aberration formation. In contrast, reversing the order of exposure led to a rapid decay in the synergistic effect, with a measured half-life of approximately 20 min. This decay strongly implicates short-lived intermediates—such as unrepaired single-strand breaks or transient chromatin alterations—in mediating the synergy when X-rays are delivered first. The interaction between lesions was therefore found to be highly time-sensitive, with DNA repair kinetics playing a central role in shaping interactive outcomes. Consistent with this observation, studies in murine intestinal crypts showed that mixed neutron and X-ray irradiation produced non-additive effects when delivered within a few hours of each other but fully additive outcomes when sufficient time was allowed for sublethal damage repair—supporting the importance of lesion recovery windows in determining synergy or independence [69].

A key takeaway is that temporal parameters—such as lesion persistence, inter-exposure intervals, and the kinetics of repair pathway engagement—can decisively influence whether interactions are synergistic, additive, or even antagonistic. While short inter-exposure intervals may allow persistent lesions or intermediates from the first radiation to interfere with the cellular response to the second, longer delays may permit resolution, thereby diminishing interactive effects. For instance, pretreatment with water-filtered near-infrared radiation before X-ray exposure was reported to delay DNA damage repair while reducing apoptosis in full-thickness skin models, suggesting that certain non-ionizing modalities, despite complicating repair, might even help mitigate the harmful effects of ionizing radiation through the modulation of stress signaling pathways [19].

#### 4.1.3. Cell-Cycle-Specific Interaction Effects

Moreover, the temporal dependence discussed above aligns with cell-cycle-specific differences in DNA repair dynamics. Studies in human lymphocytes have helped elucidate how cell cycle stage and lesion persistence may shape interactive outcomes. Experiments evaluating UV and X-ray co-exposures in quiescent (G_0_-phase) and stimulated (G_1_-phase) lymphocytes revealed that the presence or absence of active repair processes fundamentally alters the response to mixed radiation. In G_0_-phase cells, UV-C irradiation administered immediately before X-rays consistently produced a two-fold increase in dicentric chromosome formation relative to X-rays alone [5,7]. In contrast, G_1_-phase lymphocytes showed no evidence of synergy, with combined exposures yielding only chromatid-type aberrations typical of UV damage and dicentric frequencies indistinguishable from X-rays alone, suggesting that progression into G_1_ allows for repairing or processing lesions before they interact [4]. This difference indicates that the cell’s position in the cycle determines whether damage responses remain compartmentalized or converge to produce more complex chromosomal changes.

Cell-cycle-specific effects have been widely observed in mixed-radiation studies. In synchronized V79 cells irradiated in late S-phase, sequential exposure to high-LET particles followed by X-rays produced marked synergism, even though this phase was relatively resistant to either radiation alone, indicating phase-dependent susceptibility to combined damage [89]. Within a similar framework, analysis of sequential exposures across defined stages showed the greatest interaction in late S-phase, with smaller effects at the G_1_/S border and in G_2_, consistent with variations in repair capacity during the cycle [88]. In plateau-phase V79 cells—comprising a predominantly G_1_ population—mixed neutron–gamma exposures yielded survival curve modifications closely resembling those produced by post-irradiation treatment with β-araA, a DNA polymerase inhibitor that fixes potentially lethal damage by inhibiting post-replication repair; results indicated that the interaction effect in this context reflects repair constraints characteristic of G_1_ [83]. In a different context, combined UV-emitting nanoparticles and ionizing radiation in 3D lung cancer spheroids enhanced cytotoxicity through apoptosis, necrosis, and persistent G_2_/M arrest, illustrating that mixed exposures can exert phase-specific effects by overwhelming DNA repair pathways active during late cell cycle stages [107].

#### 4.1.4. Mechanistic Insights from Non-Ionizing Radiation Combinations

In non-ionizing radiation combinations, synergistic effects have been observed across various biological systems, often involving complementary mechanisms of photobiological damage. In *E. coli*, simultaneous UV-A and UV-B exposure resulted in a nearly 100-fold increase in inactivation compared to UV-B alone, driven by UV-A absorption through thiouridine (s^4^U) residues in tRNA, which impaired translation and thereby reduced the cell’s capacity to repair subsequent DNA damage [26]. In human skin in vivo, exposure to natural sunlight produced a distinct response characterized by elevated MMP-1 expression, reduced type I procollagen, and inflammatory cell infiltration—effects amplified under full-spectrum exposure compared to filtered conditions, consistent with contributions from multiple spectral components [22]. This pattern implies that when photoreactive wavelengths converge, they can engage parallel cellular sensors and signaling cascades in ways that transcend mere summation, effectively reprogramming both acute stress responses and longer-term adaptations. In microbial systems, sequential UV-A and UV-C exposure produced translational arrest that impaired recovery and synergistic inactivation in wild-type *E. coli*, an effect absent in strains lacking *thiI*-dependent tRNA modifications, highlighting the role of non-DNA targets in modulating radiation sensitivity [27].

In a human skin model of allergen challenge, mixed ultraviolet and visible light (mUV/VIS) exposure produced dose-dependent suppression of mast-cell-mediated wheal formation at suberythematous doses—an effect not seen with UV-A or visible light alone. This inhibitory response was attributed to the combined action of different wavelengths on histamine release from cutaneous mast cells, suggesting that such protocols may achieve immunomodulation with lower UVB doses, thereby reducing phototoxicity risks while retaining efficacy with possible translational therapeutic applications [21].

A key deduction from these findings is that synergy in non-ionizing radiation exposures often reflects the combined engagement of diverse cellular targets and signaling processes, which together generate biological effects that differ qualitatively from those induced by single-wavelength exposures.

#### 4.1.5. Mechanisms and Functional Consequences of Combined Ionizing Radiation Exposure

Mechanistic insights from combinations of ionizing radiation types reveal that synergy frequently stems from the intersection of distinct lesion types and the breakdown of repair coordination. When alpha particles are combined with X-ray, the resulting DNA damage is not merely greater in magnitude but also altered in quality—evidenced by slower resolution of DNA repair foci, complex chromosomal aberrations, and sustained ATM and p53 signaling [43,45]. Moreover, the fact that such supra-additive effects were observed even at modest combined doses suggests that repair mechanisms may be overwhelmed even when simple additivity would otherwise be anticipated. In this line of thought, a study in AL cells showed that combined alpha particle irradiation and X-ray challenge generated supra-additive effects and increased the incidence of complex, discontinuous CD59^−^ mutations [54], again indicative of qualitatively more complex lesion outcomes, which are associated with persistent genomic instability. 

Transcriptional reprogramming offers insight into how cells perceive and prioritize competing damage signals. Co-exposures such as alpha–X-ray and neutron–photon combinations induced distinct expression signatures, including upregulation of DNA damage response genes (e.g., FDXR, GADD45A, MDM2) and suppression of translation-related pathways such as EIF2/mTOR signaling [42,62].Combined neutron and gamma irradiation engaged a broad spectrum of transcriptional alterations, reflecting nearly the full complement of genes and pathways activated by each radiation alone, and exceeding those induced by gamma rays [72]. This overlapping transcriptional response points to the integration of high- and low-LET damage, characteristic of mixed-field responses. Neutron–photon mixtures also triggered enhanced TP53 activity and immune dysregulation as neutron fractions increased [61], suggesting multilayered stress integration beyond additive effects. Importantly, these effects occur even in the absence of detectable cytotoxicity, indicating that transcriptional signatures capturing DNA repair engagement, metabolic disruption, and immune modulation may serve as early predictors of synergistic stress—crucial for biodosimetry and mechanistic understanding of mixed-field exposures.

Clonogenic survival data link molecular changes to functional outcomes. Supra-additive losses in colony formation under combined neutron–gamma or alpha–X-ray exposure reflect cumulative disruptions to proliferation and genome stability [39,53,56,75,78,87]. Sequential exposure studies have further shown that pre-irradiation with alpha particles can markedly reduce the shoulder of subsequent X-ray survival curves without appreciably changing slope within the studied dose range, indicating persistent sublethal damage capable of interacting with later insults [52]. Moreover, experiments in yeast have demonstrated clear synergistic effects when alpha and gamma radiation are applied simultaneously, with mutation frequencies exceeding additive expectations by over 30% [57]. An important clarification to be made is that not all systems or experimental conditions yield detectable synergy; in AA8 cells, survival curves under simultaneous mixed-beam exposure were consistent with additive predictions, suggesting that under these specific parameters, combined irradiation did not enhance clonogenic loss [51]. This variability reinforces the role of cell-specific repair competence and exposure sequence in shaping whether synergy manifests—an essential consideration in therapeutic planning and radioprotective modeling.

#### 4.1.6. Multiscale Biological Effects and Uncommon Exposure Scenarios

Beyond nuclear targets, mixed-field exposures have been found to affect broader cellular structures and physiological systems, including membrane organization in blood cells [80], neuroimmune processes in the central nervous system [81], systemic metabolic responses reflected in serum fluorescence changes [79], and pro-inflammatory lipidomic alterations indicating dysregulated lipid metabolism [85]. Such effects demonstrate that combined radiation can disrupt multiple levels of biological organization, which has important implications for risk assessment in complex exposure scenarios, including spaceflight and radiological emergencies.

Interestingly, even rarely tested combinations such as neutron–proton co-exposure have revealed subtype-specific effects on cancer stem cell populations, indicating that such combinations may produce differential responses not observed with individual modalities [86]. Likewise, environmental models involving complex or atypical exposure scenarios—such as mixed radionuclide contamination in plants or alpha–beta inhalation in rodents—demonstrate that chronic, low-dose combined exposures can lead to notable effects such as mutation accumulation and functional tissue impairment over time [58,60]. It thus becomes evident that radiation synergy is not merely a byproduct of experimental design but a genuine biological phenomenon that occurs in real-world contexts. As such, mixed-field effects must be integrated into predictive models of environmental risk, long-term health outcomes, and radiological protection—especially in scenarios dominated by chronic, low-dose exposures.

Considered in their entirety, the radiobiological findings discussed throughout this section indicate that, under defined conditions, combined radiation exposures can lead to amplified or qualitatively distinct biological outcomes, driven by interactions at the levels of DNA damage formation, repair kinetics, and stress response signaling. A deeper understanding of these dynamics is essential for refining risk assessment models and for effectively harnessing combinatorial effects in both therapeutic interventions and protective strategies in space exploration.

### 4.2. Determinants and Mechanisms of Synergy

Synergy—defined in this review as a biological effect exceeding the sum of individual radiation exposures and referred to elsewhere in this review as “supra-additive” or “non-additive” outcomes—typically emerges not from simple dose accumulation but from the interaction of distinct stressors in ways that challenge or exceed a cell’s adaptive capacity. Across the combinations examined in Section 4.1, synergistic outcomes are linked to circumstances in which early damage persists or leaves molecular “footprints” (e.g., persistent γH2AX/53BP1 foci, chromatin compaction changes, or mitochondrial ROS carryover) that alter the processing of subsequent insults. These conditions are often shaped by exposure sequence, radiation quality, dose rate, oxygenation status, and the repair competence and cell-cycle position of the responding system, with cell type–specific repair capacity also strongly influencing whether synergy manifests.

While in many cases, mixed-field outcomes are found to be additive or even antagonistic—sometimes due to adaptive responses, checkpoint-mediated slowing, or induced buffering—synergy appears to be more likely when interacting lesions or stress signals converge on repair- or signaling-limited states, such as saturation of homologous recombination capacity or checkpoint arrest without full repair. At the molecular level, co-exposures have been shown to shift transcriptional and post-translational programs and repair pathway priorities in ways that can be qualitatively distinct from single-modality responses—reconfiguring metabolic balance and immune signaling. Such emergent patterns are often not simple amplifications of known responses but reflect the integration of disparate inputs across damage sensing, checkpoint coordination, and recovery pathways.

Recognizing synergy as a product of interaction rather than accumulation is essential for predictive modeling, since its occurrence depends on a defined set of parameters—dose, radiation quality, timing, and biological context—that determine whether damage responses remain independent or are re-shaped into outcomes exceeding additive expectations.

### 4.3. Therapeutic and Protective Strategies

Clinical studies indicate that combining different radiation types often enhances therapeutic effects beyond what would be expected from single modalities, albeit not always without collateral adversities. For example, proton–photon regimens have yielded promising control rates in glioblastoma, oropharyngeal, and skull base tumors but also revealed significant late complications such as leukoencephalopathy, mucosal toxicity, and temporal lobe injury [91,92,93]. Other clinical studies combining neutrons and photons revealed improved tumor control and survival in anaplastic astrocytoma, albeit with frequent radiation injury [118,119], again reflecting the observed balance between therapeutic benefit and treatment-related risk. Harnessing the advantages of combined radiations for therapeutic aims therefore requires continual refinement of planning, both dosimetric and methodological, as well as reinforcement through preclinical and radiobiological experimentation. In this context, it is worth noting that the reported favorable outcomes of partial breast irradiation using photon–electron beams were made possible by prior high-precision, Monte Carlo–based planning and overall careful treatment design [111]. Ultimately, the therapeutic value of mixed-beam strategies lies in their capacity to exploit complementary physical and biological effects, but their clinical success depends on anticipating adverse effects and integrating radiobiological insight into treatment protocol.

Beyond external beams, hybrid approaches that merge radiation with nanotechnology or molecular targeting are being explored. PDT combined with photon RT was shown to enhance tumor killing in multiple models [102,103,104,105,106]. Radiosensitization via targeted radiopharmaceuticals or nanoscintillator-triggered UV damage suggests that combinatorial designs can exploit biological vulnerabilities in novel ways to boost therapeutic efficacy by enhancing localized radiation delivery and increasing tumor-specific effects [95,96,107,108]. Further combinations involving targeted modalities, such as HDR brachytherapy in conjunction with external beam radiation therapy (EBRT), have shown promise in certain contexts, providing more precise tumor targeting, enhanced tumor shrinkage, and reduced normal tissue exposure [110].

In medical, environmental, and spaceflight settings involving combined radiation exposures, protective strategies are essential to reduce associated risks. These methods have been assessed in radiation pairings across the spectrum, aiming at attenuating damage through supplementary protective agents or at promoting adaptive responses by use of a preceding irradiation. The latter methodological approach is of specific interest in the scope of this review. For instance, in medical applications, techniques such as photobiomodulation, which uses pre-irradiation with therapeutic low-intensity light, exemplify how such interventions can modulate adaptive cellular responses and influence defense systems, potentially activating protective signaling pathways and supporting tissue preservation [19]. Similarly, pre-exposure to low-intensity laser irradiation in rats has shown radioprotective effects by enhancing antioxidant defense and supporting hematological recovery [20]. Furthermore, one example of low-dose X-ray pre-exposure in the context of space radiation research has illustrated how adaptive responses can be triggered to improve radiation tolerance in cellular models, providing further insight into radiation’s potential adaptive effects [154]. Overall, the takeaway is that carefully tailored radiation strategies, informed by the timing, dose, and type of exposure, can create new paths for both amplifying therapeutic efficacy and enhancing environmental and space radiation protection, provided these factors are precisely controlled. Ongoing research is essential to delineate the mechanisms and optimize protocols for safe and effective implementation.

Altogether, mixed-field approaches offer significant promise for increasing treatment specificity, minimizing collateral damage, and revealing tumor-selective response windows. Future strategies may benefit from personalizing beam selection and delivery sequence based on tumor type, molecular profile, and repair capacity. In parallel, the development of radioprotective interventions—such as photobiomodulation or adaptive preconditioning—may help safeguard healthy tissues across a range of exposure contexts, from clinical radiotherapy to occupational, environmental, and spaceflight scenarios. As mechanistic understanding of these interactions deepens, such approaches could contribute to comprehensive radiation management strategies.

### 4.4. Space and Environmental Risk Contexts

Space radiation research increasingly shows that combined-field exposures, especially galactic cosmic ray simulations blending protons with high-LET ions, can produce complex biological responses that defy predictions based on single-beam or purely dose-proportional models. Rather than scaling uniformly with total dose, these effects arise from interactions between radiation qualities and exposure sequences, with even low-dose mixed fields reported to disrupt such diverse systems and functions as cognition, synaptic integrity, and behavioral performance in ways often found to differ by sex and genetic background [135,163]. Similar order-dependent sensitivities have emerged in space radiobiological models, where the timing and sequence of proton and high-LET ion irradiation were found to determine not only the magnitude of neoplastic transformation in vitro [130,131] but also, in animal studies, whether tumors develop and how aggressively they progress [136]. These observations strongly support the need for protective strategies that take into account the interplay of radiation quality, sequence, and timing.

In space-related research, chronic exposure has been identified as a key factor influencing biological risk. For instance, in *K-ras*^LA1^ mice, chronic GCRsim regimens were shown to increase lung tumor incidence more than acute exposures [165]. However, this distinction is not limited to typical radiobiological endpoints; several neurobehavioral studies have deliberately contrasted chronic and acute dosing schedules to uncover differences in the persistence, scope, and nature of the effects. In prefrontal cortex function, chronic GCRsim exposure was linked to sustained deficits in attentional performance and slower reaction times, while acute exposure also disrupted neurotransmitter networks but showed less pervasive long-term alterations in dopamine signaling [163]. Similarly, in hippocampal and cortical circuits, chronic irradiation produced thinning of postsynaptic densities, changes in axon myelination, and enduring impairments in synaptic plasticity, changes that acute exposures appeared to trigger less robustly or more transiently [162]. Even in open-field navigation tasks, acute irradiation caused more pronounced, immediate deficits in spatial navigation under lighted conditions, whereas chronic dosing led to subtler but longer-lasting alterations in path integration and exploration [164]. The comparison extends to mixed-field studies where fractionated or prolonged exposures impaired novel object recognition and fear conditioning more consistently than single acute doses [171]. These investigations suggest that chronic regimens do not simply deliver a diluted version of acute effects over time but instead engage biological processes that cumulatively reshape neural networks, behavior, and cancer risk profiles in ways that are especially relevant to the continuous irradiation astronauts would experience on extended missions.

A special entry in the data set, the NASA Twins Study, provides an integrative framework to understand how long-duration spaceflight affects human biology across molecular, cellular, and cognitive domains. Rather than acting as discrete stressors, galactic cosmic radiation (GCR), microgravity, circadian disruption, and confinement appear to converge on shared pathways regulating genomic stability, immune resilience, and normal neurocognitive function. Notably, chronic exposure to GCR was shown to define genome instability in this mission, evidenced by the accumulation of chromosomal inversions and other cytogenetic aberrations that persisted for months after return. The observed telomere elongation during flight, followed by pronounced shortening upon landing, raises critical questions about whether this dynamic reflects transient adaptive remodeling or persistent genomic vulnerability. Similarly, the DNA damage responses triggered by sustained GCR exposure may remain incompletely resolved, potentially elevating oncogenic risk over time. Beyond genomic instability, the study implicates coordinated shifts in transcriptional networks and microbiome composition that may modulate inflammatory tone and metabolic regulation. The persistence of gene expression perturbations, along with sustained alterations in immune mediators and cognitive performance, suggests that protective measures during spaceflight remain insufficient. These observations indicate that the interplay of GCR and other spaceflight stressors drives a complex, multi-system remodeling with health consequences that likely extend well beyond the mission timeframe [175].

Such findings reflect a pivotal turning point in space radiation biology, enabled in large part by NASA’s strategic investment in GCRsim technology and the development of multibeam irradiation platforms like those at the NASA Space Radiation Laboratory (NSRL). The ability to deliver rapid, sequential exposures of biologically relevant ions—mirroring real cosmic ray compositions—has significantly advanced the field from single-ion approximations to nuanced, systems-level modeling. Coupled with omics-driven analyses, behavioral assays, and translational studies like the NASA Twins Study, this technological leap allows for integrative frameworks that capture the interactive, non-linear nature of spaceflight stressors. As long-duration missions to the Moon and Mars approach, future research will increasingly depend on refining these models, identifying individual susceptibility markers, and testing personalized or adaptive countermeasures that address the cumulative burden of mixed-radiation and environmental exposures.

While the spaceflight context has catalyzed some of the most advanced modeling of mixed-radiation effects, such scenarios are not directly representative of terrestrial conditions. Nevertheless, the principles uncovered—such as synergistic interactions, timing-dependent outcomes, and system-level disruption—stress the broader need to investigate combined exposures on Earth. Although everyday environments do not involve showers of galactic ions, people are routinely subjected to overlapping or sequential low-LET sources, radionuclide mixtures, and co-exposures with chemical, thermal, or biological stressors. The insights from GCRsim studies thus can serve as a scientific impetus, urging environmental radiation research to move beyond isolated-dose models and account for real-world exposure complexity.

On Earth, mixed-radiation exposures can arise in various contexts, such as ultraviolet radiation in combination with background gamma fields in high natural radiation areas, or beta–gamma emissions associated with radionuclide contamination from industrial or military activity. Although these exposures differ in scale and composition from those encountered in space or laboratory models, they may still present layered biological challenges. As evidence suggests that interactions between radiation types can diverge from simple additive expectations, closer examination is in order. Continued investigation into such terrestrial scenarios is important for refining our understanding of mixed-exposure effects, supporting environmental protection strategies, and strengthening risk assessment and remediation frameworks.

### 4.5. Methodological Challenges and Experimental Gaps

This review identifies major gaps in combinatorial radiation research. Some radiation pairings—such as protons with UV or beta ([16,17,18,170])—remain largely underexplored despite potential biological significance. Even for more extensively studied combinations like X-ray–neutron or X-ray–alpha, systematic mapping of dose–effect relationships and mechanistic modeling is often lacking.

Moreover, although timing, dose ratio, and radiation order have emerged as critical parameters modulating synergy, many studies treat these variables as secondary rather than primary design features. Systematic investigations of inter-exposure intervals, particularly in time-resolved or cell-cycle-synchronized systems, are relatively rare and represent an important area for future study. Cross-comparative work that spans cell type, species, or tissue specificity is also scarce, making it difficult to generalize findings across biological contexts.

Another under-addressed area involves combinatorial exposures including more than two radiation types. While many studies have tested complex mixtures such as 5-, 6- or 7-ion simplified GCRsims in space radiation models, terrestrial analogs involving triplet exposures (e.g., UV–gamma–alpha or beta) have received little attention to date. There is also a clear need for more integrative studies that combine ionizing and non-ionizing modalities in realistic exposure sequences, such as low-dose UV exposure followed by diagnostic imaging or therapeutic irradiation.

Furthermore, many existing studies have predominantly focused on acute, high-dose exposures, whereas real-world scenarios—whether medical, environmental, or space-related—involve chronic or fractionated mixed exposures. Therefore, experimental paradigms that more closely reflect the complexity of human exposure patterns are required. Bridging these gaps will be important for constructing predictive models of mixed-radiation effects, optimizing therapeutic combinations, and refining risk estimates for spaceflight and environmental contamination scenarios. Achieving this will require interdisciplinary collaboration and coordinated efforts to standardize experimental protocols. Fortunately, recent advances in dosimetry, high-throughput molecular assays, and computational modeling now provide the necessary tools to systematically dissect mixed-radiation effects across diverse biological systems and exposure conditions. Effectively leveraging these capabilities is critical for the field to move beyond partial or descriptive observations and toward more comprehensive mechanistic understanding and improved predictive accuracy.

In addition to possible limitations in the available studies, this systematic review faced several methodological challenges of its own. To begin with, no formal review protocol was prepared and registered; therefore, other research teams cannot fully reconstruct the initial scope or trace every decision made throughout the review process. Moreover, as previously discussed, the diversity of biological systems, endpoints, and study objectives limited the relevance of conventional appraisal tools. Consequently, narrative synthesis was selected over statistical approaches due to wide data variability, which made quantitative analysis unfeasible. While appropriate for such data, a potential drawback is that this method relied more on expert judgment to interpret and integrate findings across studies. Furthermore, the search process, while broad and iteratively refined across multiple databases, was conducted manually, introducing a minor risk of omitting relevant studies, despite repeated reviewer screenings.

Another methodological concern arose from the wide temporal distribution of studies across radiation combinations. Certain pairings—notably UV and X-rays—were predominantly investigated during the 1960s–1990s, using experimental approaches that often predate contemporary molecular techniques or the use of in vivo models. Few recent studies have revisited these combinations using modern methodologies, and therefore, their effects cannot always be evaluated under current biological standards. As such, while these studies provide useful mechanistic insights, they may not fully reflect current experimental standards or align with our modern understanding of the biological complexity involved. Nevertheless, they were intentionally included to capture the full conceptual scope of mixed-radiation effects. Rather than excluding them due to age, we chose to integrate their findings carefully, using them to illuminate conserved response patterns, while avoiding unwarranted generalizations. This approach aims to preserve the scientific value of legacy data without irreparably compromising the interpretive rigor of the review.

As a closing remark, the challenges encountered in preparing this extensive review are acknowledged in the interest of methodological clarity, yet they do not compromise the coherence or scientific validity of the review’s findings.

### 4.6. Limitations and Future Directions

One of the main challenges in mixed-radiation research lies not only in the scarcity of data but also in how current studies are designed, conducted, and contextualized. Investigations into radiobiological, medical, and space-related exposures are often pursued within separate research domains, each shaped by its own priorities, methodologies, and regulatory frameworks. This disciplinary isolation makes it difficult to compare results, integrate findings, or construct unified frameworks for understanding biological response. As a result, efforts to build predictive models—whether theoretical, computational, or mechanistic—face significant hurdles. Such models are essential not only for estimating health risks but also for developing effective radioprotection strategies. Yet their development is hindered by fragmented data, inconsistent endpoints, and a lack of shared experimental standards. In addition, potential synergistic effects may be shaped not only by radiation type and timing but also by differences in dose rates among combined exposures—an underexplored variable that may significantly influence biological outcomes and remains difficult to replicate experimentally. Compounding this, most studies prioritize acute or short-term outcomes, with far fewer addressing how mixed-radiation effects evolve over time, across generations, or within realistic ecological and clinical contexts. For example, the extremely low dose rates characteristic of ambient space radiation—where synergistic effects may unfold gradually over time—remain experimentally inaccessible at ground-based facilities, which reflects a fundamental limitation in our ability to model the biological complexity of real spaceflight conditions.

To move the field forward, we need better coordination between disciplines, along with shared databases that connect radiation types, biological systems, and outcomes. Examples of such efforts include the NASA Open Science Data Repository and the European Radiobiological Archives (ERA), hosted by the German Federal Office for Radiation Protection (BfS), which offer structured datasets from both spaceflight and ground-based experiments. These resources support broader comparisons and enable the identification of recurring patterns across studies. Advancing the field also demands robust international coordination, standardized methodologies, and full integration of simulation and modeling platforms into experimental design. Collectively, these steps can help unify fragmented research into a coherent framework for guiding risk assessment, medical planning, and spaceflight readiness.

## 5. Conclusions

This systematic review—drawing on over 170 studies and encompassing an exceptional variety of radiation pairings—demonstrates that the biological effects of combined radiation exposures are neither trivial nor reliably additive. From fundamental radiobiology to medical applications and spaceflight scenarios, evidence consistently shows that interactions between different radiation types can yield emergent effects, including both synergistic and antagonistic responses. These outcomes are often shaped by radiation quality, dose ratio, sequence, and the biological system in question.

Crucially, the findings challenge the assumption that total dose alone can predict biological risk. Instead, they point to the need for a better understanding of how different radiation types interact over time and across levels of biological organization—from molecular repair mechanisms to whole-organism physiology. Although the field remains fragmented and methodologically uneven, the available data support the development of integrative models and coordinated research strategies that more accurately reflect the complexity of mixed-radiation exposures.

Conclusively, the evidence presented in this review reflects a pressing need to rethink how we approach radiation exposure in both research and practice. Whether in medical, space, or environmental contexts, the biological impact of radiation cannot be fully understood without accounting for interactions across type, order, timing, and biological scale. This complexity is not merely theoretical—it is central to designing safer therapies, preparing for deep space missions, and protecting life on Earth. Moving beyond reductionist models is essential to predict, mitigate, and ultimately manage the multifaceted nature of mixed-radiation effects.

## Figures and Tables

**Figure 1 biomolecules-15-01282-f001:**
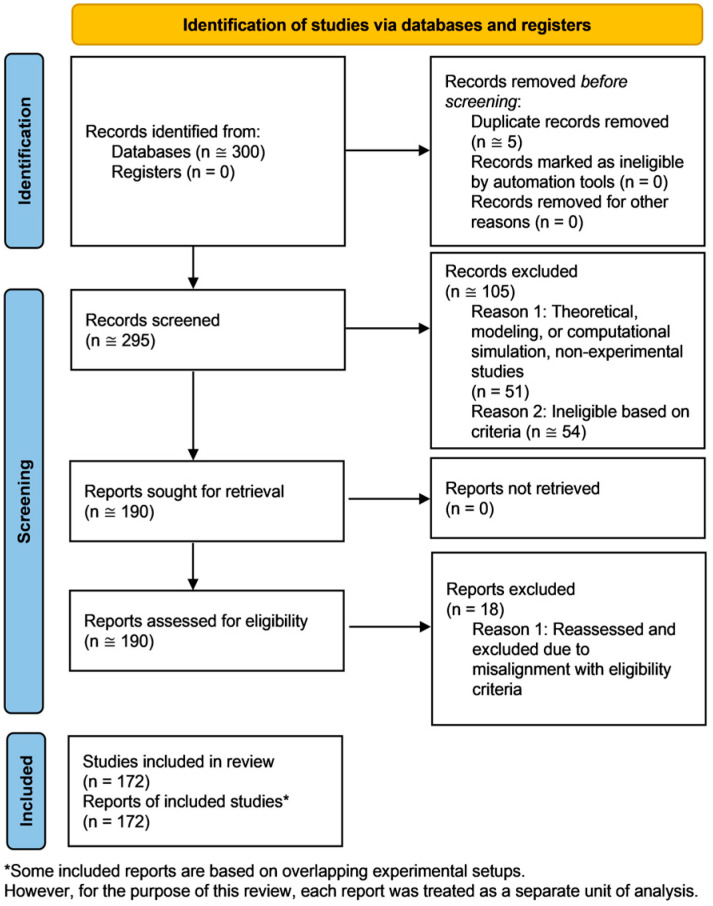
PRISMA 2020 flow diagram detailing the study selection process according to PRISMA guidelines (BMJ 2021; 372: n160). A total of 300 records were identified through databases. After duplicate removal and screening, 190 reports were assessed for eligibility. Of these, 18 were excluded upon reassessment due to misalignment with eligibility criteria. The final number of included studies was 172. This work is licensed under CC BY 4.0. To view a copy of this license, visit https://creativecommons.org/licenses/by/4.0/ (accessed on 20 June 2025).

**Figure 2 biomolecules-15-01282-f002:**
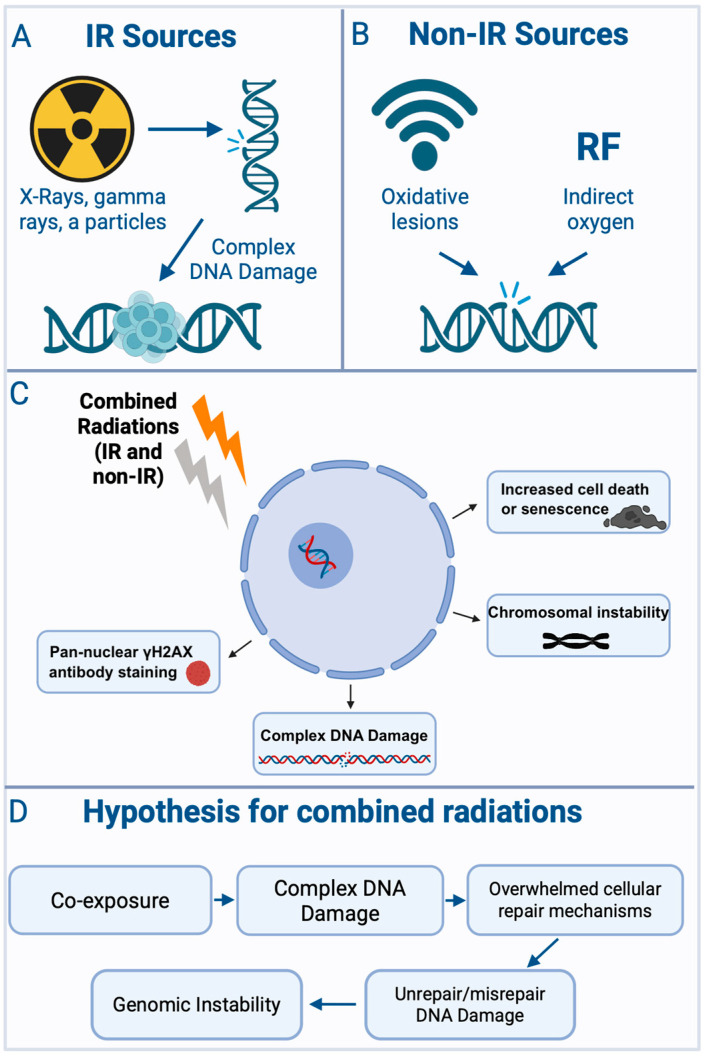
Hypothesis for combined exposure effects. (**A**) Ionizing radiation (IR) sources cause direct and severe complex DNA damage, often involving clustered lesions that are difficult to repair; (**B**) non-ionizing radiation (non-IR) sources, such as radiofrequency (RF) fields, have been reported to induce oxidative stress primarily through indirect mechanisms involving reactive oxygen species (ROS) and oxygen intermediates, which may contribute to DNA strand breaks and base modifications; (**C**) combined exposures to IR and non-IR can result in synergistic or additive genotoxic effects, including DNA damage (detectable by assays such as γH2AX foci formation) and chromosomal aberrations. These combined effects may increase the likelihood of cell death, senescence, or chromosomal instability; (**D**) proposed hypothesis for combined radiation effects: co-exposure to IR and non-IR could enhance the formation of complex DNA damage, which can overwhelm cellular repair mechanisms, leading to unrepaired or misrepaired DNA, driving genomic instability and contributing to long-term biological consequences. The figure was created using BioRender, available at https://app.biorender.com/illustrations (accessed on 20 June 2025).

**Table 1 biomolecules-15-01282-t001:** Radiobiological studies on combinations of non-ionizing and ionizing radiation.

Type and Dose	Biological Systems	Biological Endpoints	Effects	Ref.
UV-C (254 nm): 70 erg/mm^2^; X-rays: 150 rad; irradiations 2–6 h post-PHA stimulation	PHA-stimulated human lymphocytes (G_1_ stage)	Dicentric and chromatid-type chromosome aberrations (first mitosis)	• No synergistic effect observed • Dicentric yields from combined exposures matched X-rays alone • UV induced only chromatid-type aberrations • Suggests G_1_ cells possess enhanced repair capacity preventing UV–X-ray interaction	[4]
UV-C (253.7 nm): 40–100 erg/mm^2^; X-rays (260 kV): 125–200 rad; exposures ≤ 30 s apart	Human peripheral blood lymphocytes (unstimulated, G_0_ stage)	Dicentric chromosome formation (cytogenetic analysis)	• ~2-fold synergistic ↑* in dicentrics vs. X-rays alone • Synergy consistent across UV doses and exposure order • Suggests interaction between X-ray-induced breaks and UV-mediated repair inhibition or lesion overlap	[5]
UV-C (254 nm): 65–95 erg/mm^2^; X-rays (260 kVp): 150 rad; exposures given in sequence with <30 s interval	Human peripheral blood leukocytes from healthy donors; human peripheral blood leukocytes from Down’s syndrome patients	Chromosome aberrations (dicentrics); DNA repair synthesis ([^3^H]thymidine incorporation)	• ↑ Dicentrics after combined UV + X-rays vs. X-rays alone (2× in controls, 27% in Down’s syndrome) • ↓* UV-induced DNA repair synthesis in Down’s syndrome cells, synergism suggests shared repair pathways	[6]
UV-C (254 nm): 6 J/m^2^; X-rays (260 kVp, 0.5 Gy/min): 1.5 Gy; timing between irradiations varied from <30 s to 90 min	Human peripheral blood lymphocytes in G_0_ phase	Dicentric chromosome yield per cell	• UV followed by X-rays produced a stable 2-fold ↑ in dicentrics across all timing intervals • X-rays followed by UV showed decreasing synergistic effect with ~20 min half-life • Indicates short-lived DNA lesions responsible for early chromosome exchange events	[7]
UV-C (254 nm): 100–175 erg/mm^2^; X-rays (50 kVp): 450–700 rad; applied sequentially within <40 s; some experiments up to 6 h intervals	Chinese hamster V79 cells (synchronous populations in mid-S-phase)	Cell survival (colony-forming ability)	• Combined exposure resulted in additive ↓ in survival, removal of shoulder in survival curves • Order of exposure modulated effect, with UV preceding X-rays causing greater loss of sublethal damage repair capacity	[8]
UV-C (254 nm): up to 1000 ergs/mm^2^; X-rays (50 kVp): up to ~10 krads/min	*E. coli* K-12 wild-type; DNA repair mutants (*uvrA*, *uvrB*, *uvrC*, *recA*, *recB*, *recC*, *exrA*, *polA*)	Cell survival; DNA single-strand break repair (alkaline sucrose gradients)	• ↑ X-ray sensitivity after UV pretreatment (wild-type and *uvr* mutants) • UV inhibits type III repair of X-ray-induced DNA SSBs • No synergism in *recA*, *recB*, *recC*, *exrA* mutants	[9]
Near-UV (365 nm): up to 2.5× 10^6^ J/m^2^; X-rays (250 kVp): up to 19.6 krad (2 krad/min); UV administered ~3–5 min before X-rays	*E. coli* K12 strains (wild-type W3110, *polA* mutant P3478, *recA recB* mutant SR111)	Clonogenic survival; single-strand break rejoining (alkaline sucrose gradients); DNA degradation (TCA-insoluble material)	• Pretreatment with 365 nm UV enhanced X-ray lethality (up to 3.4× slope ↑ in survival curves) • Synergism absent in *recA* recB mutants • 365 nm inhibited type II and III DNA repair • Full-medium incubation partially restored repair in wild-type and *recA recB* but not in *polA*	[10]
UV-C (254 nm): up to 960 ergs/mm^2^; X-rays (150 kVp): up to 40 krads	*E. coli B/r* and *Bs-1* strains; 5-BU substituted DNA and purine-starved cells	Cell survival (colony-forming ability)	• ↑ X-ray sensitivity after UV pretreatment in *B/r* (up to 3×) • No synergism in 5-BU-substituted or purine-starved cells • UV and 5-BU inhibit X-ray repair • No effect in *Bs-1* cells	[11]
UV-C (254 nm): 1–120 J/m^2^; X-rays (100 kVp): 6.7–26.6 krad or alpha particles (4.5 MeV, LET 140–180 keV/μm): 4–48 krad	Diploid *Saccharomyces cerevisiae* strains (wild-type, *rad2* mutant, *rad9* mutant)	Colony-forming ability	• ↑ Synergistic loss of colony-forming ability when X-rays precede UV • Reduced UV survival shoulder, altered slopes depending on repair genotype • Synergism absent or reduced in *rad2* and *rad9* mutants	[12]
UV-C (254 nm): 170 J/m^2^; gamma rays: 90 krad; sequential exposures, no delay	*Schizosaccharomyces pombe* (wild-type 972h^−^) and *rad1* mutant	Inactivation (colony survival); recovery kinetics	• Strong synergistic interaction in wild-type cells • Survival dropped > 100× beyond additive prediction • No synergism in recombinational repair-deficient *rad1* mutant • Recovery kinetics depend on second radiation type, not first	[13]
UV-C (253.7 nm): 270–810 J/m^2^; gamma rays: 0.1–0.3 Mrad; exposures ≤ 5 min apart	*Micrococcus radiophilus*	Survival/colony-forming ability	• Synergistic killing effect with UV pretreatment (3.5× increased gamma ray sensitivity) • Additive effect with gamma pretreatment	[14]
UV-B (300 nm): 20, 40, 60, 80, 100, and 120 s exposure time, fixed distance of 50 cm; gamma rays: 500–3000 Gy	Wild-type and mutant *Bacillus velezensis* strains from soil and oil samples	Functional microbial endpoints: surfactin yield; isoform distribution (MALDI-TOF); antifungal bioactivity (zone of inhibition); emulsification capacity (E24); colony morphology	• Mutant AF-UVγ2500 showed ~2× higher surfactin yield (1.62 g/L vs. 0.85 g/L) • Stronger antifungal activity, 3× isoform abundance • Sequential UV and gamma ray mutagenesis more effective than either treatment alone	[15]
Beta particles (^32^P, internal): time-integrated decay (up to 25 mc/mg P); UV-C (254 nm): 120–530 erg/mm^2^; exposures sequential with varied timing and order	*Salmonella typhimurium* strain LM2	Inactivation (colony-forming ability)	• Strong synergistic effect between ^32^P decay and UV damage • Reciprocity observed (UV→^32^P and ^32^P→UV) • Synergy eliminated by photoreactivation, suggesting interaction between nonlethal DNA lesions (e.g., thymine dimers and ^32^P-induced strand breaks) within ~25 nucleotide pairs	[16]
Electron beam (32 MeV): 21–28 krad; UV-C (254 nm): 450 erg/mm^2^; exposures ≤ 2 min apart	*E. coli B/r*	Survival/colony-forming ability	• Synergism observed • Inhibited by 3 h liquid holding between irradiations regardless of order • Attributed to excision repair activity	[17]
UV-C (254 nm): 150 ergs/mm^2^; Protons (LET 20 keV/µm): 8–40 krads	*E. coli B/r* (wild-type, log-phase)	Survival (loss of reproductive capacity)	• UV pre-exposure sensitized cells to protons • Survival curve slope increased 1.5–1.7× • Synergy not observed when reverse order applied or with other radiation types • Suggests repair system involvement	[18]
NIR (600–1400 nm, non-thermal): 360 kJ/m^2^ (30 min); X-rays (90 kV, 5.23 Gy/min): 1–4 Gy; sequential exposure with NIR pretreatment 30 min before X-rays; temperature-controlled setup	Human full-thickness skin model (primary dermal fibroblasts and keratinocytes)	DNA double-strand breaks (53BP1, γH2AX); cell proliferation (BrdU, Ki-67); apoptosis (TUNEL); morphology (H&E)	• NIR pretreatment delayed repair of X-ray-induced DSBs • Protected fibroblasts from apoptosis • Counteracted X-ray-induced proliferation inhibition in keratinocytes • No morphological disruption observed • Photobiomodulation modulated radiation stress response	[19]
Low-intensity laser radiation (670 nm): 5.3–16 J/cm^2^ over 3–4 days, finished 24 h before gamma irradiation; gamma rays (^137^Cs): 1 or 3 Gy whole-body;	Wistar rats (peripheral blood in vivo)	Cell counts (RBC, WBC, LYM); enzyme activity (SOD, catalase); blood oxygenation	• No synergistic toxicity • Laser pretreatment ↑ leukocyte counts, antioxidant enzyme activity, and oxygenation • Evidence of radioprotective effects from laser pre-irradiation	[20]

* ↑ indicates all language forms related to increase; ↓ indicates all language forms related to decrease.

**Table 2 biomolecules-15-01282-t002:** Radiobiological studies on combinations of non-ionizing and non-ionizing radiation.

Type and Dose	Biological Systems	Biological Endpoints	Effects	Ref.
UVA (320–400 nm): 0.5, 1, 2 J/cm^2^; VIS (395–600 nm): 2, 4, 6, 8 J/cm^2^; Mixed UVB (5%, 310–320 nm est.), UVA (25%, 320–400 nm), visible light (70%, 395–600 nm) = mUV/VIS: 2, 4, 6, 8 J/cm^2^ (i.e., 0.1–0.4 J/cm^2^ UVB)	Human volunteers (ragweed-allergic patients, n = 19); forearm skin tested	Skin prick test (SPT) wheal size; allergen-induced immediate hypersensitivity response	• Only mUV/VIS (not UV-A or VIS alone) caused significant, dose-dependent inhibition of allergen-induced wheal formation (up to 83% at 8 J/cm^2^) • Inhibition occurred even at suberythematous doses • Suggests synergistic action of UV and VIS light on mast cell-mediated response	[21]
UV-B (290–320 nm); UV-A (320–400 nm); VIS (400–740 nm); Near-infrared (predominantly IR-A, 760–1400 nm); combined and filtered exposure (natural sunlight fractions)	Human skin tissue (in vivo biopsies)	Photoaging biomarkers (MMP-1, MMP-9, expression, type I procollagen levels), inflammatory cell infiltration (neutrophils, macrophages)	• Full-spectrum sunlight elicited stronger biomarker changes than UV-filtered or heat-only conditions • Composite effects consistent with contributions from multiple spectral components • Enhanced MMP-1 expression, reduced type I procollagen, inflammatory cell infiltration • Neutrophil recruitment required UV, whereas macrophage infiltration also occurred with visible/IR and heat	[22]
UV-A (315–400 nm): 25 J/cm^2^; UV-B (280–315 nm): 0.014 J/cm^2^, alone and in combinations over 600 days	Albino hairless mice (SKH:HR1 strain)	Tumor induction (SCC, papillomas, solar keratoses); tumor latency (t_50_); size-specific incidence	• UV-A and UV-B showed additive effects on SCC induction • No significant synergism or antagonism • Papillomas more frequent under UV-A • UV-B enhanced papilloma induction by UV-A	[23]
UV-A (379.68 nm): ~11.65 µW/cm^2^; UV-B (305.22 nm): ~8.65 µW/cm^2^; 6 h/day exposure for up to 27 days	Scyphozoan jellyfish polyps (*Aurelia* sp.)	Asexual reproduction (budding rate), survival (mortality), substrate detachment, tentacle condition (retraction/loss), feeding behavior	• Strong synergistic effect: combined UV-A + UV-B caused 100% mortality by day 21 • No budding, total loss of attachment and feeding • UV-B alone reduced reproduction and health • UV-A alone had minimal impact • Combined exposure drastically worsened all outcomes	[24]
UV-B (280–315 nm): 1080–3600 J/m^2^; UV-C (100–280 nm): 280–930 J/m^2^	*Sclerotinia sclerotiorum*; tomato plants	Sclerotia germination; mycelial growth; ROS production; lipid peroxidation; SOD/CAT activity; disease severity; chlorophyll levels; fruit yield; defense gene expression	• Combination had strongest pathogen suppression and yield/quality improvement • ↑* Defense gene expression and antioxidants	[25]
UV-A (365 nm): 49 W/m^2^; UV-B (311 nm): 8 W/m^2^; simultaneous exposure for 6–8 h/day	*E. coli* MG1655 (wild-type and mutant strains)	Colony-forming ability; log_10_ CFU reduction; whole-genome mutations; tRNA photodamage (UVA absorbance, s^4^U-mediated)	• Strong synergistic inactivation effect (~100× more than UV-B alone) • Synergy traced to thiouridine (s^4^U) in tRNA absorbing UV-A and impairing protein synthesis, reducing DNA repair • Mutants with *ThiI* gene alterations lost synergy, confirming mechanistic link	[26]
UV-A (365 nm): 600–1200 mJ/cm^2^; UV-C (268 nm): 2.5–20 mJ/cm^2^; sequential exposure with UV-A first, intervals 0–24 h; LED-based setup	*E. coli* K12 MG1655 (wild type) and SP11 (*ThiI* mutant)	Inactivation (CFU assay), growth delay (OD600), single-cell division time, tRNA photodamage	• Strong synergistic inactivation in wild type (up to 5.7 log_10_ reduction with UV-C after UV-A) • Effect absent in *ThiI* mutant lacking s^4^U tRNA modification • Synergy linked to UV-A-induced translational arrest via tRNA damage • Effect persists up to 24 h post UV-A	[27]
UV-LEDs (267 nm, 275 nm, 310 nm): 0.384 mW/cm^2^; combined exposures (267/275, 267/310, 275/310 nm) matched for irradiance; fluences of 8.78–23.04 mJ/cm^2^ for 3–4 log inactivation	*E. coli* (strain CGMCC 1.3373) in water suspension	Inactivation efficiency (log reduction), photoreactivation and dark repair	• No synergistic effect observed for combined wavelengths • 267 nm UV-LED had highest inactivation efficiency • 275 nm showed strongest resistance to reactivation, likely due to protein damage • Photoreactivation was dominant over dark repair	[28]
UV-C (222 nm): 0.32 mW/cm^2^; UV-C (275 nm): 0.50 mW/cm^2^; delivered simultaneously for 12–20 s	*E. coli* ATCC 15597 (bacteria) and PR772 (bacteriophage) in PBS suspension	Log_10_ microbial inactivation; photoreactivation and dark repair; DNA damage; ROS production	• Marked synergistic inactivation of *E. coli* (synergism coefficient up to 1.92) • No synergism for PR772 • DWUV suppressed photoreactivation in both organisms • Enhanced ROS and protein damage likely mechanism for synergy in bacteria	[29]
UV-C (260 nm): 38–122 mJ/cm^2^; UV-C (280 nm): 41–89 mJ/cm^2^; UV-C (260|280 nm): 41–105 mJ/cm^2^	*E. coli*, MS2 coliphage, Human adenovirus type 2 (HAdV2), *Bacillus pumilus* spores	Inactivation kinetics (log_10_ reduction), DNA/RNA damage (qPCR)	• No synergistic effect observed • Combined 260|280 nm inactivation matched additive sum of individual wavelengths • No enhanced DNA/RNA damage or energy efficiency • Supports fluence-based independence of dual UV wavelengths	[30]
UV-C (222 nm): 1.0–2.4 mJ/cm^2^; UV-C (282 nm): 0.8–2.1 mJ/cm^2^; fluences for 5-log inactivation: *E. coli* 2.4–2.6 mJ/cm^2^, *E. faecalis* 3.6–5.4 mJ/cm^2^	*E. coli* and *Enterococcus faecalis* in synthetic water (pH 6.4–7.0)	Log_10_ microbial inactivation (CFU count); photoreactivation and dark repair	• Strong time-based synergistic effect in all dual-wavelength combinations (φ = 1.3–3.8) • No dose-based synergy • Complete inactivation achieved rapidly • Synergy linked to combined protein damage and DNA damage mechanisms • No repair observed after DWUV exposure	[31]
UVA (365 nm): 1.7–52 J/cm^2^; UVC (265 nm): 4.2–20 mJ/cm^2^	*E. coli* (ATCC 11229) and coliphage MS2 (ATCC 15597-B1) in PBS suspension	Log_10_ inactivation; photoreactivation and dark repair; ROS-mediated effects (inferred via scavenger assays)	• UV-A pretreatment significantly enhanced UV-C inactivation of *E. coli* (up to +2.2 log) • Eliminated photoreactivation by impairing self-repair (via hydroxyl radicals inside cells) • No synergy observed for MS2 • Simultaneous UV-A+UV-C decreased *E. coli* inactivation (photoreactivation effect)	[32]
UVA (369 nm), UVB (288 nm), UVC (271 nm), dual UV (288/271 nm, 369/288 nm, 369/271 nm): 0.75–6.75 mJ/cm^2^	*E. coli*, *Staphylococcus epidermidis*, *S. Typhimurium*, *Serratia marcescens*, *Pseudomonas alcaligenes* on agar (simulated food surface)	Log_10_ inactivation (colony count)	• Significant synergistic effect observed only for 288/271 nm (UV-B/UV-C) on E. coli, *S. epidermidis*, and *S. Typhimurium* • Synergy ratios 0.20–0.87 • No synergy with 369/271 or 369/288 combinations • Pulsed and continuous modes equally effective at same dose	[33]
UVA-LED (365 nm): 3240 mJ/cm^2^; UVC-LED (275 nm): 375–750 mJ/cm^2^, applied sequentially in recirculating water system	*E. coli* (ATCC 8099) in aquaculture recirculating water	Survival (log inactivation), photoreactivation and dark reactivation rates	• Sequential UVA-UVC irradiation produced ~2–3 log higher inactivation than UVC alone • UVA pretreatment enhanced bactericidal efficacy and reduced bacterial reactivation • Higher UVC doses further suppressed reactivation	[34]
UV-A (365 nm): 700 W/m^2^; UV-C (254 nm): 0.7 W/m^2^; simultaneous irradiation for 6 min in 96-well plate format	*Vibrio parahaemolyticus* WT and *recA*/*lexA* mutants; cultures in LB broth	DNA damage (CPDs, 8-OHdG); log survival (colony-forming ability); SOS-dependent repair capacity	• Simultaneous UV-A+UV-C caused synergistic bactericidal effect (log survival −3.3) vs. additive effects of single/seq. exposure (−2.1) • CPD repair suppressed • Synergy absent in SOS-deficient mutants, implicating RecA/LexA-dependent repair in survival	[35]
UV-C doses: 5–20 mJ/cm^2^; irradiance: 0.194 mW/cm^2^ (260 nm), 0.314 mW/cm^2^ (280 nm), 0.473 mW/cm^2^ (260/280 combined); simultaneous exposure	Enteroviruses (CVA10, Echo30, PV1, EV70) in water suspension; propagated in BGMK cells	Log_10_ inactivation (infectivity via ICC-RTqPCR)	• No synergistic effect observed • 260 nm alone was most effective • Dual 260/280 nm either matched or underperformed vs. 260 nm • 280 nm less effective overall • Results consistent with nucleic acid absorption peak near 260 nm	[36]
UV-C (222 nm): 0–25 mJ/cm^2^; UV-C (254 nm): 0–25 mJ/cm^2^; UV-C (255/265/285 nm): 0–25 mJ/cm^2^ each	MS2 bacteriophage (virus surrogate) in water suspension (host: *E. coli* Famp)	Log_10_ inactivation (PFU count)	• Significant synergy observed for LP or excimer lamps followed by LEDs • Enhanced disinfection vs. additive predictions • Reverse sequences less effective • Excimer + LP sequence showed highest energy efficiency • Supports order-dependent synergy in UV-UV disinfection	[37]
UVC (222 nm or 280 nm); 405-nm blue light; pretreatment: 30 s (222-nm: 7.1 mJ/cm^2^, 280-nm: 1.2 mJ/cm^2^); 405-nm: up to 48 h, 86.4 J/cm^2^	*E. coli*, *Listeria monocytogenes*, *Staphylococcus aureus*, *S. Typhimurium*, *Pseudomonas aeruginosa* (in vitro)	Survival/colony counts; membrane integrity; ROS generation	• Synergistic bactericidal effect on *E. coli*, *L. monocytogenes*, *S. Typhimurium* • Minor for *S. aureus* • Antagonistic for *P. aeruginosa* • Synergy linked to ↑ ROS and membrane damage	[38]

* ↑ indicates all language forms related to increase.

**Table 3 biomolecules-15-01282-t003:** Radiobiological studies on combinations of ionizing and ionizing radiation.

Type and Dose	Biological Systems	Biological Endpoints	Effects	Ref.
Alpha particles (LET 100–238 keV/µm): 0.1–0.2 Gy; X-rays (80 keV): 0.1–0.6 Gy	Human TK6 cells (wild-type and *hMYH* knockdown)	Clonogenic survival, mutant frequency	• Mixed beams show synergy in wild-type cells (survival) • MYH^−^ cells resistant to survival loss but show high mutant frequency • Oxidative stress role unclear	[39]
Alpha particles (^241^Am, 2.88 MeV, LET ~129 keV/µm): 0.25–2 Gy; X-rays (225 kVp, 0.59 Gy/min): 0.25–3 Gy; applied sequentially with intervals from 15 min to 6 h	PC-3 prostate cancer cells and U2OS osteosarcoma cells	Clonogenic survival	• Sequential mixed-field exposures showed significant sublethal damage repair • RBESLD ~2.8–3.7 • Repair kinetics similar to X-rays • Order of exposure slightly modulated survival at late timepoints	[40]
Alpha particles (2.9 MeV, LET ~140 keV/μm, 11–45% of total dose); X-rays (250 kVp, dose rate 0.1 Gy/min); total dose of 1–10 Gy; sequential exposure (alpha then X-ray)	T-1 human kidney cells; aerobic and hypoxic conditions	Clonogenic survival (aerobic and hypoxic conditions)	• Alpha particle irradiation ↑* RBE (~2.3 at 10% survival) and ↓ OER sharply (~1.0) • Mixed radiation ↓ OER further • Observed trends aligned with theory for mixtures of high- and low-LET radiation	[41]
Alpha particles (LET 90.9 keV/μm); X-rays (190 keV); 1:1 dose ratio mixed beam (e.g., 1 Gy = 0.5 Gy alpha particles + 0.5 Gy X-rays); total dose of 0.5–2 Gy; simultaneous exposure using custom irradiation setup	Human peripheral blood lymphocytes (from 4 donors)	DNA damage response gene expression (*FDXR*, *GADD45A*, *MDM2*, *BBC3*, *CDKN1A*, *XPC*, qPCR at 24 h post-exposure)	• Mixed beams induced gene expression levels ≥ alpha alone • Synergy detected in 3 of 4 donors using “envelope of additivity” • *FDXR* most responsive • ATM inhibition decreased response, indicating role in synergistic effect	[42]
Alpha particles (LET 90.92 keV/μm): 0.13–0.54 Gy; X-rays (190 kV): 0.20–0.80 Gy; mixed beams included 0.20X + 0.07 alpha, 0.40X + 0.13 alpha, and 0.40X + 0.27 alpha (doses in Gy); simultaneous exposure via dual-source setup	Human peripheral blood lymphocytes (PBL) from 1 donor	Chromosomal aberrations (simple vs. complex, via FISH in chromosomes 2, 8, and 14)	• Significant synergistic effect at the level of complex aberrations for two highest mixed doses • Linear-quadratic dose–response for complex events • ↑ damage complexity suggests higher-than-additive biological effect	[43]
Alpha particles (LET 97–238 keV/μm): 0.13–1.33 Gy; X-rays (190 kV): 0.25–2.00 Gy; mixed-beam doses: 0.38, 0.77, 1.53 Gy; simultaneous exposure using custom dual-source irradiator at 37 °C	Human peripheral blood lymphocytes (1 male donor)	Micronucleus (MN) frequency in binucleated cells; MN size	• All mixed-beam doses showed statistically significant synergistic effects (average 1.8× higher MN than additive prediction) • Linear dose–response • Synergy attributed to impaired repair of X-ray-induced damage by prior alpha exposure	[44]
Alpha particles (LET 100–172 keV/μm): 0–1 Gy; X-rays (80 keV): 0–1 Gy (always 1:1 ratio in mixed exposures)	Human osteosarcoma U2OS cells expressing 53BP1-GFP	DNA double-strand break focus formation and decay (53BP1 foci); ATM and p53 activation	• Strong synergistic interaction observed in both small and large DSB foci • Slower focus decay and prolonged ATM/p53 signaling suggest overwhelmed DNA repair • Synergy most pronounced at lower total doses	[45]
Alpha particles (LET 100–238 keV/μm): 0.13–0.32 Gy; X-rays: 0.20–0.80 Gy; mixed beams: 25% alpha + 75% X-rays (e.g., 0.53 Gy ≅ 0.13 alpha + 0.40 X); simultaneous exposure using MAX dual-source system	Human VH10 fibroblasts (immortalized)	γ-H2AX focus formation and repair kinetics (IRIF); small vs. large focus quantification	• No dose–response synergy detected at 1 h • Mixed-beam exposure delayed formation and decay of large foci compared to predicted additive effect, indicating a transient impairment of DNA damage response	[46]
Alpha particles (0.223 Gy/min, LET ~91 keV/μm); X-rays (0.052–0.068 Gy/min); total dose of 0–2 Gy; 1:1 ratio in mixed beam; exposures via dual-source platform on blood discs (simultaneous delivery)	Human peripheral blood lymphocytes (PBLs, from 3 donors)	DNA damage (alkaline comet assay), repair kinetics, phosphorylated DDR proteins (ATM, DNA-PK, p53), gene expression (qPCR)	• Mixed beams caused significantly more DNA damage than additive prediction (synergy via envelope analysis) • Delayed repair kinetics • Highest activation of DDR proteins and gene expression vs. either radiation alone • Results support synergistic impairment of repair by clustered + dispersed DNA damage	[47]
Alpha (LET 100–238 keV/μm); X-rays (peak 80 keV); 0.5 Gy each; simultaneous delivery using a dual-source irradiator	Human osteosarcoma U2OS cells	53BP1-GFP foci kinetics, area, intensity, mobility (live microscopy), chromatin dynamics	• Mixed-beam foci showed unique dynamic behavior • Intermediate size, highest intensity, low mobility, and persistent signal • Distinct from additive prediction • Results support synergistic effect via impaired repair and complex DSB clustering at chromatin domains	[48]
Alpha particles (LET 91 keV/μm); X-rays (190 kV, 3:1 ratio); total dose of 2 Gy, acute or 0.4 Gy × 5 fractionated	Human microglial HMC3 cells; cultured in vitro on Mylar-covered disks	γH2AX foci, CDKN1A and MDM2 expression (qPCR), IL-1β (ELISA), NF-κB/STING phosphorylation, phagocytosis capacity	• Fractionated alpha or alpha + X-ray exposure→stronger pro-inflammatory response and DNA damage signaling than acute • ↑ IL-1β, CDKN1A, MDM2, STING/NF-κB activation, and phagocytosis • Responses returned to baseline by day 14	[49]
Alpha particles (^241^Am): 0.05–1 Gy; X-rays (150 kVp, 0.356 Gy/min): 0.05–1 Gy	BEAS-2B (human lung epithelial); SVEC4-10EHR1 (mouse endothelial) cells	γ-H2AX foci count, size distribution, and decay (dephosphorylation rate) over 24 h	• Alpha-induced foci were larger and dephosphorylated more slowly than X-ray-induced foci • Radiation type correctly identified in >80% of blinded tests • Individual alpha and X-ray doses estimated within 12% error using mixed-beam exposure data	[50]
Alpha particles: 0.166–0.994 Gy; X-rays: 0.25–1.0 Gy; mixed beam 1: 75% X-rays + 25% alpha (0.333–1.327 Gy); mixed beam 2: 50% X-rays + 50% alpha (0.249–0.999 Gy); simultaneous exposure using custom MAX irradiator at 37 °C	Chinese hamster ovary (CHO) cells (AA8)	Clonogenic survival (colony formation assay)	• No synergistic effect observed • Mixed-beam survival data fell within or near predicted additive models • Envelopes of additivity and mathematical modeling confirmed additivity for both mixed-beam conditions	[51]
Alpha particle priming doses (LET ~140 keV/μm): 0.5, 2, or 2.5 Gy; X-rays (250 kV, 3 Gy/min): multiple doses up to ~12 Gy; irradiations separated by ≤3–4 min	V79 Chinese hamster lung fibroblast cells	Clonogenic survival	• Sequential exposure with ≤4 min delay showed strong synergistic effect at 2.5 Gy alpha • Survival curve shoulder (Dq) nearly eliminated • X-ray survival curve slope (Do) unchanged • Synergy attributed to alpha-induced damage interfering with sublethal damage repair from X-rays	[52]
Alpha particles (5.50 MeV, 0.18 Gy/min): 0–3 Gy; X-rays (280 kVp, 0.75 Gy/min): 0–15 Gy; mixed exposures: 0.06 or 1 Gy alpha + graded X-ray doses	Rat lung epithelial cells (F344, LEC)	Cell survival (clonogenic), micronuclei induction (FISH), mitotic delay	• Simultaneous exposures caused greater-than-additive effects on cell killing and micronuclei frequency • High-dose alpha (1 Gy) removed shoulder from survival curve • Synergistic slopes observed in micronuclei assays even at low alpha dose (0.06 Gy)	[53]
Alpha particles (Columbia University charged-particle microbeam): 1 or 20 particles per nucleus; X-rays: 0.02–0.5 Gy pretreatment, 4h before alpha exposure; 3 Gy challenge, 4 h after alpha exposure	Human–hamster hybrid AL cells (CHO–human chromosome 11)	Mutation at CD59 locus (analysis via multiplex PCR)	• Low-dose X-ray priming (0.02–0.1 Gy) suppressed bystander mutagenesis (~58–62% reduction) • 0.5 Gy priming had minimal effect • Bystander cells showed elevated mutant yield after 3 Gy X-ray challenge (supra-additive) • Priming + alpha exposure increased complex CD59− mutation spectrum	[54]
Alpha particles (5.50 MeV, 0.223 Gy/min): 0.83 Gy; gamma rays (662 keV, 0.372 Gy/min): 1.02 Gy; total dose of 1.85 Gy; 5 min transfer time between irradiations	U2OS human osteosarcoma cells stably expressing NBS1-GFP	DNA repair foci frequency, size, intensity, and mobility (NBS1-GFP live-cell imaging)	• Stronger synergistic effect observed for α→γ sequence • Slower repair kinetics, larger and more persistent foci • ↑ Intensity and ↓ mobility • γ→α induced faster decay and lower focus intensity • Results suggest order-dependent DDR engagement and impaired repair after alpha priming	[55]
Alpha particles (LET ~91 keV/μm): 2.5 Gy; gamma rays (0.73 Gy/min): 2.5 Gy; fractionated regimens used as well	Breast cancer (MDA-MB-231), Osteosarcoma (U2OS)	γH2AX foci (TEM, immunofluorescence), colony formation, viability	• Mixed beam causes more γH2AX foci • Stronger reduction in viability/colony formation • Delayed chromatin recompaction enhances cell kill	[56]
Alpha (LET ~126 keV/μm, 50 rad/min): ~25% of total dose; gamma (LET ~0.31 keV/μm, 154 rad/min): ~75% of total dose	Diploid yeast (*S. cerevisiae*, strain BZ34)	Mutation frequency (reversion to arginine independence)	• Statistically significant synergistic effect observed • Reversion frequency 1.34× higher than additive prediction • Enhanced mutagenic effect attributed to interaction between low- and high-LET damage pathways	[57]
Alpha, beta, and gamma radiation (mixed radionuclides from Chernobyl fallout including ^134^Cs, ^137^Cs, ^144^Ce, ^154^Eu, etc.): total dose 1–515 mSv (chronic); gamma rays (^60^Co source): 0.1–29, 600 mSv (chronic)	Barley (Hordeum vulgare, waxy mutant line); field-grown in contaminated plots and gamma-field controls	Waxy-reversion frequency in haploid pollen; mutation frequency in generative cells	• Combined radionuclide IR caused higher mutation rates per mSv than external gamma • Mutagenicity not explained by dose alone • Enhanced genotoxicity linked to multi-type exposure, chemical synergies, and heterogeneous contamination	[58]
Beta (^90^Sr-^90^Y, low LET): 1.2–4.8 krad; gamma rays (^60^Co, low LET): 1.2–4.8 krad; beta and gamma combined (varied proportions): total 1.2–4.8 krad at 8.4 or 17.8 rad/min	Soybean plants (Glycine max cv. Hill) at unifoliolate leaf stage; grown to maturity in field	Survival, plant height, lateral growth frequency and length, vegetative yield, seed yield	• Combined exposure affected lateral growth and yield depending on beta/gamma dose component • Gamma slightly more damaging overall • Interaction effects seen in vegetative vs. reproductive response • Dose-rate and composition sensitivity evident	[59]
Alpha particles (5.50 MeV): ~0.9–4.9 kBq/g lung; beta particles (0.062 MeV): ~0.4–2.2 MBq/g lung	F344/Crl rats (inhalation, in vivo)	Radiation pneumonitis mortality rate; respiratory function (vital capacity, compliance, CO diffusion)	• Combined exposure produced additive effects • Validated hazard models for lethality and morbidity • Alpha radiation ~7× more biologically effective than beta • Respiratory dysfunction and fibrosis observed	[60]
X-rays (250 kVp): 3 Gy; neutrons (IND-spectrum, high-LET): 0.15–0.75 Gy (5–25% of total 3 Gy dose); neutrons (83%) + gamma (17%): total 0.75 Gy	Human peripheral blood (ex vivo from 5 donors)	Gene expression profiling (microarray, RT-qPCR); TP53 signaling; immune suppression	• Neutron–photon mixtures caused increasing transcriptomic alterations with higher neutron % • 25% neutron had strongest TP53 and immune effects • 22 genes uniquely responded to neutron-containing exposures, suggesting gene-level synergism even without simultaneous irradiation	[61]
X-rays (250 kVp): 3 Gy; neutrons (IND-spectrum, broad energy range simulating Hiroshima 1–1.5 km epicenter): 0.75 Gy or contributing 5% (0.15 Gy), 15% (0.45 Gy), or 25% (0.75 Gy) of 3 Gy total mixed dose	C57BL/6 mice (peripheral blood, 7 days post-exposure)	Gene expression (microarray, RT-qPCR); repression of translation and ribosomal gene sets	• Strong synergistic effect at the gene expression level • Mixed exposures→unique suppression of mRNA translation, tRNA processing, EIF2/mTOR signaling, and ribosomal protein genes linked to bone marrow failure • Effects not seen in pure X-ray group • Synergy evident even at 5% neutron	[62]
X-rays (250 kVp): 1 or 3 Gy; monoenergetic neutrons (0.35, 0.45, 5.9, 13.7 MeV): 0.1 or 0.3 Gy; sequential exposures with <2 min delay (neutrons always first)	C3H 10T½ mouse fibroblast cells	Oncogenic transformation (type II/III foci); clonogenic survival	• No significant synergistic effect • Transformation frequencies matched additive prediction across all neutron energies • Low-dose combined exposures showed additive behavior even with high-LET neutrons	[63]
X-rays (140 kVp): variable dose per fraction (up to ~30 Gy total); neutrons (3 MeV): variable dose per fraction (up to ~25 Gy total); mixed fields included combinations with photon contamination levels of 11%, 32%, 53%, and 72%	Mouse foot skin (WHT/Gy fBSVS mice)	Acute skin reaction scoring	• Strong synergistic interaction during simultaneous exposure • Full quadratic-term interaction confirmed • Synergy declined but persisted up to 6 h delay • Dose-response curves shifted with increasing photon content, supporting sublethal damage interaction model	[64]
X-rays (180 kVp): up to ~635 rad; neutrons (14 MeV): up to ~430 rad; various single and sequential doses used; exposures spaced by 5–10 min	L5 mouse fibroblast cell line (subclone of L cells); suspension culture in Eagle MEM	Cell reproductive capacity (colony formation > 50 cells)	• Survival lower in sequentially irradiated cells vs. single modality • Combined exposures showed interaction • RBE values for neutrons 2.5–1.5 vs. X-rays • Evidence of partial overlap but mechanistic differences between neutron and X-ray cell killing	[65]
X-rays (250 kVp): 0.2 Gy priming and 1 Gy challenge; neutrons (LET 60–70 keV/μm): 0.2 Gy priming and 1 Gy X-ray challenge; Bragg peak negative pi mesons (LET 35–55 keV/μm): 0.2 Gy priming and 1 Gy X-ray challenge	V79-379A Chinese hamster fibroblasts	Colony-forming ability (>50 cells); plating efficiency; survival after challenge; induction of adaptive response	• Sequential high-LET priming (neutrons/pions) ↑ resistance to subsequent X-rays if 4–6 h elapsed • Adaptation required minimum 0.2 Gy priming • Effect was transient, protein-synthesis-dependent, and indicative of inducible DNA repair mechanisms	[66]
X-rays (250 kVp, 3 Gy/min): up to 8.5 Gy; fast neutrons (3.15 MeV, 0.47 Gy/min; 11.3% gamma contamination): up to 3.75 Gy; sequential exposures with 6 min delay (room temp), or 3 h delay (with recovery at 37 °C)	V79 Chinese hamster fibroblasts (in suspension)	Clonogenic survival	• Synergistic interaction observed with 6 min delay • ↓ X-ray survival curve shoulder (Dq), especially after neutron priming • No interaction with 3 h delay • Effects consistent with partial inhibition of repair between exposures	[67]
X-rays (250 kVp, 150 rad/min): doses combined with neutrons in split exposures; fast neutrons (25 MeV, Fermilab): 280 or 420 rad; fast neutrons (0.86 MeV, JANUS): up to ~1.6 krad	V79 Chinese hamster cells	Colony-forming ability	• Increased survival after neutron + X-ray fractionation • Εvidence of repair of sublethal damage • Sequential exposure (neutron followed by X-ray) enhanced sublethal damage repair vs. neutron alone	[68]
X-rays (250 kVp): 1250 rad; fast neutrons (produced by 16 MeV deuterons on Be target): 540 rad; sequential irradiation with 15 min to 4 h interval	Mouse jejunum (crypt cells)	Crypt survival	• Combined X-rays and neutrons caused additive or interacting effects depending on time between doses • Interaction seen if <4 hrs between • Sublethal damage from neutrons and X-rays showed similar repair kinetics • Full repair→additive effects • Short interval→partial repair and enhanced damage	[69]
Neutron beam (HB11, mixed field of protons from neutron capture and induced/incident gamma): 0.25–1.7 Gy; gamma rays (^60^Co source, used for calibration): 0.25–3 Gy; exposures in water phantom at 37 °C	Human peripheral blood lymphocytes (PBLs) from 6 donors	Chromosomal aberrations (dicentric chromosomes)	• No synergistic effect observed • Mixed beam-induced dicentric yields matched additive predictions from gamma + calculated proton components • RBE values: 3.0 (mixed beam), 7.2 (protons) • Results consistent across varying neutron/gamma ratios	[70]
Neutrons (0.1–8 MeV): 0.02 Gy/min; gamma rays: 0.1 Gy/min; total dose: 0.5, 0.75, or 1.0 Gy; mixed beams 1:1 neutron/gamma; sequential exposure (within 8 min) in both orders (γ→n and n→γ)	Human peripheral blood mononuclear cells (PBMCs) from 3 donors	Gene expression (RT-qPCR of *FDXR*, *BBC3*, etc.); chromosomal aberrations (dicentric chromosome assay, DCA)	• No synergistic effect detected for either endpoint • Mixed-beam responses were additive regardless of sequence • RBEs for neutrons ranged from 1.39 to 3.91 (gene expression) and 7.30 (dicentrics) • Gamma–neutron order had no significant influence	[71]
Νeutrons (2.5 MeV): 1.42 Gy; gamma rays (^137^Cs source): 1.42 Gy; combined neutron + gamma: 0.71 Gy each; 252Cf neutrons: 0–0.71 Gy	I: Human peripheral blood from 3 healthy adult males; II: AHH-1 human lymphocytes	Gene expression profiling (RNA-seq, qPCR); dose–response of *BAX*, *DDB2*, *FDXR* (qPCR)	• Neutrons induced more differentially expressed genes and pathways than gamma rays • Combined exposure activated genes overlapping with both mono-exposures • *BAX*, *DDB2*, *FDXR* showed neutron-specific dose-responses, suggesting their role as molecular targets of neutron damage	[72]
Fast neutrons (~6 MeV, 2–3% photon contamination): up to 7.5 Gy; gamma rays (^60^Co source): up to 7.5 Gy; mixed exposures with 60% gamma + 40% neutrons and vice versa, delivered sequentially with <3 min interval	Chinese hamster V79 cells	Clonogenic survival (colony formation assay)	• Survival after mixed exposure was independent of sequence • Synergistic effects reflected in altered survival curve slopes • Zaider–Rossi model accurately described both sequential and simultaneous outcomes	[73]
Neutrons (0.5 eV–10 MeV); gamma rays (0.1–4 MeV); IR-8 nuclear reactor mixed field; simultaneous exposure at total doses from 25 mGy to 2 Gy; compared to gamma-only irradiation in the same dose range.	Neonatal mouse neural stem/progenitor cells (NSCs/NPCs) cultured in vitro	DNA double-strand breaks (γH2AX foci), focus size and repair kinetics, neurosphere formation, cell survival	• Low doses (25–50 mGy) stimulated proliferation • >100 mGy reduced survival • Gamma–neutron IR had higher RBE (max 9.7) vs. gamma alone • Induced large γH2AX foci with slow repair • Sensitivity of NSCs/NPCs to mixed radiation highlighted	[74]
Mixed field of fission neutrons and gamma rays (MUTR reactor); neutron dose rate: 0.32 Gy/min; gamma dose rate: 0.42 Gy/min; gamma rays (^60^Co control): 0.15–1.3 Gy/min; total doses up to ~7 Gy	V79-4 Chinese hamster lung fibroblast cells (monolayer culture)	Clonogenic survival	• Dose-dependent loss of clonogenic survival • Mixed neutron–gamma field caused increased cell killing via interaction effects • Nanodosimetry model suggested localized DNA damage at nanoscales • Steeper survival decline than with gamma rays alone • Clonogenic assay confirmed greater loss of reproductive capacity	[75]
Neutrons (14.7 MeV): 10–15 rad/min (5 ± 2% gamma contamination); gamma rays (^60^Co): 150 rad/min; combined doses with 50% or 75% gamma ray contribution; sequential exposures with immediate succession or extended up to 45 min	Chinese hamster ovary cells (in flasks, room temperature)	Clonogenic survival	• Significant synergistic effect observed at both 50% and 75% gamma ray mix • Survival lower than additive prediction • Minimal impact of sequence or delay up to 45 min • Katz model predictions matched experimental interaction magnitude	[76]
Neutrons (14.8 MeV, 4.8 rad/min, 40% of total dose); gamma rays (^60^Co, 8 rad/min, 60% of total dose); total dose varied across experiments; exposures conducted both simultaneously and sequentially (with ~5 min interval between modalities for sequential exposure)	V79 Chinese hamster lung fibroblast cells (monolayers at 37 °C)	Clonogenic survival	• Simultaneous exposure produced a supra-additive effect • 29% lower survival at 10% level vs. additive prediction • Sequential exposures (either order) matched additive model • Results indicate dose-rate sensitive synergism when beams are mixed	[77]
Neutrons (14.8-MeV, 2 cGy/min, 40% of the beam); gamma rays (^60^Co, 3 cGy/min, 60% of the beam); delivered as pulsed beams in 3 min intervals; total dose varied; exposures given either sequentially (alternating pulses) or simultaneously (combined pulses)	V79 Chinese hamster lung fibroblasts (monolayers at 37 °C)	Clonogenic survival	• Simultaneous exposures caused significantly more cell killing than sequential • Dose ratio at 1% survival = 1.08 • Survival curves fitted better with quadratic models • Confirms supra-additive effect under simultaneous delivery • Supports interaction between damage types	[78]
Mixed gamma–neutron radiation (TRIGA Mark-F reactor; ~60% gamma, ~40% neutron): 570–30,000 rads; delivered in steady-state (1000 rad/min) or pulsed mode (88% of dose in <40 ms)	Male Sprague–Dawley rats; serum and urine sampled 6–72 h post-exposure	Fluorescence intensity in serum (360, 465 nm) and urine (400, 425 nm)	• Serum fluorescence at 465 nm ↓ dose-dependently at 24 h (570–9300 rads) • 360 nm peak ↑ at 72 h (non-dose-dependent) • Urine fluorescence at 425 nm ↑ with dose (1000–30,000 rads) • Limited utility as biological dosimeter	[79]
Mixed neutron–gamma radiation (light water-moderated research reactor; neutron peak energy 0.4–0.6 MeV; ~30% gamma component)l total dose of 0.5, 2, or 4.5 Gy; whole-body exposure in rotating cage	Male F1 (C57BL × DBA2) mice; erythrocytes, lymphocytes, platelets isolated from whole blood	Changes in membrane lectin-binding capacity; alterations in cell surface morphology; intracellular membrane structural changes	• ↑ Lectin-binding observed as early as 30 min • Lymphocytes most sensitive (up to 2.5× increase) • Dose- and time-dependent oscillatory membrane responses • Membrane ultrastructure altered (filopodia, vacuolization, ER dilation) • Effects not dose-proportional above 0.5 Gy	[80]
Gamma rays (^60^Co, ~2.0–2.5 MeV): 0.1 Gy; mixed neutron (0.5–10 MeV) and gamma radiation (2.0–2.5 MeV): 1 Gy	C57BL/6J male mice; hippocampus and brain immune cells analyzed	Behavioral performance (open-field test, Morris water maze, MWM); microglial and astrocyte populations (flow cytometry); cytokine and neurotrophin levels (qPCR, ELISA)	• Combined irradiation impaired hippocampus-dependent memory (unlike gamma, neutron alone) • ↑ M2 microglia, astrocytes, BDNF and NT-4 expression • ↓ TNF-α and IL-1β vs. gamma, neutron • Suggests anti-inflammatory neuroadaptation post gamma-priming	[81]
Mixed neutron–gamma radiation (neutrons: 0.5–3.0 MeV, ~85% of dose; gamma rays: <10% of dose): 375 rads or 1000 rads	ICR female mice (germfree and conventional); small intestine	Mucosal atrophy, crypt regeneration, occurrence of diarrhea, survival time	• At 375 rads, mucosal recovery occurred in both groups before death • At 1000 rads, lesion progression and death occurred earlier in conventional mice • Villus atrophy, lipid-filled cells, and delayed regeneration observed in germfree mice • Epithelial denudation not seen	[82]
Neutrons (43 MeV): 4–8.5 Gy; gamma rays (^60^Co): up to 20 Gy; delivered sequentially with short intervals (0–3 h) between modalities	V79 Chinese hamster cells in plateau phase (G_1_ arrest)	Clonogenic survival (colony formation assay)	• Interaction effect shown as significant reduction in or elimination of survival curve shoulder • Damage from both modalities appears to involve similar PLD • Response mimics that seen with β-araA treatment	[83]
Epithermal neutron source (Studsvik: includes fast neutrons > 1 MeV and γ-rays): total dose 8.2–16.2 Gy/h, depth-dependent; epithermal neutron source (Birmingham: neutrons ≤ 1 MeV and γ-rays): total dose 0.58–1.04 Gy/h, depth-dependent	Chinese hamster V79 fibroblast cells	Clonogenic survival	• ↓ Clonogenic survival with depth • ↑ RBE values (Studsvik > Birmingham) • Evidence of high-LET and low-LET interaction enhancing biological damage	[84]
Neutrons (0.2–9 MeV, 0.96 Gy/h, ~17% gamma ray component): 0.15–0.75 Gy; gamma rays (0.17 Gy/h): 0.03–0.15 Gy; X-rays (1.23 Gy/min): 2.1–2.82 Gy; total dose of 3 Gy; equitoxic 0.9 Gy (0.75 Gy neutrons + 0.15 Gy gamma rays)	Mouse (C57BL/6J) serum	Changes in serum lipid classes (LPC, PC, TG, DG, PS, CE, SM, LPE)	• Synergistic pro-inflammatory and hyperlipidemic lipidomic changes • ↑ LPC/PC ratio (inflammatory biomarker) • ↑ TG and PS	[85]
Fast neutrons (14.5 ± 1.04 MeV, high LET): 0.7 Gy (2 Gy EQD); protons (67–83 MeV, spread-out Bragg peak): 2 Gy (2 Gy EQD); combined sequential exposure (neutron→proton, 2 h interval): 4 Gy EQD	Human breast cancer cell lines (MCF-7, MDA-MB-231)	Cancer stem cell fraction (ALDH+/CD44+/CD24−); stemness gene expression (*OCT4*, *NANOG*, *SOX2*)	• Combined exposure decreased CSC fraction additively in MCF-7 and antagonistically in MDA-MB-231 • No significant changes in stemness gene expression • Response depended on cell line and radiation sequence	[86]
Neon ions (425 MeV/amu, LET ~234 keV/μm, 500–600 rad/min): up to 6 Gy; X-rays (225 kVp, 270 rad/min): up to 8 Gy; sequential exposures with 0–24 h interval	V79 Chinese hamster lung fibroblasts (asynchronous monolayer)	Clonogenic survival	• Synergistic reduction in survival with both sequences • Strongest effect with minimal delay • High-LET priming eliminated shoulder of X-ray curve • Low-LET priming steepened neon ion curve • Synergy diminished with 3–24 h interval, indicating repairable, interacting damage types	[87]
Neon ions (425 MeV/u, LET ~234 keV/μm): 3.3 Gy; X-rays (225 kVp): 5.5 Gy; Argon ions (570 MeV/u, LET ~117 keV/μm): 2.04 or 3.57 Gy	Chinese hamster V79 fibroblasts (synchronized at GI/S, mid-S, and late-S phases)	Clonogenic survival	• Strong synergistic effect, greatest in late S-phase • Interaction diminished with ≥3 h delay • Survival response depended on cell cycle stage and priming radiation type • Supports repairable, phase-specific sublethal damage interaction	[88]
Deuterons (50 keV/μm): 2 or 5.6 Gy; ^3^He ions (96 keV/μm): 2.5 or 4 Gy; ^3^He ions (160 keV/μm): 4 Gy; X-rays (50 kVp): graded doses following each high-LET dose	Chinese hamster V79 cells (synchronized in late S-phase)	Clonogenic survival	• Strong synergistic effects observed across all LETs and doses •↑ ER with LET and priming dose • Mixed irradiation decreased survival more than additive prediction • Results consistent with interaction of sublethal damage from high- and low-LET radiation	[89]
Priming dose: 7 Gy neon ions (557 MeV/u, LET 115–240 keV/μm) or 20 Gy X-rays (225 kVp, 6 Gy/min); top-off X-ray doses: 7.5, 15, or 25 Gy given at 0.5, 4, or 24 h later	Rat rhabdomyosarcoma tumors (R2C5 subline in WAG/Rij rats)	Tumor growth delay (doubling time to 2× volume)	• No significant synergistic effect • Growth delays similar regardless of whether X-rays followed neon ions or X-rays • Top-off doses produced near-additive outcomes even at 0.5 h interval • Rapid sublethal damage repair likely prevented interaction	[90]

* ↑ indicates all language forms related to increase; ↓ indicates all language forms related to decrease.

**Table 4 biomolecules-15-01282-t004:** Therapeutic studies involving combined radiation.

Type and Dose	Biological Systems	Biological Endpoints	Effects	Ref.
X-rays (6–10 MV): 50.4 Gy in 28 fractions; proton beams (250 MeV): 46.2 GyE in 28 fractions, delivered as concomitant boost > 6 h after X-rays	Supratentorial glioblastoma multiforme patients (n = 20) [Clinical study]	Overall survival, progression-free survival, toxicity	• Median survival 21.6 months; 1- and 2-year survival rates 71.1% and 45.3% • Manageable acute hematologic toxicity • Occasional late leukoencephalopathy	[91]
Photons (^60^Co gamma rays or 6 MV X-rays): 50.4 Gy in 28 fractions; proton beams (250 MeV): 25.5 GyE in 17 fractions as concomitant boost (minimum 6 h interval); total 75.9 GyE in 45 fractions over 5.5 weeks	Stage II–IV oropharyngeal squamous cell carcinoma patients (n = 29) [Clinical study]	Locoregional control, disease-free survival, acute and late toxicity	• 5-year locoregional control 84% • Pronounced acute mucosal toxicity (mucositis, dysphagia) • Grade 3 late effects in 11% (fibrosis, trismus, vocal cord paralysis)	[92]
X-rays (4–10 MV): ~50–55 Gy; proton beams (160 MeV): 16–28 GyE, combined sequentially in daily fractions; total prescribed dose 66–83 CGE	Patients with skull base and cervical spine chordomas and chondrosarcomas (n = 621) [Clinical study]	Local control, overall survival, normal tissue toxicity	• 10-year local control: chondrosarcoma 94%, chordoma 54% • Male chordoma patients had better outcomes • Main late toxicities included temporal lobe injury (13%), optic neuropathy (4.4%), endocrinopathy (40%)	[93]
Protons (95–105 MeV); carbon ions (^6^C, 400 MeV/n); total RBE-weighted dose ≅ 8.4 ± 0.2 Gy; four sequential exposure schemes with 30–45% ^12^C contribution; intervals: 0–4 h; sequence: p→C or C→p	Chinese hamster fibrosarcoma cells (B14-150) [In vitro study]	Clonogenic survival	• Significant synergistic effect observed only in C→p sequence with 45% ^12^C contribution (K = 0.65) • Effect diminished with 30% ^12^C • Antagonism seen in p→C sequence (K > 1) • ↑* Survival with longer interval in p→C, but ↓* in C→p	[94]
Proton beam (OPTIS2; Bragg peak; LET ≅ 0.5–2 keV/μm): 7.5 Gy or 5 Gy; Targeted radionuclide therapy with ^177^Lu-Folate or ^177^Lu-PSMA-617 (β^−^, Eₘₐₓ ≅ 0.5 MeV; LET ≅ 0.2 keV/μm): ~7.5 Gy (8.5 MBq) or ~5 Gy (1.25 MBq)	CD1 nude mice with KB xenografts; BALB/c nude mice with PC-3 PIP xenografts [Preclinical study]	Tumor growth delay (TGDI_2_/_5_), median survival, relative tumor volume	• Combination therapy showed additive or synergistic effect depending on tumor model • KB model showed significant synergy (↑ TGDI_2_/_5_, no endpoints) • PC-3 PIP model showed additive trend only • Combination well tolerated in both cases	[95]
X-rays (160 kV): 2 Gy; ^177^Lu-PSMA-617: ~40 MBq (400 pmol), administered 4 h after EBRT (preclinical part of the study)	LNCaP xenografts in BALB/c nu/nu mice [Preclinical study]	Tumor growth delay, median survival (in mice, preclinical)	• Combined external beam RT + RLT in mice prolonged tumor doubling time 2.7-fold vs. RT alone • Median survival extended from 22.5 days (untreated) to 44 days	[96]
Protons (67–83 MeV, LET ~ 3 keV/µm); heavy recoils (HR, induced by 14.5 MeV neutrons, LET ~ 290 keV/µm); total dose: 6.6–6.8 Gy (RBE); varying p/HR ratios (60/40%, 80/20%); intervals: 0–8 h	B14-150 Chinese hamster fibrosarcoma cells (confluent monolayers) [In vitro study]	Cell survival (clonogenic assay)	• HR→p sequence with 40% HR most effective for reducing survival • p→HR sequence showed partial recovery (T½ ~1.1–1.3 h) • Combinations showed mostly antagonistic interaction (K > 1) • Survival inversely related to HR dose contribution	[97]
Proton pencil beam scanning (96–104 MeV; LET ≅ 0.5–1 keV/μm): 40 Gy × 2 (total 80 Gy); neutron radiation (14.1 MeV; high-LET ≅ 100 keV/μm): 5 Gy; sequential exposure in mice with 3 h interval: neutrons before or after protons; neutron dose ~15% of total; CT-guided tumor targeting	SHK mice with solid Ehrlich ascites carcinoma [Preclinical study]	Tumor growth suppression, skin radiation reactions (RTOG/EORTC), relapse frequency, remission duration, survival	• All groups showed tumor suppression • Neutron-after-proton group showed milder skin toxicity and better tolerance • Neutron-before-proton group showed severe toxicity and shortest survival • Combined exposures ↑ relapse rate and ↓ lifespan vs. protons alone	[98]
Protons (88–109 MeV); neutrons (14.5 MeV, D-T generator); total dose ~8.6 Gy; neutron/proton dose ratios of 30:70 or 40:60; sequential exposures with 0–8 h delay; survival modeled vs. independent action	Chinese hamster fibrosarcoma cells (B14-150) [In vitro study]	Clonogenic survival (colony assay)	• All neutron–proton schemes showed synergistic effects (K < 1) • Strongest synergy when neutrons delivered first and comprised 40% of dose • Survival significantly below additive prediction • No recovery observed in neutron-first sequence	[99]
Priming dose: 0.075 Gy X-rays; challenging dose: 1.75 Gy ^137^Cs gamma rays; 6 h interval between doses; exposures in mice and ex vivo human thymocytes	Mouse thymocytes (C57BL/6J); human pediatric thymocytes (1 mo–3 yrs) [In vitro study]	Cell cycle (sub-G_1_), DNA damage (γH2AX), apoptosis (Caspase-3, PARP1), ferroptosis (xCT, GPX4), epigenetic markers (DNMTs, TDG, MBD4)	• Strong synergistic response: earlier and enhanced apoptosis, cell cycle arrest, ferroptosis, and DNA damage response in combined vs. single dose • Priming dose→sensitized “radiation awareness” state • Consistent mouse–human similarity in response	[100]
X-rays (320 kVp, whole thorax): 12.5 or 13 Gy; soft X-rays (10 kVp, dorsal skin, 10% surface): 30 Gy	WAG/RijCmcr rats; lung and skin monitored for 210 days post-irradiation [Preclinical study]	Survival (IACUC criteria), breathing interval, lung collagen (fibrosis), mast cell count, skin wound area	• Combined lung + skin irradiation delayed pneumonitis onset and improved survival vs. thorax alone (13 Gy) • Lung collagen ↓ by skin co-irradiation • Captopril enhanced skin healing and delayed lung injury further	[101]
X-rays (150 kV, orthovoltage): 9 Gy; photodynamic therapy (THPTS, 760 nm): 20 J/cm^2^	Bladder cancer organoids (T-24, RT-112) [In vitro study]	Organoid viability, multimodal cell death (apoptosis, ferroptosis, pyroptosis), quantified immune infiltration (e.g., Jurkat migration assay)	• IR + PDT showed additive/synergistic cytotoxicity • Multimodal cell death • Non-malignant tissue unaffected • ↑ T-cell infiltration	[102]
PDT (non-coherent light, 370–680 nm): 30 mW/cm^2^, 90–180 s; RT (gamma rays, ^60^Co source, 1.0–1.62 Gy/min): 2 Gy (range tested: 0–15 Gy)	Ehrlich ascites carcinoma cells in BALB/c mice [Preclinical study]	Tumor growth inhibition, plasma membrane damage (Trypan blue assay), DNA damage (chromosomal aberrations)	• PDT damaged membranes • RT caused DNA breaks • Combo had additive tumor inhibition (~33–38%) • HPde acted as dual sensitizer	[103]
X-rays: 2, 10, 20 Gy; PDT: 2.5 J/cm^2^ at 690 nm; PDT given ~10 min before RT	Heterocellular pancreatic cancer spheroids (MIA PaCa-2, Capan2, AsPC-1) co-cultured with patient-derived fibroblasts [In vitro study]	Viability (live/dead staining), necrosis, apoptosis (flow cytometry), DNA damage (γ-H2AX), proliferation (PCNA)	• Low-dose PDT and RT showed synergistic effects • PDT→necrosis and apoptosis • RT decreased spheroid growth • Combination→smaller, less viable spheroids than expected additively • Effects varied by cell line	[104]
X-rays (50 kV): 0–10 Gy; nanoparticles: MC540-SAO:Eu@mSiO_2_ at 50 µg/mL; X-PDT combines RT and nanoparticle-mediated PDT; applied 5 min after injection	Radioresistant human NSCLC cells (H1299); subcutaneous and intrathoracic mouse tumor models [In vitro and preclinical study]	Cell viability (MTT), clonogenic survival, apoptosis/necrosis, DNA damage (comet, γ-H2AX), lipid peroxidation (ROS), tumor growth delay	• Strong synergistic effect: X-PDT significantly more effective than RT alone in vitro and in vivo • Enhanced apoptosis, necrosis, DNA, and lipid damage • Tumor suppression in deep tissues • No systemic toxicity observed	[105]
PDT (non-coherent light, 730 nm): 45 mW/cm^2^, 30–108 J/cm^2^; indocyanine green (ICG) 50 μM; RT (X-rays, 100 kVp): 2–8 Gy; combination: 4 Gy X-rays + 60 J/cm^2^ light + 50 μM ICG	MCF-7 breast cancer cells [In vitro study]	Cell viability (MTT assay)	• ICG alone non-toxic but effective photosensitizer • Combo of ICG + light + X-ray killed 96.6% cells • Low-dose X-ray enhances PDT efficacy	[106]
RT (320 kV X-rays): 4 or 8 Gy; UV-C (200–280 nm)-emitting nanoscintillators (LuPO_4_: Pr^3^⁺, Nd^3^⁺): 2.5 mg/mL	A549 lung cancer 3D spheroid model [In vitro study]	Tumor spheroid growth; cell death pathways (apoptosis, necrosis); cell cycle arrest (G2/M)	• ↑ Tumor growth inhibition (up to 30% size reduction with uniform NP distribution) • ↑ Apoptosis and necrosis vs. radiation alone • ↑ Permanent G2/M cell cycle arrest • Effect present under hypoxia but reduced vs. normoxia • No nonspecific toxicity from nanoparticles alone	[107]
RT (X-rays, 320 kVp): 2–4 Gy; UVC (220–285 nm, via LuPO_4_:Pr^3^⁺ NPs, generated in situ by X-rays; NP concentration: 0.5–7.5 mg/mL (optimal: 2.5 mg/mL)	HFF1 normal human fibroblasts, XP17BE UV-sensitive fibroblasts [In vitro study]	Clonogenic survival, CPD formation (ELISA)	• ↑ Cell killing with combined treatment (↓ survival to ~2% at 7.5 mg/mL NPs + 2 Gy) • ↑ CPD formation (~50% equivalent to 15 J/m^2^ UV-C) • ↑ Effect in XP17BE vs. HFF1 cells • NPs alone cause mild dose-dependent toxicity at ≥5 mg/mL	[108]
RT (X-rays (6 MV): 2 Gy; NIR laser (808 nm): 2 W/cm^2^ for 3–20 min at 43 °C	Human glioblastoma U87MG cells [In vitro study]	Colony formation; cell viability	• ↓ Plating efficiency with IUdR-PLGA-NGO + X-ray + NIR vs. all other groups • ↑ Nanoparticle uptake • Enhanced radio- and thermo-sensitization • No cytotoxicity from laser or nanoparticles alone	[109]
Pelvic external beam RT (6 or 15 MV photons): 45 Gy in 25 fractions; HDR brachytherapy (^192^Ir): 4 fractions of 7–7.5 Gy each during RT (some fractions after RT)	Cervical cancer patients (n = 13, FIGO stages IB–IIIB) [Clinical study]	Tumor volume kinetics (MRI), gross tumor volume (GTV) and high-risk clinical target volume (HR-CTV) reduction	• Tumor volume reduction ~70–75% by first brachytherapy fraction • Further modest shrinkage thereafter • Early initiation of brachytherapy feasible during EBRT	[110]
Modulated electrons (9–15 MeV); IMRT (6 MV photons); total dose 38.5 Gy in 10 fractions over 5 days; combined delivery per fraction	Cohort of 7 breast cancer patients with early-stage disease treated post-lumpectomy [Clinical study]	Acute skin toxicity (CTCAE)	• Combined MERT + IMRT achieved comparable target coverage to IMRT alone but reduced dose to ipsilateral lung and heart • No grade ≥ 2 toxicity • Cosmetic results rated as excellent/good in most cases	[111]
RT (X-rays): 6 Gy; NIR light (730 nm): 0.4–0.8 W/cm^2^ for 5–8 min; UCNP@NBOF-FePc-PFA at 80–100 μg/mL	Murine breast cancer cells (4T1.2), U251 glioma cells; BALB/c mice (tumor-bearing) [In vitro and preclinical study]	Tumor cell apoptosis, ROS generation	• Highly synergistic tri-modal effect (radiotherapy + photothermal + photodynamic therapy) • ~96% tumor inhibition in vivo • Massive cell apoptosis • ↑ ROS and temperature under dual irradiation	[112]
X-rays (6 MV): 2–6 Gy; diode laser (808 nm): 1.0–1.5 W/cm^2^ for 10 min; applied sequentially ± Pt nanoparticles (100 μg/mL)	B16/F10 melanoma cells [In vitro study]	Cell viability (MTT assay), intracellular ROS production	• Combined X-ray + laser irradiation with PtNPs significantly decreased viability (~80–90% reduction) vs. either modality alone • Synergistic ROS generation observed	[113]
X-rays (6 MV): 4 Gy; NIR laser (808 nm): 1 W/cm^2^ for 3 min at ~ 42 °C; RT applied 4–6 h after the first laser irradiation	Mouse 4T1 TNBC tumor model (BALB/c nude mice); 4T1 cells in vitro [In vitro and preclinical study]	ROS generation; tumor growth delay; colony formation; apoptosis (TUNEL)	• Strong synergistic effect: 60% complete tumor eradication in vivo • Highest ROS production in combined group • Lowest colony survival and strongest apoptosis in vitro • INS NPs showed excellent tumor targeting and magnetic guidance	[114]
NIR-PIT (690 nm laser): external exposure (50 + 100 J/cm^2^); interstitial exposure (50 + 100 J/cm); combined exposure (25 + 50 J/cm^2^ external + 25 + 50 J/cm interstitial)	EGFR-positive A431-luc tumor xenografts in nude mice [Preclinical study]	Tumor volume, bioluminescence (viability), survival	• Combined external/interstitial light led to greatest tumor volume reduction and survival vs. either alone • Improved light delivery and tumor coverage enhanced treatment efficacy	[115]
Carbon ions (320 MeV/n, LET 46.6 keV/μm): 0.8–4.4 Gy; X-rays (4 MV): 2–8 Gy; carbon 0.4–2.2 Gy + X-ray 1–4 Gy; exposures within 15 min or 72 h apart	Human salivary gland cancer cells (HSG) [In vitro study]	Clonogenic survival (colony formation assay)	• No synergistic effect observed • Combined exposures followed additive prediction model based on GyE • Survival curves and parameters aligned with additive model • Effect independent of irradiation sequence	[116]
Carbon ions (290 MeV/u, LET 13–100 keV/μm): 2.0–6.8 Gy; silicon ions (490 MeV/u, LET 55 keV/μm): 3.0 Gy; argon ions (500 MeV/u, LET 85 keV/μm): 2.5–3.0 Gy; iron ions (500 MeV/u and 200 MeV/u, LET 200–860 keV/μm): 1.75–3.5 Gy; X-rays (150 or 200 kVp): 8.0 Gy (priming or test dose)	V79 Chinese hamster cells [In vitro study]	Clonogenic survival	• ↑ Cell killing with sequential ion + X-ray exposure vs. single beams • ↓ SLDR with increasing LET; high-LET ions (≥80 keV/μm) cause largely irreparable damage • LET-dependent reduction in repairable fraction • Evidence of combinatorial suppression of SLDR	[117]
Whole-brain photons (^60^Co source): 45 Gy (1.5 Gy/fraction); neutrons (deuteron–deuterium source, RBE~4.5): 5.2 Gy boost; neutrons given 5–20 min before photons, over 6 weeks	Patients with anaplastic astrocytoma and glioblastoma multiforme (n = 44) [Clinical study]	Tumor control, median survival, histologic response	• Anaplastic astrocytoma (AA) median survival 40.3 months vs. 11 months for glioblastoma multiforme (GBM) • Neutron–photon sequencing within minutes associated with improved survival in AA • High rates of tumor necrosis observed	[118]
Whole-brain photons (1.5 Gy/fraction, total 45 Gy); neutron boosts (3.6–6.0 Gy total, 12 fractions twice weekly, given within 3 h of photons	Supratentorial glioblastoma multiforme and anaplastic astrocytoma patients (n = 190) [Clinical study]	Overall survival, radiation injury (autopsy pathology)	• Median survival 9.9 months (glioblastoma) and 22 months (anaplastic astrocytoma) • Higher neutron doses showed trend to worse survival in astrocytoma • Autopsies revealed frequent tumor sterilization but extensive radiation injury	[119]
Fast neutrons (30 MeV d-Be, <5% gamma contribution): up to 35.78 Gy (fractionated, 5 n); gamma rays (^60^Co, 1.00 Gy/min): up to 104.4 Gy (fractionated, 5 gamma); mixed beam (2n + 3 gamma over 5 days): neutrons ~6.78–7.66 Gy, gamma ~66 Gy total	C3H mice bearing syngeneic NFSa fibrosarcoma [Preclinical study]	Cell survival (lung colony assay)	• Mixed-beam (N-γ-γ-γ-N) yielded survival and TCD_50_ curves indistinguishable from calculated additive effects • No interaction observed • Neutron RBE ~3 vs. gamma rays • Fractionation increased Do and extrapolation number for gamma rays but not neutrons	[120]
Proton beams (130–165 MeV spread-out Bragg peak): 2–8 Gy; in situ thermal neutrons and boron neutron capture (α,^7^Li): 20–80 ppm ^10^B, 2 h pre-irradiation	Human tumor cell lines (HSG, MG63, SAS, G-361) [In vitro study]	Clonogenic survival	• Proton beams with 80 ppm boron increased RBE up to 1.63 and SER up to 1.57 vs. protons alone • Higher intracellular boron correlated with greater sensitization	[121]
BNCT; mixed neutron (thermal: 22.4%, epithermal: 2.4%, fast: 16.7%) + gamma (58.5% of total dose) beam: 1.25 Gy per fraction (neutron)/2 Gy per fraction (gamma)	CHO-K1 Chinese hamster ovary cells [In vitro study]	Clonogenic survival; DNA double-strand breaks (53BP1 foci count and size)	• Fractionated neutron IR led to fewer foci but larger size • Higher D0 vs. single dose • Suggests persistent clustered DSBs due to high LET • Gamma IR showed less foci size difference • Importance of damage complexity in fractionated high-LET fields highlighted	[122]
BNCT; alpha particles (3.2 MeV, LET ~120 keV/μm): 0.5–2.0 Gy; gamma rays (^60^Co): 3.4–8.6 Gy; mixed doses matched for equivalent biological effect (e.g., 0.5 Gy alpha + 3.4 Gy gamma); simultaneous exposure using dual-source setup at 10 °C	V79-4 Chinese hamster cells [In vitro study]	Clonogenic survival (colony formation assay)	• No significant synergistic effect observed • Mixed-beam survival closely matched additive prediction • Results suggest lack of sublethal damage interaction with alpha particles under these conditions	[123]
BNCT; alpha particles: 2 or 2.5 Gy; X-rays: variable doses; simultaneous vs. non-simultaneous delivery assessed	V79 Chinese hamster lung fibroblast cells [In vitro study]	Cell survival (clonogenic assay)	• Strong synergistic effect on cell killing observed only at 2.5 Gy alpha when combined simultaneously with X-rays • Survival curve steepened, suggesting alpha exposure impairs DNA repair from low-LET X-rays	[124]
BNCT; neutron mixed beam (thermal <0.5 eV: 25%, epithermal 0.5–10 keV: ~2.6%, fast >10 keV: 18–19%, LET range not specified): 0.9–1.0 Gy; gamma rays (^60^Co, 40 mGy/min): 0.9–1.0 Gy (controls)	CHO-K1 (wild-type) and xrs5 (Ku80-deficient) Chinese hamster ovary cells [In vitro study]	Clonogenic survival; DNA double-strand breaks (53BP1 foci count, size, spatial distribution)	• RBE at 10% survival: 3.3 (CHO-K1) and 1.2 (xrs5) • Focus number and size similar to gamma rays, but neutron-induced foci were spatially clustered • Indicates potential for more complex DNA damage from neutron components	[125]
Gadolinium neutron capture therapy (GdNCT): ~56.8 μg ^157^Gd/g tumor; boron neutron capture therapy (BNCT): ~158 μg ^10^B/g tumor; thermal neutron irradiation: 1 × 10^9^ neutrons cm^−2^·s^−1^ for 60 min	Mice bearing recurrent head-and-neck tumors (HTB-43 xenografts) [Preclinical study]	Tumor regression, cancer stem cell depletion, survival prolongation	• Nearly complete tumor eradication • Suppression of recurrence biomarkers (TGF-α, p53, CD44, PGE2, HIF-α) • Enhanced apoptosis and necrosis	[126]
^252^Cf neutrons (0.2–0.3 Gy/hr), ^137^Cs gamma rays (0.7–0.85 Gy/hr) and ^60^Co gamma rays (1.25–2 Gy/min); total body irradiation with doses up to ~13 Gy, depending on endpoint and mix ratio	Balb/c mice, whole-body irradiation model; gastrointestinal and bone marrow systems assessed [Preclinical study]	GI-50 (6–10 day survival) and BM-50 (30 day survival) syndromes	• Dose required to cause syndromes ↓ sharply up to 35% neutron contribution, then plateaued • Minimal repair in neutron-rich exposures • Mixed beams ≥35% neutrons behaved like high-LET radiation • Fractionation and dose-rate effects negligible for ^252^Cf but substantial for photons	[127]

* ↑ indicates all language forms related to increase; ↓ indicates all language forms related to decrease.

**Table 5 biomolecules-15-01282-t005:** Combined space radiation studies: ground simulations and NASA Twins Study.

Type and Dose	Biological Systems	Biological Endpoints	Effects	Ref.
Two-ion exposure; protons (1 GeV/n, 0.5 Gy/min): 2Gy; Fe ions (1 GeV/n, 1 Gy/min): 0.75 Gy; sequential exposure with intervals of 2, 30, or 60 min; cells kept at 37 °C between exposures	Human mammary epithelial cells (CH184B5F5/M10)	Chromosome aberrations (mBAND on chromosome 3); intra- and inter-chromosomal exchanges; inversion, deletion, translocation frequency	• Highest aberration frequency observed at 30 min interval • Dual exposure yielded more damage than predicted sum • Supports enhanced susceptibility to Fe damage during early repair phase after proton exposure • Synergy likely driven by interaction of partially repaired lesions	[128]
Two-ion exposure; protons (1 GeV/amu): 1 cGy; Fe ions (1 GeV/amu): 1cGy; applied sequentially with intervals from 3 min to 24 h	AG01522 normal human skin fibroblasts	Micronucleus formation and 53BP1 foci induction in irradiated and bystander cells	• Direct exposure→similar DNA damage levels regardless of single or combined exposure • Bystander response unchanged in signaling cells • Prior proton exposure suppressed bystander response in recipient cells	[129]
Two-ion exposure; protons (1 GeV); titanium ions (1 GeV/n, LET 108.1 keV/μm) or iron ions (1 GeV/n, LET 151.3 keV/μm); protons: 0–20 cGy; HZE dose: 20 cGy; sequential exposure with 15 min delay	Primary human fibroblasts	Neoplastic transformation (anchorage-independent growth in soft agar); clonogenic survival	• Marked synergistic ↑* in transformation when protons preceded HZE by 15 min • Effect evident even at 1 cGy proton + 20 cGy HZE • Split doses of same ion species did not replicate synergy • Results emphasize protons’ role in sensitizing to subsequent HZE exposure	[130]
Two-ion exposure; protons (1 GeV/n); iron or titanium ions (both 1 GeV/n; Fe LET 151.3 keV/μm, Ti LET 108.1 keV/μm; protons: 20 cGy; Fe or Ti ions: 20 cGy; 2.5 min to 48 h in-between irradiations; reverse order also tested	Primary human neonatal fibroblasts	Anchorage-independent growth (soft agar assay); clonogenic survival	• Marked synergistic effect when protons preceded Fe or Ti by 2.5 min–1 h (Fe) or up to ~6 h (Ti) • Transformants per survivor ~3× additive prediction • No synergy when HZE delivered first or at longer intervals • Survival unaffected, suggesting transformation-specific interaction	[131]
Two-ion exposure; ^1^H (150 MeV/n): 0.5 Gy; oxygen ions (^16^O, 600 MeV/n): 0.1 Gy; whole-body irradiation with a 1 h interval	Male C57BL/6 mice; hippocampus (dentate gyrus, CA1)	Short-term spatial memory (Y-maze); dendritic complexity (Sholl); spine density (mushroom, stubby); synaptic marker expression (Nr2a, Nr2b, GluR1, synapsin-1, drebrin, SAP97)	• Impaired memory and ↓ novel arm recognition • ↓ mushroom spines, ↑ stubby spines in DG and CA1 • Altered dendritic arborization and complexity • ↑ GluR1, Nr2a, synapsin-1, drebrin • Hippocampal remodeling consistent with cognitive dysfunction	[132]
Two-ion exposure; ^1^H (150 MeV/n): 0.5 Gy; ^16^O (600 MeV/n): 0.1 Gy	Male C57Bl/6J mice, hippocampal neurons	Short-term memory (Y-maze), recognition memory (novel object recognition, NOR), dendritic morphology, spine density, SNP analysis	• Memory deficits • Reduced novel object recognition • ↓ mushroom spine density • Altered dendritic morphology (↑ in dentate gyrus, ↓ in CA1) • ↑ SNPs in Txnrd2/3	[133]
Two-ion exposure; ^1^H (1 GeV, LET 0.223 keV/µm): 3 × 17 cGy every other day; ^56^Fe (1 GeV/nucleon, LET 151.4 keV/µm): 15 cGy, 2 days after last ^1^H dose (see related commentary)	Cardiovascular system (murine heart)	Left ventricular ejection fraction, posterior wall thickness, LV end-systolic and end-diastolic pressure, dP/dtmax, dP/dtmin; cardiac fibrosis (Masson’s trichrome); infarct size; VEGF-A, p-Akt, p-Erk1/2 expression	• Sequence-dependent cardiac effects: increased fibrosis and LV hypertrophy (^56^Fe + ^1^H) • Impaired post-MI recovery and increased infarct size (^1^H + ^56^Fe)	[134]
Three-ion exposure; protons (1 GeV, 60%); ^16^O ions (250 MeV/n, 20%); ^28^Si ions (263 MeV/n, 20%); total doses: 0, 25, 50, or 200 cGy; sequentially delivered in rapid succession to mimic cosmic radiation exposure	B6D2F1 (C57BL/6J × DBA2/J F1) male and female mice	Behavioral (home-cage activity, depressive behavior), cognitive (object recognition, fear conditioning), molecular (BDNF, CD68, MAP-2 levels), microbiome diversity	• 50–200 cGy impaired object recognition • 50 cGy ↑ depressive behavior and activity (males) • Radiation altered BDNF (↓ in males), CD68 (↑ in females) • Gut microbiome diversity ↑ in dose-dependent fashion	[135]
Three-ion exposure; protons (120 MeV/n): 20 cGy; helium (250 MeV/n): 5 cGy; silicon (300 MeV/n): 5 cGy; whole-body irradiation in 3 orders (H→He→Si, Si→He→H, and H→He +24h→Si); total dose: 30 cGy; CDDO-EA given 3 days pre-IR to 1 day post-IR	*K-ras*^LA1^ lung cancer-susceptible mice; lung tissue and plasma	Lesion number (hyperplasia, adenoma, atypia, carcinoma); plasma MDA levels (oxidative stress marker); tumor incidence	• H→He→Si sequence ↑ premalignant lesions, MDA, and adenocarcinomas • Delaying Si by 24 h or using Si→H→He sequence ↓ effects • CDDO-EA countermeasure normalized lesion count and MDA • Sequence- and timing-dependent cancer risk highlighted	[136]
Three-ion exposure; protons (1000 MeV, LET 0.24 keV/μm): 1.2 Gy; Si ions (500 MeV/n, LET 54 keV/μm): 0.15 Gy; Fe ions (600 MeV/n, LET 190 keV/μm): 0.15 Gy; total dose: 1.5 Gy (whole-body, sequential rapid switching)	WAG/RijCmcr male rats (6 mo); heart, kidney, blood; 270-day follow-up	Perivascular cardiac fibrosis, blood pressure, serum cholesterol, renal histology, cytokine levels (IL-5, IL-18, IL-17A), macrophage (CD68+) infiltration	• 1.5 Gy→perivascular fibrosis and ↑ systolic BP • ↑ CD68+ cells in heart/kidney • Cytokine shifts at 30–60 d • Single-ion beams did not induce pathology • Threshold for fibrosis likely between 0.75–1.5 Gy	[137]
Four-ion simplified mixed field GCR (Smf-GCR); protons 1000 MeV/n, He 250 MeV/n, O 325 MeV/n, Si 300 MeV/n; LET 0.22–69 keV/μm); total dose of 0.5 Gy (0.3 Gy H, 0.1 Gy He, 0.05 Gy O, 0.05 Gy Si) delivered in sequence over ~15 min	Male and female *Apc*^1638N/+^ mice (intestinal tumor model)	Intestinal tumor count and classification (adenoma vs. carcinoma)	• Smf-GCR induced more GI tumors and carcinomas than gamma rays • Heavy-ion fraction (O + Si) accounted for >95% of tumorigenic effect • Males had higher tumor burden • Data highlight heavy-ion dominance in GCR-associated GI cancer risk	[138]
Five-ion simplified GCRsim (H 1000 MeV/n, Si 600 MeV/n, He 250 MeV/n, O 350 MeV/n, Fe 600 MeV/n, plus H 250 MeV/n); total dose of 150 cGy; single whole-body exposure	C57BL/6 male mice (6 months old)	Cardiac function (echocardiography, MRI, pressure-volume loops), aortic histology	• ↑ Arterial elastance • ↓ Preload-recruitable stroke work • Elastin fiber disruption in aorta • Modest ↓ in cardiac output and stroke volume	[139]
Five-ion simplified GCRsim (protons 250/1000 MeV, He 250 MeV/n, O 350 MeV/n, Si 600 MeV/n, Fe 600 MeV/n); total dose of 15 or 50 cGy	Male and female C57Bl/6J mice, immune and endocrine systems (blood, adrenal glands)	Organ weights (thymus, spleen, adrenals), plasma hormone levels (aldosterone, corticosterone), immune cell profiles (phagocytosis, NLR), transcriptomics	• ↓ Thymus/spleen/adrenal weights in males • ↓ Aldosterone in males • ↑ NLR in females (3 days) • ↑ Phagocytosis in males • Sex-specific gene expression changes at 14 days	[140]
Five-ion simplified GCRsim [protons (1 GeV, 250 MeV), He (250 MeV/n); O (350 MeV/n), Si (600 MeV/n), Fe (600 MeV/n)]; total dose of 50 cGy; whole-body irradiation	Male BALB/c mice (n = 12, irradiated vs. sham); hippocampus and bone marrow; tissues analyzed 3 months post-IR	Short-term and spatial memory (Y-maze and Morris water maze); neural cell population changes (astrocytes, NPCs, microglia, oligodendrocytes, assessed via flow cytometry; cytogenetics (G-banding, SKY); proteomics (TMT-based differential protein expression analysis)	• GCR exposure impaired short-term and spatial memory • No glial cell changes • Chromosome aberrations ↑ 4× vs. sham • 113 proteins differentially expressed (fold change > 1.5) • Network analysis linked protein shifts to cognition and neurodegeneration	[141]
Five-ion simplified GCRsim; protons (1000 MeV, LET 0.20 keV/μm): 17.5 cGy; Si (600 MeV/n, LET 50.4 keV/μm): 0.5 cGy; He (250 MeV/n, LET 1.60 keV/μm): 9 cGy; O (350 MeV/n, LET 20.9 keV/μm): 3 cGy; protons (250 MeV, LET 0.40 keV/μm): 19.5 cGy; Fe (1000 MeV/n): 0.5 cGy; total dose: 50 cGy or 100 cGy depending on group.	Male and female mice; hippocampus, blood, cortex	Spatial learning (RAWM), sociability, social memory, recognition memory; microglia phenotype (CD68, CD107a), synaptic density (PSD-95, Synapsin-1), blood monocyte levels	• GCRsim caused sex-specific spatial learning deficits (males only) • Linked to microglia activation and ↑ synapses • Microglia depletion reversed deficits • Early blood monocyte levels predicted late cognitive decline in males	[142]
Five-ion simplified GCRsim (H, He, O, Si, Fe): 0.5 or 0.75 Gy; gamma rays: 0.75 or 2 Gy; whole-body irradiation	Male and female C57BL/6J wild-type and APPswe/PS1dE9 transgenic mice; brain, heart, kidney, plasma	Spatial memory (Y-maze), anxiety (EPM, OFT), sensorimotor gating (PPI), rotarod, MRI volumes, gene expression (VLCAD, Casp3, GLUT4), cytokine levels	• No effect on Aβ • GCRsim and gamma caused sex-specific MRI and behavior changes • GCRsim ↓ VLCAD, Casp3, GLUT4 • Male Tg mice more affected neurologically • GCRsim ↑ hippocampal volumes and ventricular enlargement • Minor kidney/heart fibrosis and altered cytokines	[143]
Five-ion simplified GCRsim (H, He, O, Si, Fe); total dose of 500 mGy delivered over ~4.5 h in 6 sequential beams; TGF-βRI inhibitor (IPW-5371) administered in diet pre- and post-IR	BALB/c (male) and CD1 (male/female) mice; cardiac tissue and plasma collected 12–20 weeks post-IR	Cardiac structure/function, collagen deposition, capillary density, immune markers (CD2, CD4, CD45, TLR4), TGF-β1 expression	• GCRsim caused minor cardiac changes • ↑ Collagen and ↓ TLR4 in CD1 males mitigated by TGF-β inhibition • CD1 females showed ↑ capillaries and ↓ ventricular mass • Combined GCRsim + inhibitor altered immune cell marker profiles in both sexes	[144]
Five-ion simplified GCRsim (H 1000 MeV/n, He 250 MeV/n, O 350 MeV/n, Si 600 MeV/n, Fe 600 MeV/n); total dose of 5, 15, or 50 cG; whole-body irradiation	Male and female C57BL/6J mice; undisturbed home-cage behavior	Species-typical behaviors: burrowing, grooming, rearing, nest building (Deacon score); Neuroscore test battery (7 tasks)	• No sensorimotor deficits detected • Sex- and dose-specific changes in burrowing (↑ at 15 cGy in females) and grooming (↑ at 50 cGy in females) • Nestlet construction differed by sex and dose • More robust female performance • Delayed effects subtle and behavior-dependent	[145]
Five-ion simplified GCRsim beam; total dose of 15 cGy (0.5 cGy/min over ~20 min)	Male Wistar rats (8–9 months old)	Sleep architecture (NREM, REM, TST), EEG spectral power (delta, theta, alpha, sigma, beta bands), core body temperature (CBT)	• ↓ Dark period TST • ↓ NREM • ↓ REM • ↓ NREM delta power • ↓ REM theta power • ↑ NREM/REM alpha and sigma power • ↓ CBT during light period	[146]
Five-ion simplified GCRsim beam; total dose of 0.5 Gy delivered over ~20 min	Female C57Bl/6 mice (liver, heart, plasma, soleus muscle); human 3D microvessel cultures (HUVEC-derived)	Microvessel integrity (angiogenesis, collapse), DNA double-strand breaks (53BP1 foci), mitochondrial function, inflammatory pathways (cytokines, ISGs)	• ↓ Microvessel collapse • ↓ DNA DSBs • ↓ Inflammatory signaling (TNF-α, IL-6) • ↑ Mitochondrial function rescue after miRNA antagomir treatment	[147]
Five-ion simplified GCRsim beam; total dose of 0.75 Gy; gamma rays: 2 Gy; whole-body irradiation	Male and female transgenic mice (APP;E3F, APP;E4F); hippocampus, plasma, feces	Locomotion/anxiety (open field), motor coordination/learning (rotarod), recognition memory (NOR), spatial memory (Y-maze); hippocampal Aβ pathology (S97, ThioS), brain ApoE levels, plasma cytokine and lipids levels; gut microbiome composition (16S rRNA)	• Modest cognitive and neuropathological effects of GCRsim vs. sham • Radiation interacted with sex, genotype, travel • GCRsim ↓ plasma IL-6, TNF-α, HDL • Long-term microbiome shifts correlated with plaque burden and memory • ApoE genotype shaped responses	[148]
Five-ion simplified GCRsim (1 GeV/n protons 35%, 250 MeV/n protons 39%, 250 MeV/n helium 18%, 350 MeV/n oxygen 6%, 600 MeV/n silicon 1%, 600 MeV/n iron 1%; total dose of 10 cGy	Male and female Wistar rats	Risk-taking propensity (RTP), decision-making performance, processing speed	• ↑ Risk-taking in females (↓ profitable choices at 30–60 days) • ↑ Decision latency in males (~2× slower at 30 days) • Performance recovery by 90 days	[149]
Five- or six-beam simplified GCRsim; total dose of 5 or 30 cGy; whole-body irradiation	Male C57BL/6J mice; hippocampus, cortex	Hippocampal inhibitory synaptic activity, LFP oscillations, spatial and recognition memory, anxiety behavior	• Mixed-ion exposure disrupted hippocampal GABAergic signaling • Slowed sharp-wave ripple frequency • Impaired memory and ↑ anxiety-like behavior at 30 cGy	[150]
Five-ion simplified GCRsim (1 GeV/n protons 35%, 250 MeV/n protons 39%, He 18%, O 6%, Si 1%, Fe 1%); total dose of 10 cGy	Female Wistar rats	Task switching performance, stimulus-response training success, switch cost errors	• ↓ Switch task accuracy (−20%) • ↑ Perseverative errors (anterograde interference) • ↑ Failure to complete training stages • No change in response times	[151]
Five-ion simplified GCRsim (1 GeV/n protons 17.5 cGy, 250 MeV protons 19.5 cGy, He 9 cGy, O 3 cGy, Si 0.5 cGy, Fe low dose); total dose of 50 cGy	Male and female CD1 mice, retina tissue	BRB integrity (AQP-4, PECAM-1, ZO-1 expression), oxidative stress (4-HNE), apoptosis (TUNEL assay)	• ↑ AQP-4 expression (female > male) • ↑ PECAM-1 expression (male > female) • ↓ ZO-1 expression • ↑ Oxidative stress (4-HNE) • ↑ Retinal apoptosis	[152]
Five-ion simplified GCRsim (1 GeV/n protons 35%, 250 MeV/n protons 39%, He 18%, O 6%, Si 1%, Fe 1%); total dose of 75 cGy	H9c2 myoblasts (rat), ES-D3 pluripotent cells (murine), Hy926 endothelial cells (human)	Mitochondrial function (MTT, TMRE), oxidative stress (DHE), cell senescence (ONPG), neoplastic transformation (Afp-tdTomato expression)	• ↓ Mitochondrial function • ↑ Oxidative stress • ↑ Senescence markers • ↑ Neoplastic transformation markers (Afp-tdTomato)	[153]
X-rays (160 kVp): 0.1–1.0 Gy pretreatment, 8 Gy challenge; five-ion simplified GCRsim [1000 MeV/n proton: 26.25 cGy; 250 MeV/n proton: 29.25 cGy, helium (250 MeV/n): 13.50 cGy, oxygen (350 MeV/n): 4.50 cGy, silicon (600 MeV/n): 0.75 cGy, iron (600 MeV/n): 0.75 cGy]: 75 cGy challenge	H9c2 rat cardiomyoblast cells	Cell doubling time, MTT viability, mitochondrial ROS (DHE/MitoSox), membrane potential (TMRE)	• X-ray pretreatment restored replicative capacity and ↓ cytosolic superoxide after GCRsim • Strongest adaptive effect at 0.5–1.0 Gy • Mitochondrial metrics unaffected • Supports low-dose X-ray hormesis as potential countermeasure	[154]
Five-ion simplified GCRsim: 10 cGy; ^4^He (250 MeV/n, LET 1.6 keV/μm): 10 cGy; whole-body irradiation of male Wistar rats; comparisons between single-ion (He) and complex-ion (GCRsim) groups	Male Wistar rats	Attentional set-shifting performance (ATSET): attempts to reach criterion (ATRC, error rate), mean correct latency (processing speed), stage-specific set shifting (SD, IDR, EDS, etc.), practice effects (practice savings ratio).	• GCRsim and He exposure both impaired SD stage of ATSET • GCRsim-exposed rats required more time/iterations to solve tasks • PE ↓ (50% slower than pretest vs. 30% improvement in sham) • UCFlex deficits also observed • Findings suggest diminished cognitive flexibility and ↓ learning from repetition after space radiation exposure	[155]
Five-ion simplified GCRsim; protons (1000 MeV/n, 35%), Si (600 MeV/n, 1%), He (250 MeV/n, 18%), O (350 MeV/n, 6%), Fe (600 MeV/n, 1%), final proton fraction (250 MeV/n, 39%); total dose 500 mGy delivered sequentially; amifostine pretreatment (107 or 214 mg/kg i.p. 1 h prior)	C57Bl/6J male and female mice	Behavioral testing (novel object recognition), locomotion/habituation and anxiety-like behavior (open field), home-cage activity (light/dark cycle), whole-brain cFos immunoreactivity and connectivity	• Combined heavy-ion irradiation impaired novel object recognition in males but not females • Amifostine pretreatment mitigated deficits in males and modulated brain regional connectivity • Sex-specific behavioral effects observed	[156]
Five-ion simplified GCRsim (mixed field; total dose 0.1–2.0 Gy); also separately evaluated same dose of: proton (150 MeV, LET 0.54 keV/μm), carbon (600 MeV/n, LET 9.18 keV/μm), iron (600 MeV/n, LET 172.4 keV/μm), and gamma rays (^137^Cs)	Latently CMV-infected Kasumi-3 human myeloblast cells	CMV reactivation (viral DNA load, qPCR), cell viability (% live cells), cell size (μm), viral genomic variation (sequencing), viral transcriptomics (UL49)	• All IR types induced CMV reactivation in dose- and time-dependent fashion • GCRsim triggered UL49 upregulation (logFC 1.48) • Carbon/iron led to strongest CMV load • No genomic variation • Cell viability and size affected by LET	[157]
Six-beam simplified GCRsim (74% protons, 18% helium, 6% oxygen, 1% silicon, 1% iron); total dose of 10 cGy	Male Wistar rats	Approach time, pull duration, movement accuracy (misses/contacts), reach endpoint concentration	• ↑ Pull duration • Mild ↑ approach time (some individuals) • No significant change in movement accuracy or reach concentration vss Sham at 72 h	[158]
Six-ion simplified GCRsim; protons (1 GeV, LET 0.24 keV/μm); He (250 MeV/n, LET 1.6 keV/μm); O (250 MeV/n, LET 25 keV/μm); Si (263 MeV/n, LET 78 keV/μm); Ti (1 GeV/n, LET 107 keV/μm); Fe (1 GeV/n, LET 151 keV/μm); total dose of 25, 50, or 200 cGy; rapid sequential whole-body irradiation	B6D2F1 male and female mice (n = 99); cortical and hippocampal brain tissue, feces (microbiome)	Locomotion and anxiety-like behavior (open field), recognition memory (object recognition), depression-like behaviour (forced swim), associative learning and memory (contextual/cued fear conditioning, passive avoidance); cortical and hippocampal BDNF, CD68, MAP-2 levels (ELISA); gut microbiome composition (16S rRNA sequencing)	• 50 and 200 cGy impaired object recognition and passive avoidance memory in females • Altered BDNF/CD68 patterns in males • Microbiome composition changed in sex- and dose-dependent ways • Strong link between microbiota and behavioral metrics	[159]
Six-beam simplified GCRsim [H (1 GeV); Si (600 MeV/n); He (250 MeV/n); O (350 MeV/n); Fe (600 MeV/n); H (250 MeV)]; total dose: 5 cGy or 30 cGy; whole-body irradiation	Middle-aged male mice; hippocampus (CA1)	Memory updating, discrimination index (DI), long-term potentiation (LTP), p-cofilin expression	• 30 cGy GCR impaired memory updating and LTP • 5 cGy had mild or no effect • Systemic HDAC3 inhibition reversed LTP impairment • ↓ p-Cofilin suggests cytoskeletal mechanism • First evidence of epigenetic rescue of GCR-induced synaptic dysfunction	[160]
Seven-ion simplified GCRsim (H, He, C, O, Si, Ti, Fe; 20–1000 MeV/n); chronic exposure: 2.08 cGy/day × 6 days/week × 4 weeks (total 50 cGy); whole-body irradiation	Female *Apc*^Min/+^ mice; mammary tissues and serum	Mammary tumor incidence, ductal morphology, serum estradiol and SPP1, ERα/ERRα expression (IHC/qPCR)	• ↑ Ductal overgrowth, ↑ mammary tumors (24% vs. 5%) • ↑ Serum estradiol, ERα, ERRα, SPP1 expression • Estrogen and inflammatory signaling linked to tumorigenesis • Conserved expression patterns in human breast cancer tissues	[161]
Thirty-three-beam GCRsim (33 separate beams involving H, He, C, O, Si, Ti, Fe, and other ions; various LETs and energies); acute dose: ~40 cGy in 2 h; chronic dose: ~50 cGy over 24 sessions; whole-body irradiation	C57BL/6J mice (n = 178 male, 91 female); hippocampus, mPFC, corpus callosum	Memory updating (object in updated location, OUL), recognition memory (NOR), anxiety-like behavior (light–dark box, LDB), social interaction (SIT), aggression/dominance (tube dominance); hippocampal synaptic plasticity (LTP), excitatory/inhibitory synaptic currents (sEPSC/sIPSC), synapse morphology (PSD length, spine size)	• Chronic GCR impaired memory updating in both sexes • Acute exposure disrupted excitatory synaptic signaling • LTP ↓ in both sexes • Chronic GCR thinned PSD in large spines, altered myelination in small/large axons • Sex-specific deficits in behavior and synaptic function	[162]
Thirty-three-beam GCRsim (H, He, C, O, Si, Ti, Fe, etc.; 40–49.9 cGy total); acute dose: 40 cGy in 2 h; chronic dose: ~50 Gy over 4 weeks; whole-body irradiation	Male C57BL/6 mice; prefrontal cortex (PFC)	Reward sensitivity/motivation (touchscreen-based economic demand), sustained attention and reaction time (psychomotor vigilance tasks); PFC neurotransmitter responsiveness (DA, 5-HT, NE, Glu, GABA levels)	• No change in motivation, but attentional deficits and slowed reaction times in GCR groups • DA signaling blunted in PFC • Chronic GCR ↑ all neurotransmitters under stimulation • Acute and chronic GCR reorganized neurotransmitter networks (DA, 5-HT, NE) • DA-GABA-Glu connectivity disrupted • Suggests persistent PFC network dysfunction	[163]
Thirty-three-beam GCRsim (H, He, C, O, Si, Ti, Fe, etc.); acute dose: 40 cGy in 1 day; chronic: 50 cGy over 24 days (2.08 cGy/day); whole-body irradiation	Male and female C57BL/6J mice	Locomotion (distance traveled, stop time), stop clustering, home base stability, edge preference (% stops at periphery), navigation accuracy (path circuity), progression speed (peak speed), heading changes (directional control).	• Acute exposure caused slower, more circuitous return paths under light conditions • Chronic exposure had less disruption • Light-dependent deficits in spatial navigation emerged only with acute irradiation	[164]
Thirty-three-beam GCRsim (H, He, O, Si, Ti, Fe, etc.) ± neutrons (10 cGy); acute (1.5–2 h) or chronic (4–6 week) exposure to 50, 75, or 100 cGy; 10 cGy neutron added 6 mo after acute 75 cGy GCRsim	*K-ras*^LA1^ lung cancer-susceptible mice (male/female); lung tissue and plasma; 1-year follow-up	Adenocarcinoma incidence, premalignant lesion number/size, survival, lipid peroxidation (MDA assay)	• GCRsim dose- and schedule-dependent ↑ in lung adenocarcinoma • Chronic exposure > acute • 10 cGy neutrons post-GCRsim ↑ malignancy • No survival impact except with neutrons • Lesion size ↑ at 100 cGy • Implications for Mars mission cancer risk	[165]
Thirty-three-beam GCRsim; acute dose: 0.75 Gy over ~1.5 h, whole-body irradiation ± antioxidant CDDO-EA pre/post	Female C57BL/6J mice (6-month-old); dentate gyrus and cortex; 14.25-month follow-up	Pattern separation and cognitive flexibility (location discrimination reversal, LDR), stimulus–response acquisition/extinction, social interaction, recognition memory (NOR), locomotion and anxiety-like behaviour (open field), hippocampal neurogenesis (DCX+ immature neurons)	• GCRsim caused deficits in cognitive flexibility and ↓ DCX+ neurons • CDDO-EA mitigated LDR impairments • Anxiety, sociability, and locomotion unaffected • Female resilience differed from male studies	[166]
Thirty-three-beam GCRsim; acute dose: 0.75 Gy over 1.5 h; whole-body irradiation	Male C57BL/6J mice (n = 22–24/group)	Sociability, social novelty preference, anxiety-like behavior, object recognition memory (NOR)	• 33-GCR did not alter NOR or anxiety but blunted preference for social novelty • CDDO-EA did not prevent this effect • CDDO-EA + GCR also impaired sociability • Findings highlight CNS vulnerability to complex mixed-field space radiation and need for targeted neuroprotective countermeasures	[167]
Thirty-three-beam GCRsim (seven ion species across 20–1000 MeV/n); total dose of 50 cGy	Female *Apc*^Min/+^ mice, mammary gland tissues	Ductal proliferation, ductal overgrowth, preneoplasia markers (Spp1, Rrm2)	• ↑ Ductal branching and hyperplasia • ↑ Cyclin D1+ cell proliferation • ↑ Spp1 expression (gene and protein) • ↑ Rrm2 expression (mRNA and protein)	[168]
X-rays (190 kVp): 0–2 Gy; alpha particles (LET 90.9 keV/μm): 0–2 Gy; mixed beam (X-rays + alpha, 1:1 dose ratio): 0–2 Gy total (0–1 Gy each component)	Human peripheral blood lymphocytes (2 male donors, in vitro)	Chromosomal aberrations, mRNA expression (FDXR, CDKN1A, MDM2), alternative transcription	• Synergistic increase in chromosomal aberrations and gene expression across seasons • Alpha > X-ray effectiveness • No synergism in alternative transcription • Inter/intra-donor variability observed	[169]
UV-B: 25–100 J/m^2^; protons (LET ~4.7 keV/μm): 0.25–0.5 Gy; gamma rays: 0.5 Gy; sequential exposure (≤20 min apart)	Human non-malignant cells: HaCaT keratinocytes, Hs27 fibroblasts, CRL 9855 monocytes, PBMCs	DNA damage (γH2AX, dicentrics), gene expression, viability (MTS, LDH), genomic instability	• Marked synergistic effects in co-exposed fibroblasts and keratinocytes • ↑ γH2AX foci, pan-nuclear staining, stress gene upregulation, and ↓ viability • Synergy less evident in UV-B-sensitive monocytes • Gamma + UV-B mimicked proton + UV-B responses	[170]
Neutrons (5 MeV p/d on Be target, LET 10–200 keV/μm): 0.33 Gy; photons (concomitant): 0.07 Gy (acute); neutrons/photons (^252^Cf, LET ~100 keV/μm): 0.4 Gy total at ≤1 mGy/day (chronic); 7-ion simplified GCRsim (H, He, C, O, Si, Ti, Fe ions, various energies): 0.4 Gy (acute ~2 h or 19 fractions over 1 month)	C3H male and BALB/c female mice	Locomotion and anxiety-like behavior (open field), recognition memory (NOR), contextual/cued fear conditioning	• Chronic mixed-field exposure impaired novel object recognition • Acute GCRsim ↓ exploratory behavior • Fractionated GCRsim→trend toward lower fear learning • Aspirin failed to mitigate radiation effects and worsened object recognition in sham controls	[171]
^40^Ar (550 MeV/n, 86 keV/μm), ^28^Si (100 MeV/n, 150 keV/μm), ^56^Fe (115 MeV/n, 442 keV/μm); X-rays (150 kVp); dose combinations ranged from 1–11 Gy total; simultaneous exposure	Hamster V79 fibroblasts; human lymphocytes	Clonogenic survival (V79); chromosome 2 aberrations (FISH-PCC, lymphocytes)	• Additive effects for Ar and Si ions with X-rays • Slight non-additive deviation (mild synergy) observed for Fe ions, especially at 1:1 dose ratio • Fe + X-ray survival and chromosomal damage slightly exceeded predicted additive response	[172]
Gamma rays (661.7 keV): 0.4 Gy; ^12^C ions (450 MeV/n, 10.3 keV/μm): 0.14 Gy; whole-body gamma ray exposure followed 24 h later by head-only ^12^C ions	Adult male Wistar rats; nucleus accumbens (NAc) and dorsal striatum (dST)	Locomotor activity, grip strength, monoamine and choline metabolism (HPLC), gene/protein expression of STX1A and SNCA (qPCR, immunoblotting)	• IR caused hyperlocomotion and enhanced intrasession habituation • ↑ Choline and α-synuclein, ↓ STX1A in NAc • ↓ 5-HIAA • STX1A protein ↓ in dST • Suggests link to vesicle trafficking and neurotransmission modulation	[173]
Gamma rays (661.7 keV): 0.4 Gy; ^12^C ions (450 MeV/n, 10.3 keV/μm): 0.14 Gy; whole-body gamma irradiation daily for 3 days, followed by acute ^12^C head-only exposure on day 4	Wistar rats (n = 14), pituitary gland analyzed post-mortem	C/EBP-β protein isoform levels (LAP*, LAP, LIP, Western blot); mRNA expression of C/EBP-β isoforms (qPCR)	• No change in mRNA levels, but 1.76× increase in C/EBP-β LIP isoform protein in irradiated rats • Suggests translation-stage regulation • Indicates ER stress and potential apoptosis signaling via HPA axis modulation	[174]
Galactic cosmic rays (GCRs); microgravity, circadian disruption, and other spaceflight stressors; estimated GCR dose: ~76 mGy (physical), ~146 mSv (effective) over 340 days; complex exposure mix including HZE particles; environmental stressors act in parallel	One monozygotic twin in space (TW) vs. Earth-based twin (HR)	Multi-system effects: transcriptomic, epigenetic, proteomic, metabolomic, immune, cardiovascular, ocular, microbiome, cognitive	• Multiple synergistic effects inferred • Persistent chromosomal inversions (DNA damage), gene dysregulation, telomere elongation→rapid shortening • Altered immune networks, inflammation, cognitive decline • Consistent with complex biological interaction of radiation with other spaceflight stressors	[175]

* ↑ indicates all language forms related to increase; ↓ indicates all language forms related to decrease.

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
