# Peer review of "Combined Radiations: Biological Effects of Mixed Exposures Across the Radiation Spectrum"

_biomolecules, 2025, doi:10.3390/biom15091282_

Round 1

Reviewer 1 Report

Comments and Suggestions for Authors

The manuscript, titled “Combined Radiations: Biological Effects of Mixed Exposures Across the Radiation Spectrum,” compiles reports from over 170 studies that examine the biological effects of combined radiation treatments. The findings are categorized into three domains: radiobiological, therapeutic, and space radiation. It was discovered that combined radiation treatments-whether ionizing with non-ionizing, non-ionizing with non-ionizing, or ionizing with ionizing-often exhibited synergistic effects. The manuscript discusses the various factors that influence the different outcomes of these combined radiation treatments. It also addresses methodological challenges and experimental gaps while emphasizing the need for more thorough mechanistic investigations. Overall, the manuscript is well-written and provides a comprehensive overview of the current understanding of the biological effects of combined radiation, as well as insights into future research directions. Some issues need to be addressed:

  1. The organization of information in the tables is unclear. I strongly suggest that the authors categorize the studies based on the investigation systems: human (e.g., cells and skin models), animals (e.g., jellyfish polyps), plants, yeast, microbes (for humans or plants), and viruses (e.g., enteroviruses, bacteriophages, etc.). This approach would better highlight the radiobiological effects on cells across different domains, as biological responses-such as DNA repair capacity-can vary significantly between different systems, particularly between eukaryotes and prokaryotes.
  2. The order of treatments with different types of radiation in the tables is only specified in some studies. Is there a reason that this kind of information is missing in many of the other investigation reports?
  3. Including the major goal of each study would make the tables more informative.
  4. In Table 4, human subjects, human cell lines, organoids, mouse and rat models, and animal cells are listed. It would be clearer if the study models were categorized into clinical, pre-clinical, and in vitro systems.
  5. The title for Table 5 ismisleading. Most of these experiments (with the exception of the NASA Twin Study) were conducted using space radiation simulations and should be labeled as such.
  6. Reference [113] does not appear to refer to a biological system, and Reference [128], which discusses biological polymers, should not be classified as a biological system either.
  7. The text related to each table should be placed immediately after the individual table, rather than grouping all the tables together at the beginning of the manuscript.
  8. The content in the Discussion section is somewhat redundant. There are several repetitive points that were already mentioned in the Results section. For instance, the observation that mixed radiation fields can reshape transcriptional networks, even in the absence of overt cytotoxicity, and that the NASA Twins Study provided translational support mirroring results observed in rodent and cell-based GCR simulations, among others.

Author Response

Combined Radiations: Biological Effects of Mixed Exposures
Across the Radiation Spectrum

Parousis-Paraskevas et al., Biomolecules, MDPI

1st Submission Date: 23 June 2025
Date of 1st Review: 16 Jul 2025 01:58:16

Response to the Comments Made by Reviewer 1

Dear Reviewer 1,

For a start, we would like to thank you for taking the time to compile such useful and insightful comments. In fact, your comments ignited a creative process through which many more than those errata you initially pointed out were tracked and corrected. Below are given the detailed answers to your specific comments.  

Comment 1:

The organization of information in the tables is unclear. I strongly suggest that the authors categorize the studies based on the investigation systems: human (e.g., cells and skin models), animals (e.g., jellyfish polyps), plants, yeast, microbes (for humans or plants), and viruses (e.g., enteroviruses, bacteriophages, etc.). This approach would better highlight the radiobiological effects on cells across different domains, as biological responses-such as DNA repair capacity-can vary significantly between different systems, particularly between eukaryotes and prokaryotes.

Response 1:

Thank you for this comment. We are aware that it is common practice in publications to sort table entries from biological experiments based on the biological system. However, our main focus was on the specific radiation combinations in each case. That said, we took this comment into sincere consideration and proceeded to regroup the systems into coherent sets within each radiation combination. For example, in Table 1, radiation combinations go from UV–X-rays to UV–gamma rays, to UV–beta, UV–protons, and NIR–X-rays, laser–gamma rays, whilst systems follow a human-to-rodents-to-other-animals-and-plants-to-microbes order, without interfering with the given radiation combinations’ order.

Comment 2:

The order of treatments with different types of radiation in the tables is only specified in some studies. Is there a reason that this kind of information is missing in many of the other investigation reports?

Response 2:

A comment on target, thank you. The reason for this is that the studies were inserted in the tables in relation to their accompanying texts; that is, in cases where exposure order was of specific interest, the corresponding texts and table entries included a reference to—or more extensive commenting on—this information. So, it was related to what our interest lay with in each case.

Comment 3:

Including the major goal of each study would make the tables more informative. In Table 4, human subjects, human cell lines, organoids, mouse and rat models, and animal cells are listed. It would be clearer if the study models were categorized into clinical, pre-clinical, and in vitro systems.

Response 3:

Similar to Comment 1, we also considered regrouping the existing entries according to the tripartite structure of in vitro, pre-clinical, and clinical studies. However, as previously stated, our aim here was to emphasize the specific combinations of radiation types. For this reason, all tables begin with the radiation types listed in the first column. Categorizing the entries according to this otherwise typical and useful structure—particularly in biological contexts—would require mixing the radiation types, thereby shifting the focus away from them. What we decided to do instead was add a special tag in brackets in every biological system cell of the therapeutic studies table (Table 4), indicating study types (e.g., [Preclinical Study] or [In vitro Study]).

Comment 4:

The title for Table 5 is misleading. Most of these experiments (with the exception of the NASA Twin Study) were conducted using space radiation simulations and should be labeled as such.

Response 4:

Thank you for this comment. You are right to point out that, with the exception of the NASA Twin Study, the experiments listed in Table 5 were conducted using ground-based simulations of space radiation. To more accurately reflect this distinction, we have revised the table title to “Combined Space Radiation Studies: Ground Simulations and NASA Twin Study.” We hope this clarifies the nature and context of the included studies.

Comment 5:

Reference [113] does not appear to refer to a biological system, and Reference [128], which discusses biological polymers, should not be classified as a biological system either.

Response 5:

A very helpful comment, which led us to find and substitute these two, as well as other studies which had to be replaced others better matching our combined exposure criteria, their corresponding texts receiving modifications as well. All related changes are summarized in the following table:

Replaced entries

(short title)

Citation (June 23, 2025)

Replacement study

(full citation)

Waldren, C.A.; Johnson, R.T. Analysis of interphase chromosome damage by means of premature chromosome condensation after X- and ultraviolet-irradiation (1974)

[9]

Han, A.; Elkind, M.M. Ultraviolet light and X-ray damage interaction in Chinese hamster cells. Radiation Research 1978, 74, 88–100.

Guffey, J.S.; Wilborn, J. In Vitro Bactericidal Effects of 405-nm and 470-nm Blue Light (2006)

[34]

Zhang, W.; Zhao, W.; Sun, Y.; Sun, Y.; Wang, B.; Liu, M.; Qiu, Z.; Wang, Y.; Sun, Z.; Hu, P.; et al. Effects of multi-wavelength ultraviolet radiation on the inactivation and reactivation of Escherichia coli in recirculating water system. Aquaculture Reports 2025, 41, 102688, doi:10.1016/j.aqrep.2025.102688.

Yamamoto, N.; Ikeda, C.; Yakushiji, T.; Nomura, T.; Katakura, A.; Shibahara, T.; Mizoe, J. Genetic effects of X-ray and carbon ion irradiation in head and neck carcinoma cell lines(2007)

[85]

Guerra Liberal, F.D.C.; Thompson, S.J.; Prise, K.M.; McMahon, S.J. High-LET radiation induces large amounts of rapidly-repaired sublethal damage. Scientific Reports 2023, 13, 11198, doi:10.1038/s41598-023-38295-3.

Yu, Z.; Hong, Z.; Zhang, Q.; Lin, L.-C.; Shahnazi, K.; Wu, X.; Lu, J.; Jiang, G.; Wang, Z. Proton and carbon ion radiation therapy for locally advanced pancreatic cancer: A phase I dose escalation study (2020)

[86]

Mizumoto, M.; Tsuboi, K.; Igaki, H.; Yamamoto, T.; Takano, S.; Oshiro, Y.; Hayashi, Y.; Hashii, H.; Kanemoto, A.; Nakayama, H.; et al. Phase I/II trial of hyperfractionated concomitant boost proton radiotherapy for supratentorial glioblastoma multiforme. International Journal of Radiation Oncology*Biology*Physics 2010, 77, 98–105, doi:10.1016/j.ijrobp.2009.04.054.

Guan, X.; Gao, J.; Hu, J.; Hu, W.; Yang, J.; Qiu, X.; Hu, C.; Kong, L.; Lu, J.J. The preliminary results of proton and carbon ion therapy for chordoma and chondrosarcoma of the skull base and cervical spine (2019)

[87]

Slater, J.D.; Yonemoto, L.T.; Mantik, D.W.; Bush, D.A.; Preston, W.; Grove, R.I.; Miller, D.W.; Slater, J.M. Proton radiation for treatment of cancer of the oropharynx: Early experience at Loma Linda University Medical Center using a concomitant boost technique. International Journal of Radiation Oncology*Biology*Physics 2005, 62, 494–500, doi:10.1016/j.ijrobp.2004.09.064.         

Ma, N.-Y.; Chen, J.; Ming, X.; Jiang, G.-L.; Lu, J.J.; Wu, K.-L.; Mao, J. Preliminary safety and efficacy of proton plus carbon-ion radiotherapy with concurrent chemotherapy in limited-stage small cell lung cancer (2021)

[88]

Munzenrider, J.E.; Liebsch, N.J. Proton therapy for tumors of the skull base. Strahlentherapie und Onkologie 1999, 175, 57–63, doi:10.1007/BF03038890.

Aljabab, S.; Lui, A.; Wong, T.; Liao, J.; Laramore, G.; Parvathaneni, U. A combined neutron and proton regimen for advanced salivary tumors: early clinical experience (2021)

[91]

Arbuznikova, D.; Klotsotyra, A.; Uhlmann, L.; Domogalla, L.-C.; Steinacker, N.; Mix, M.; Niedermann, G.; Spohn, S.K.B.; Freitag, M.T.; Grosu, A.L.; et al. Exploring the role of combined external beam radiotherapy and targeted radioligand therapy with 177Lu-PSMA-617 for prostate cancer - from bench to bedside. Theranostics 2024, 14, 2560–2572, doi:10.7150/thno.93249.

Marvaso, G.; Vischioni, B.; Pepa, M.; Zaffaroni, M.; Volpe, S.; Patti, F.; Bellerba, F.; Gandini, S.; Comi, S.; Corrao, G.; et al. Mixed-beam approach for high-risk prostate cancer carbon-ion boost followed by photon intensity-modulated radiotherapy: preliminary results of phase II trial AIRC-IG-14300 (2021)

[105]

Carvalho, H.D.A.; Mendez, L.C.; Stuart, S.R.; Guimarães, R.G.R.; Ramos, C.C.A.; Paula, L.A.D.; Sales, C.P.D.; Chen, A.T.C.; Blasbalg, R.; Baroni, R.H. Implementation of image-guided brachytherapy (IGBT) for patients with uterine cervix cancer: a tumor volume kinetics approach. Journal of Contemporary Brachytherapy 2016, 4, 301–307, doi:10.5114/jcb.2016.61703.

Gugliandolo, S.G.; Marvaso, G.; Comi, S.; Pepa, M.; Romanò, C.; Zerini, D.; Augugliaro, M.; Russo, S.; Vischioni, B.; Valvo, F.; et al. Mixed-beam approach for high-risk prostate cancer: Carbon-ion boost followed by photon intensity-modulated radiotherapy. Dosimetric and geometric evaluations (AIRC IG14300)(2020)

[106]

Shiba, S.; Shimo, T.; Yamanaka, M.; Yagihashi, T.; Sakai, M.; Ohno, T.; Tokuuye, K.; Omura, M. Increased cell killing effect in neutron capture enhanced proton beam therapy. Scientific Reports 2024, 14, 28484, doi:10.1038/s41598-024-79045-3.         

Alterio, D.; D’Ippolito, E.; Vischioni, B.; Fossati, P.; Gandini, S.; Bonora, M.; Ronchi, S.; Vitolo, V.; Mastella, E.; Magro, G.; et al. Mixed-beam approach in locally advanced nasopharyngeal carcinoma: IMRT followed by proton therapy boost versus IMRT-only. Evaluation of toxicity and efficacy(2020)

[107]

Kolker, J.D.; Halpern, H.J.; Krishnasamy, S.; Brown, F.; Dohrmann, G.; Ferguson, L.; Hekmatpanah, J.; Mullan, J.; Wollman, R.; Blough, R.; et al. “Instant-mix” whole brain photon with neutron boost radiotherapy for malignant gliomas. International Journal of Radiation Oncology*Biology*Physics 1990, 19, 409–414, doi:10.1016/0360-3016(90)90550-4.

Zhang, R.; Heins, D.; Sanders, M.; Guo, B.; Hogstrom, K. Evaluation of a mixed beam therapy for postmastectomy breast cancer patients: Bolus electron conformal therapy combined with intensity modulated photon radiotherapy and volumetric modulated photon arc therapy (2018)

[108]

Little, J.B.; Nagasawa, H.; Pfenning, T.; Vetrovs, H. Radiation-induced genomic instability: delayed mutagenic and cytogenetic effects of X rays and alpha particles. Radiation Research 1997, 148, 299–307.

Åsell, M.; Hyödynmaa, S.; Söderström, S.; Brahme, A. Optimal electron and combined electron and photon therapy in the phase space of complication-free cure (1999)

[112]

Míguez, C.; Jiménez-Ortega, E.; Palma, B.A.; Miras, H.; Ureba, A.; Arráns, R.; Carrasco-Peña, F.; Illescas-Vacas, A.; Leal, A. Clinical implementation of combined modulated electron and photon beams with conventional MLC for accelerated partial breast irradiation. Radiotherapy and Oncology 2017, 124, 124–129, doi:10.1016/j.radonc.2017.06.011.

Mazzucconi, D.; Agosteo, S.; Ferrarini, M.; Fontana, L.; Lante, V.; Pullia, M.; Savazzi, S. Mixed particle beam for simultaneous treatment and online range verification in carbon ion therapy: Proof‐of‐concept study(2018)

[113]

Berg, R.J.; de Gruijl, F.R.; van der Leun, J.C. Interaction between ultraviolet A and ultraviolet B radiations in skin cancer induction in hairless mice. Cancer Research 1993, 53, 4212–4217.

Haraf, D.J.; Rubin, S.J.; Sweeney, P.; Kuchnir, F.T.; Sutton, H.G.; Chodak, G.W.; Weichselbaum, R.R. Photon neutron mixed-beam radiotherapy of locally advanced prostate cancer (1995)

[116]

Laramore, G.E.; Diener-west, M.; Griffin, T.W.; Nelson, J.S.; Griem, M.L.; Thomas, F.J.; Hendrickson, F.R.; Griffin, B.R.; Myrianthopoulos, L.C.; Saxton, J. Randomized neutron dose searching study for malignant gliomas of the brain: Results of an RTOG study. International Journal of Radiation Oncology*Biology*Physics 1988, 14, 1093–1102, doi:10.1016/0360-3016(88)90384-7.

Griffin, T.W.; Davis, R.; Laramore, G.E.; Maor, M.H.; Hendrickson, F.R.; Rodriguez-Antunez, A.; Davis, L. Mixed beam radiation therapy for unresectable squamous cell carcinomas of the head and neck: the results of a randomized RTOG study (1984)

[117]

Daneshvar, F.; Salehi, F.; Karimi, M.; Vais, R.D.; Mosleh-Shirazi, M.A.; Sattarahmady, N. Combined X-ray radiotherapy and laser photothermal therapy of melanoma cancer cells using dual-sensitization of platinum nanoparticles. Journal of Photochemistry and Photobiology B: Biology 2020, 203, 111737, doi:10.1016/j.jphotobiol.2019.111737.

Griffin, T.W.; Pajak, T.F.; Maor, M.H.; Laramore, G.E.; Hendrickson, F.R.; Parker, R.G.; Thomas, F.J.; Davis, L.W. Mixed neutron/photon irradiation of unresectable squamous cell carcinomas of the head and neck: The final report of a randomized cooperative trial (1989)

[118]

Shanmugam, M.; Kuthala, N.; Kong, X.; Chiang, C.-S.; Hwang, K.C. Combined Gadolinium and Boron neutron capture therapies for eradication of head-and-neck tumor using Gd10 B6 nanoparticles under MRI/CT image guidance. JACS Au 2023, 3, 2192–2205, doi:10.1021/jacsau.3c00250.

Blanco, Y.; De Diego-Castilla, G.; Viúdez-Moreiras, D.; Cavalcante-Silva, E.; Rodríguez-Manfredi, J.A.; Davila, A.F.; McKay, C.P.; Parro, V. Effects of gamma and electron radiation on the structural integrity of organic molecules and macromolecular biomarkers measured by microarray immunoassays and their astrobiological implications (2018)

[128]

Bishawi, M.; Lee, F.H.; Abraham, D.M.; Glass, C.; Blocker, S.J.; Cox, D.J.; Brown, Z.D.; Rockman, H.A.; Mao, L.; Slaba, T.C.; et al. Late onset cardiovascular dysfunction in adult mice resulting from galactic cosmic ray exposure. iScience 2022, 25, 104086, doi:10.1016/j.isci.2022.104086.

Kumar, K.; Moon, B.-H.; Kumar, S.; Angdisen, J.; Kallakury, B.V.S.; Fornace, A.J.; Suman, S. Senolytic agent ABT-263 mitigates low- and high-LET radiation-induced gastrointestinal cancer development in Apc1638N/+ mice(2025)

[170]

Boutros, S.W.; Zimmerman, B.; Nagy, S.C.; Lee, J.S.; Perez, R.; Raber, J. Amifostine (WR-2721) mitigates cognitive injury induced by heavy ion radiation in male mice and alters behavior and brain connectivity. Frontiers in Physiology 2021, 12, 770502, doi:10.3389/fphys.2021.770502.

Comment 6:

The text related to each table should be placed immediately after the individual table, rather than grouping all the tables together at the beginning of the manuscript.

Response 6:

This is standard practice in MDPI journals, so thank you for pointing this one out. We moved the tables where they should be.

Comment 7:

The content in the Discussion section is somewhat redundant. There are several repetitive points that were already mentioned in the Results section. For instance, the observation that mixed radiation fields can reshape transcriptional networks, even in the absence of overt cytotoxicity, and that the NASA Twins Study provided translational support mirroring results observed in rodent and cell-based GCR simulations, among others.

Response 7:

We recognize that portions of the Discussion section repeated points already presented in the Results. In response to this, we have restructured both the Results and the Discussion sections; in the Discussion Section, we tried to build more purposefully on the Results, reducing redundancy and placing greater emphasis on interpretative insights and broader implications.

General Remarks

We would like to clarify that, in order to integrate all of the constructive suggestions made by the reviewers and to ensure compliance with the PRISMA 2020 reporting guidelines, several changes were made to almost all texts and to many of the table contents of this systematic review.

Reviewer 2 Report

Comments and Suggestions for Authors

General comments:

The review article entitled “Combined Radiations: Biological Effects of Mixed Exposures Across the Radiation Spectrum”, discusses the studies examining combinations of ionizing and non-ionizing radiation types, including ultraviolet light, X-rays, gamma rays, alpha and beta particles, protons, neutrons, and heavy ions in the three domains viz., radiobiological, therapeutic, and space radiation. Emphasizing mixed radiation fields is particularly relevant given scenarios like space exploration, nuclear incidents, and the development of new treatment strategies such as combined modality cancer therapies. This review deepens our insight into the biological interactions between various radiation types, an area with critical implications for public health and national defense. The review is timely required; nicely presented and very interesting. The authors precisely collected the data and organised in such a way that, the readers can be benefitted directly. The review is well structured, addressed about Methodological Challenges and Experimental Gaps as well as Limitations and Future Directions. The thought and the effort from the authors commendable.

Strengths of the Work:

  1. This comprehensive review highlights strong speculative work by bringing together recent research and theoretical insights. It examines how different types of radiation both ionizing and non-ionizing interact, whether by enhancing or counteracting each other’s effects, offering a deeper understanding to the existing knowledge of literature.
  2. The text unfolds in a coherent sequence, starting with basic mechanisms and gradually moving toward practical applications. The scientific content builds logically on the last, sustaining reader interest and reinforcing essential ideas throughout.
  3. By integrating perspectives from radiobiology, therapeutic and space radiation effects and risk assessment, this review offers a multi-dimensional understanding of mixed radiation effects. This makes it valuable for a broad scientific audience.

Minor comments

Q1: “publications available up to June 20, 2025”. If authors mention the starting year (Block period), it would be nice to readers to see the quantum of years and efforts made by radiobiologist across the world to understand the biological effects.

                                         *********************************

Author Response

Combined Radiations: Biological Effects of Mixed Exposures
Across the Radiation Spectrum

Parousis-Paraskevas et al., Biomolecules, MDPI

1st Submission Date: 23 June 2025
Date of 1st Review: 16 Jul 2025 01:58:16

Response to the Comments Made by Reviewer 2

Dear Reviewer 2,

We would like to thank you for taking the time to go through our manuscript and provide us with useful feedback. Below is given a detailed answer to your specific comment.  

Comment 1:

“publications available up to June 20, 2025”. If authors mention the starting year (Block period), it would be nice to readers to see the quantum of years and efforts made by radiobiologist across the world to understand the biological effects.

Response 1:

We agree that indicating the starting year of the literature survey provides valuable context and helps convey the span of research efforts in this field. Accordingly, we have now specified the publication range as “from the 1960s to June 20, 2025” and added the sentence “This temporal range emerged organically due to the historical spread of relevant studies and is indicative of the sustained and evolving nature of radiobiological research over multiple decades” to highlight the historical scope and cumulative contributions of the radiobiology community.

General Remarks

We would like to clarify that, in order to integrate all of the constructive suggestions made by the reviewers and to ensure compliance with the PRISMA 2020 reporting guidelines, several changes were made to almost all texts and to many of the table contents of this systematic review.

Reviewer 3 Report

Comments and Suggestions for Authors

This is an outstanding and valuable review of the literature on the effects of multiple radiation types. It can be published as is. I have three general suggestions which the authors may want to address:

(1) The issue of synergistic effects at the extemely low dose rates of ambient space radiation, which it is not feasible to simulate at ground-based facilities

(2) The potential synergistic effects of combined radiations administered at significantly different dose rates.

(3) In Section 4.6 include some illustrative examples of existing shared databases, such as the NASA Open Science Data Repository.

Author Response

Combined Radiations: Biological Effects of Mixed Exposures
Across the Radiation Spectrum

Parousis-Paraskevas et al., Biomolecules, MDPI

1st Submission Date: 23 June 2025
Date of 1st Review: 16 Jul 2025 01:58:16

Response to the Comments Made by Reviewer 3

Dear Reviewer 3,

For a start, we would like to thank you for taking the time to go through our manuscript and compile such useful and insightful comments. Below are given the detailed answers to your specific comments.  

Comment 1:

The issue of synergistic effects at the extremely low dose rates of ambient space radiation, which it is not feasible to simulate at ground-based facilities

Response 1:

Thank you for reminding us of this important limitation. In response, we have now added a related statement in Section 4.6.: “For example, the extremely low dose rates characteristic of ambient space radiation—where synergistic effects may unfold gradually over time—remain experimentally inaccessible at ground-based facilities, which reflects a fundamental limitation in our ability to model the biological complexity of real spaceflight conditions.”

Comment 2:

The potential synergistic effects of combined radiations administered at significantly different dose rates.

Response 2:

In response to this constructive comment, we have added the following sentence in Section 4.6: “In addition, potential synergistic effects may be shaped not only by radiation type and timing, but also by differences in dose rates among combined exposures—an underexplored variable that may significantly influence biological outcomes and remains difficult to replicate experimentally.”

Comment 3:

In Section 4.6 include some illustrative examples of existing shared databases, such as the NASA Open Science Data Repository.

Response 3:

A nice observation indeed, thank you. We added the following text in 4.6.: “Examples of such efforts include the NASA Open Science Data Repository and the European Radiobiological Archives (ERA), hosted by the German Federal Office for Radiation Protection (BfS), which offer structured datasets from both spaceflight and ground-based experiments.”

General Remarks

We would like to clarify that, in order to integrate all of the constructive suggestions made by the reviewers and to ensure compliance with the PRISMA 2020 reporting guidelines, several changes were made to almost all texts and to many of the table contents of this systematic review.